# Understanding Convergence and Generalization in Federated Learning through Feature Learning Theory

**Wei Huang**
RIKEN AIP
`wei.huang.vr@riken.jp`

**Ye Shi**[*]
Shanghaitech University
`shiye@shanghaitech.edu.cn`

**Zhongyi Cai**
Shanghaitech University
`caizhy@shanghaitech.edu.cn`

**Taiji Suzuki**
The University of Tokyo & RIKEN AIP
`taiji@mist.iu-tokyo.ac.jp`

## Abstract

Federated Learning (FL) has attracted significant attention as an efficient privacy-preserving approach to distributed learning across multiple clients. Despite extensive empirical research and practical applications, a systematic way to theoretically understand the convergence and generalization properties in FL remains limited. This work aims to establish a unified theoretical foundation for understanding FL through feature learning theory. We focus on a scenario where each client employs a two-layer convolutional neural network (CNN) for local training on their own data. Many existing works analyze the convergence of Federated Averaging (FedAvg) under lazy training with linearizing assumptions in weight space. In contrast, our approach tracks the trajectory of signal learning and noise memorization in FL, eliminating the need for these assumptions. We further show that FedAvg can achieve near-zero test error by effectively increasing signal-to-noise ratio (SNR) in feature learning, while local training without communication achieves a large constant test error. This finding highlights the benefits of communication for generalization in FL. Moreover, our theoretical results suggest that a weighted FedAvg method, based on the similarity of input features across clients, can effectively tackle data heterogeneity issues in FL. Experimental results on both synthetic and real-world datasets verify our theoretical conclusions and emphasize the effectiveness of the weighted FedAvg approach.

## 1 Introduction

Federated Learning (FL) (McMahan et al., 2017; Yang et al., 2019) is a distributed learning framework that involves collaboratively training a global model with multiple clients while ensuring privacy protection. The pioneering work, Federated Averaging (FedAvg) (McMahan et al., 2017), learns the global model by aggregating the local client models. It achieves satisfactory performance with both IID and non-IID client data. Since then, with the growing importance of privacy protection, federated learning emerged as a hot area of research and application (Bonawitz et al., 2019; Li et al., 2020a;b; 2023b). Numerous technologies and methods began to surface, encompassing a wider range of fields and application scenarios, including smartphone applications (Li et al., 2019; Yang et al., 2021), Internet of Things (IoT) devices (Nguyen et al., 2021; Khan et al., 2021), and more.

Despite federated learning's success, there has been limited research on understanding its convergence and generalization properties when neural networks are used as client models. From a theoretical perspective, many recent studies focus on the convergence of FedAvg using over-parameterized neural networks. For example, Song et al. (2023) show FedAvg converges linearly to a solution that achieves (almost) zero training loss through a special initialization strategy and network overparameterization. Besides, Huang et al. (2021) present a class of convergence analysis for FL with the

---

[*]Corresponding author

help of a neural tangent kernel. Similarly, Deng et al. (2022) prove that FedAvg can optimize deep neural networks with ReLU activation function in polynomial time. However, these analyses often rely on linearizing assumptions in the weight space. Moreover, current generalization analyses in FL draw from classical theories like VC dimension (Hu et al., 2022), PAC Bayes (Sefidgaran et al., 2023), and convex loss (Chen et al., 2021). These theories often require strong assumptions, such as Lipschitz continuity and (strong) convexity. As a result, they may not fully capture the success of FL. The above works analyze either the convergence or generalization properties of FL, lacking a unified theoretical framework for FL with the neural network.

To fill the theoretical gap in FL, we delve into the feature learning theory (Allen-Zhu & Li, 2023; Cao et al., 2022) to establish a unified theoretical framework to analyze both the convergence and generalization properties of FedAvg. We introduce a two-level data distribution that simulates data collection in FL. For the input feature, we use a two-patch model that mimics the latent structure of an image, containing both signal patches and noise patches. The signal patch is correlated to the label while the noise patch represents the background of the image and has nothing to do with the label. We then study the dynamics of a two-layer convolutional neural network in each client with FedAvg by comparing the learning speed of signal and noise. Once we show that the neural network in FedAvg achieves near zero training error, we then compare the generalization ability between FedAvg and the local algorithm where no communication is allowed. Under a certain condition, FedAvg can converge to the global minimum with small test errors, while the local training achieves a high test error as it memorizes the noise from the training data. This demonstrates the advantage of federated learning over standard local learning (without communication) with the help of communication between clients. Our contributions are summarized as follows:

- We demonstrate the convergence properties of the FedAvg algorithm from a feature-learning perspective. This result goes beyond the conventional lazy training approach, which relies on linearizing assumptions in weight space to ensure network parameters remain close to their initialization values throughout training. In contrast, our analysis does not depend on such assumptions and reveals that FedAvg can achieve a near-zero training error within polynomial time.

- To the best of our knowledge, we are the first to rigorously establish the exact generalization gap between FedAvg and local algorithms by leveraging effective signal-to-noise ratio in feature learning theory. Specifically, under the same conditions, we demonstrate that the test error of FedAvg exhibits an asymptotic behavior of $o(1)$, whereas local training converges to a $\Theta(1)$ test error. Additionally, our theoretical result reveals the importance of cooperation in FL based on the similarity between input data among clients from feature learning theory.

- We have conducted comprehensive experiments on both synthetic and real-world datasets. The simulation results validate our theoretical assertion regarding the superiority of weighted FedAvg, a personalization method, and provide empirical support for our insights into the cooperative aspects of this approach.

## 2 RELATED WORK

**Federated Learning** In FL, the global model is generated by weighted aggregation of the uploaded local models, where the weight is proportional to the local data sizes (McMahan et al., 2017; Yang et al., 2019). However, when the data among clients is not independently and identically distributed, the global model produced by this aggregation mechanism may fail to fit either the global data distribution or the local data distribution (Zhao et al., 2018; Li et al., 2022; Cao et al., 2023; Cai et al., 2023; Li et al., 2024). To address issues arising from data heterogeneity, various aggregation methods have been proposed. For instance, Li et al. (2022) leveraged validation sets to learn aggregation weights tailored to each client's local data distribution. In a different approach, Xu et al. (2023) addressed an optimization problem for each client, identifying an optimal linear combination of local classifier heads to serve as the new local classifier. Meanwhile, Li et al. (2023c) utilized a proxy dataset to learn both the aggregation weights and the global weight shrinking factor. Additionally, Ye et al. (2023) adjusted aggregation weights based on the disparities between local and global category distributions to enhance performance.

**Feature Learning Theory** Different from neural tangent kernel (Jacot et al., 2018) or lazy training Chizat et al. (2019), where weight updates during gradient descent training are relatively small,

feature learning theory (Allen-Zhu & Li, 2023; Cao et al., 2022; Yang & Hu, 2021) states that weight updates can be more substantial during gradient descent, allowing the network to learn patterns from the data. The feature learning has been widely explored to provide interpolation to various neural networks, such as graph neural network (Huang et al., 2023), convolutional neural network (Cao et al., 2022; Kou et al., 2023), vision transformer (Jelassi et al., 2022; Li et al., 2023a); and different training algorithms like Adam (Zou et al., 2023a), momentum (Jelassi & Li, 2022), OOD Chen et al. (2023), and Mixup (Zou et al., 2023b; Chidambaram et al., 2023). Among the works related to feature learning, Kou et al. (2023) investigate the optimization and generalization of two-layer ReLU CNNs in centralized training scenarios. In contrast, our research delves into the application of feature learning theory in Federated Learning (FL), introducing additional techniques tailored for the FedAvg algorithm.

**Theoretical Analysis of FL**   The convergence analysis of federated learning has been gaining increasing attention. Initially, researchers explored convergence in FL without employing neural network models for clients. For instance, Li et al. (2020c) analyzed the convergence of FedAvg on non-iid data, establishing convergence rates for strongly convex and smooth problems. Wang et al. (2020) provided a comprehensive framework for analyzing the convergence of federated optimization algorithms, considering heterogeneous local training progress at clients. Subsequently, the advent of over-parameterized neural networks sparked a new wave of convergence analysis for FL (Huang et al., 2021; Deng et al., 2022; Song et al., 2022). These convergence analyses may highly rely on linearizing assumptions in weight space to ensure network parameters remain close to their initialization. In terms of generalization in FL, much of the existing research has been grounded in classical theories that often require strong assumptions, such as Lipschitz continuity and (strong) convexity. For example, works like (Mohri et al., 2019; Chen et al., 2021; Masiha et al., 2021; Hu et al., 2022; Sun et al., 2023) have presented generalization results in FL based on these assumptions. Another research strand delved into variance reduction strategies for heterogeneous FL, as exemplified by works like (Woodworth et al., 2020; Murata & Suzuki, 2021; Oko et al., 2022). In contrast to these approaches, our work focuses on both the convergence and generalization of neural networks in FL based on the feature learning theory, offering a unified theoretical framework for FL.

## 3 PRELIMINARY

### 3.1 NOTATIONS

We use lower bold-faced letters for vectors, upper bold-faced letters for matrices, and non-bold-faced letters for scalars. For a vector $\mathbf{v}$, its $\ell_2$-norm is denoted as $\|\mathbf{v}\|_2$. For a matrix $\mathbf{A}$, we use $\|\mathbf{A}\|_2$ to denote its spectral norm and $\|\mathbf{A}\|_F$ for its Frobenius norm. When comparing two sequences, we employ standard asymptotic notations such as $O(\cdot)$, $o(\cdot)$, $\Omega(\cdot)$, and $\Theta(\cdot)$ to describe their limiting behavior. We use $\widetilde{O}(\cdot)$, $\widetilde{\Omega}(\cdot)$, and $\widetilde{\Theta}(\cdot)$ to hide logarithmic factors in these notations respectively. Lastly, sequences of integers are denoted as $[n] = \{1, 2, \ldots, n\}$.

### 3.2 DATA MODEL

We assume there are $K$ clients, and each client is associated with a distribution $\mathcal{D}_k$. In this work, we adopt a two-level distribution framework for FL (Yuan et al., 2021; Hu et al., 2022; Wang et al., 2021). In the first level, we select a signal vector $\boldsymbol{\mu}_k$ for client $k$ through $\boldsymbol{\mu}_k \sim P(\boldsymbol{\mu}^{(1)}, \boldsymbol{\mu}^{(2)}, \cdots, \boldsymbol{\mu}^{(C)})$, where $P$ is a discrete distribution assigning probabilities to each feature vector $\boldsymbol{\mu}^{(c)}$ for $c \in [C]$. For simplicity, we assume every $\boldsymbol{\mu}_c$ for $c \in [C]$ are orthogonal to each other. In the second level, the input feature associated to each client are sampled based on $\boldsymbol{\mu}_k$. In particular, we utilize a two-patch model for feature generation (Allen-Zhu & Li, 2023; Shen et al., 2022; Huang et al., 2023; Kou et al., 2023; Cao et al., 2022). Conditional on the Rademacher random variable $y_{k,i} \in \{-1, 1\}$, for each client $k \in [K]$, the input with index $i$ is generated from:

$$\mathbf{x}_{k,i} = [\mathbf{x}_{k,i}^{(1)}, \mathbf{x}_{k,i}^{(2)}] = [y_{k,i}\boldsymbol{\mu}_k, \boldsymbol{\xi}_{k,i}], \tag{1}$$

where noise vector $\boldsymbol{\xi}_{k,i} \sim \mathcal{N}(\mathbf{0}, \sigma_p^2(\mathbf{I} - \sum_{c=1}^{C} \boldsymbol{\mu}^{(c)}\boldsymbol{\mu}^{(c)\top}/\|\boldsymbol{\mu}^{(c)}\|_2^2))$, with $\sigma_p$ being the strength of noise. Note that $\sum_{c=1}^{C} \boldsymbol{\mu}^{(c)}\boldsymbol{\mu}^{(c)\top}/\|\boldsymbol{\mu}^{(c)}\|_2^2$ is introduced to ensure noise vector orthogonal to signal

vector. We can recover IID by setting $\boldsymbol{\mu}_1 = \boldsymbol{\mu}_2 = \cdots = \boldsymbol{\mu}_K$. Note that we use a two-patch model that mimics the latent structure of an image. The signal patch is correlated to the label while the noise patch represents the background of the image and has nothing to do with the label.

## 3.3 NEURAL NETWORK MODEL

We introduce a two-layer CNN model, denoted as $f_k$, for every client $k \in [K]$, which utilizes a ReLU activation function, defined as $\sigma(z) = \max\{0, z\}$. Given the input data $\mathbf{x}$, the CNN's output is represented as $f_k(\mathbf{W}, \mathbf{x}) = F_{+1}(\mathbf{W}_{+1}, \mathbf{x}) - F_{-1}(\mathbf{W}_{-1}, \mathbf{x})$. The sign is associated with the neuron of second layer which are fixed as either $+1$ or $-1$. Furthermore, we define $F_{+1}(\mathbf{W}_{+1}, \mathbf{x})$ and $F_{-1}(\mathbf{W}_{-1}, \mathbf{x})$ as follows:

$$F_j(\mathbf{W}_j, \mathbf{x}) = \frac{1}{m} \sum_{r=1}^{m} \left[ \sigma(\mathbf{w}_{k,j,r}^\top \mathbf{x}^{(1)}) + \sigma(\mathbf{w}_{k,j,r}^\top \mathbf{x}^{(2)}) \right], \tag{2}$$

where $m$ is the width, and $\mathbf{w}_{k,j,r} \in \mathbb{R}^d$ refers to the weight vector of the first layer in client $k$. The symbol $\mathbf{W}$ collectively represents the model's weights. Moreover, each weight in the first layer is initialized from a random draw of a Gaussian random variable, $\mathbf{w}_{j,r} \sim \mathcal{N}(\mathbf{0}, \sigma_0^2 \mathbf{I}_{d \times d})$ for all $r \in [m]$ and $j \in \{-1, 1\}$, with $\sigma_0$ regulating the initialization magnitude for the first layer's weight.

## 3.4 OBJECTIVE FUNCTION AND FEDAVG

We denote the training set as $\mathcal{S}_k \triangleq \{\mathbf{x}_{k,i}, y_{k,i}\}_{i=1}^{n_k}$ for $k \in [K]$, and overall training set as $\mathcal{S} \triangleq \{\mathcal{S}_k\}_{k=1}^K$. We aim to minimize the empirical cross-entropy loss function:

$$L_{\mathcal{S}}(\mathbf{W}) = \sum_{k=1}^{K} \frac{n_k}{n} L_{\mathcal{S}_k}(\mathbf{W}_k) \quad \text{with} \quad L_{\mathcal{S}_k}(\mathbf{W}_k) = \frac{1}{n_k} \sum_{i=1}^{n_k} \ell(y_{k,i} f(\mathbf{W}_k, \mathbf{x}_{k,i})), \tag{3}$$

where $\ell(y f(\mathbf{W}, \mathbf{x})) = \log(1 + \exp(-f(\mathbf{W}, \mathbf{x})y))$ is the logistic loss. Next, we introduce the FedAvg training algorithm, which comprises two types of updates: gradient descent updates and weight averaging updates. In each iteration, we first apply gradient descent by $E$ epochs to the CNN $f_k$ for client $k \in [K]$:

$$\mathbf{w}_{k,j,r}^{(t+1)} = \mathbf{w}_{k,j,r}^{(t)} - \eta \nabla_{\mathbf{w}_{k,j,r}} L_S(\mathbf{W}_k^{(t)}) = \mathbf{w}_{k,j,r}^{(t)} - \frac{\eta}{n_k m} \sum_{i=1}^{n_k} \ell_{k,i}'^{(t)} \sigma'(\langle \mathbf{w}_{k,j,r}^{(t)}, \boldsymbol{\xi}_{k,i} \rangle) j y_{k,i} \boldsymbol{\xi}_{k,i}$$

$$- \frac{\eta}{n_k m} \sum_{i=1}^{n_k} \ell_{k,i}'^{(t)} \sigma'(\langle \mathbf{w}_{k,j,r}^{(t)}, y_{k,i} \cdot \boldsymbol{\mu}_k \rangle) j \boldsymbol{\mu}_k, \tag{4}$$

where we define the loss derivative as $\ell_{k,i}' \triangleq \ell'(y_{k,i} f_{k,i}) = -\frac{\exp(-y_{k,i} f_{k,i})}{1 + \exp(-y_{k,i} f_{k,i})}$. Then the server update through weight averaging can be expressed as follows:

$$\overline{\mathbf{w}}_{j,r}^{(t)} = \sum_{k=1}^{K} \frac{n_k}{n} \mathbf{w}_{k,j,r}^{(t)}, \tag{5}$$

where $\overline{\mathbf{w}}_{j,r}$ is the weight at server, $n = \sum_{k=1}^K n_k$ is the total number of training samples. After a certain number $R$ of training loops until convergence, we examine the training through test error defined as $L_{\mathcal{D}_k}^{0-1}(\overline{\mathbf{W}}^{(t)}) = P_{(\mathbf{x},y) \sim \mathcal{D}_k}[y f(\overline{\mathbf{W}}^{(t)}, \mathbf{x}) < 0]$.

## 4 MAIN RESULTS

### 4.1 COEFFICIENT DYNAMICS FOR FEDERATED LEARNING

In this section, we introduce our key theoretical findings that demonstrate the convergence and generalization of federated learning. Through the application of the gradient descent rule outlined in Equation (4) and Equation (5), we observe that local weights $\mathbf{w}_{k,j,r}^{(t)}$ for all client $k \in [K]$ and

server weights $\overline{\mathbf{w}}_{j,r}^{(t)}$ are a linear combination of the random initialization $\mathbf{w}_{k,j,r}^{(0)}$, the signal vectors $\boldsymbol{\mu}_k$, and the noise vectors $\boldsymbol{\xi}_{k,i}$ for $i \in [n_k]$ and $k \in [K]$. Consequently, for $r \in [m]$, and $k \in [K]$, the decomposition of weight vector iteration can be expressed:

$$\mathbf{w}_{k,j,r}^{(t)} = \sum_{k'=1}^{K}(\alpha_{k,k',j,r}^{(t)}\mathbf{w}_{k',j,r}^{(0)} + \gamma_{k,k',j,r}^{(t)}\|\boldsymbol{\mu}_{k'}\|_2^{-2}\boldsymbol{\mu}_{k'} + \sum_{i=1}^{n_{k'}}\rho_{k,k',j,r,i}^{(t)}\|\boldsymbol{\xi}_{k',i}\|_2^{-2}\boldsymbol{\xi}_{k',i}), \quad (6)$$

$$\overline{\mathbf{w}}_{j,r}^{(t)} = \sum_{k=1}^{K}(\overline{\alpha}_{k,j,r}^{(t)}\mathbf{w}_{k,j,r}^{(0)} + \overline{\gamma}_{k,j,r}^{(t)}\|\boldsymbol{\mu}_k\|_2^{-2}\boldsymbol{\mu}_k + \sum_{i=1}^{n_k}\overline{\rho}_{k,j,r,i}^{(t)}\|\boldsymbol{\xi}_{k,i}\|_2^{-2}\boldsymbol{\xi}_{k,i}). \quad (7)$$

where $\gamma_{k,k',j,r}^{(t)}$, $\overline{\gamma}_{k,j,r}^{(t)}$ and $\rho_{k,k',j,r,i}^{(t)}$, $\overline{\rho}_{k,j,r,i}^{(t)}$ for $k, k' \in [K]$, $j \in \{-1,1\}$, $r \in [m]$, $i \in [n_k]$, serve as coefficients. We refer to Equation (6) and Equation (7) as the signal-noise decomposition of $\mathbf{w}_{k,j,r}^{(t)}$. The normalization factors $\|\boldsymbol{\mu}_k\|_2^{-2}$ and $\|\boldsymbol{\xi}_{k,i}\|_2^{-2}$ are introduced to ensure that $\gamma_{k,k',j,r}^{(t)} \approx \langle\mathbf{w}_{k,j,r}^{(t)}, \boldsymbol{\mu}_{k'}\rangle$, and $\rho_{k,k',j,r,i}^{(t)} \approx \langle\mathbf{w}_{k,j,r}^{(t)}, \boldsymbol{\xi}_{k',i}\rangle$. To facilitate a fine-grained analysis for the evolution of coefficients, we introduce the notations $\psi_{k,k',j,r,i}^{(t)} \triangleq \rho_{k,k',j,r,i}^{(t)}\mathbb{1}(\rho_{k,k',j,r,i}^{(t)} \geq 0)$, $\phi_{k,k',j,r,i}^{(t)} \triangleq \rho_{k,k',j,r,i}^{(t)}\mathbb{1}(\rho_{k,k',j,r,i}^{(t)} \leq 0)$. Consequently, we further express the vector weight decomposition (6,7) as:

$$\mathbf{w}_{k,j,r}^{(t)} = \sum_{k'=1}^{K}(\alpha_{k,k',j,r}^{(t)}\mathbf{w}_{k',j,r}^{(0)} + \gamma_{k,k',j,r}^{(t)}\|\boldsymbol{\mu}_{k'}\|_2^{-2}\boldsymbol{\mu}_{k'} + \sum_{i=1}^{n_{k'}}(\psi_{k,k',j,r,i}^{(t)} + \phi_{k,k',j,r,i}^{(t)})\|\boldsymbol{\xi}_{k',i}\|_2^{-2}\boldsymbol{\xi}_{k',i}), \quad (8)$$

$$\overline{\mathbf{w}}_{j,r}^{(t)} = \sum_{k=1}^{K}(\overline{\alpha}_{k,j,r}^{(t)}\mathbf{w}_{k,j,r}^{(0)} + \overline{\gamma}_{k,j,r}^{(t)}\|\boldsymbol{\mu}_k\|_2^{-2}\boldsymbol{\mu}_k + \sum_{i=1}^{n_k}(\overline{\psi}_{k,j,r,i}^{(t)} + \overline{\phi}_{k,j,r,i}^{(t)})\|\boldsymbol{\xi}_{k,i}\|_2^{-2}\boldsymbol{\xi}_{k,i}). \quad (9)$$

Our first key result is to turn the dynamics of weights into the dynamics of coefficients. To analyze the feature learning process of federated learning during gradient descent training, we introduce an iterative methodology, based on the signal-noise decomposition in (8) and (9) and gradient descent update (4) and (5). The following lemma offers us a means to monitor the iteration of coefficients:

**Lemma 4.1.** *The coefficients* $\gamma_{k,k',j,r}^{(t)}$, $\psi_{k,k',j,r,i}^{(t)}$, $\phi_{k,k',j,r,i}^{(t)}$ *in decomposition (8), for all* $k, k' \in [K]$, *follow the update rules governed by:*

$$\gamma_{k,k',j,r}^{(0)}, \psi_{k,k',j,r,i}^{(0)}, \phi_{k,k',j,r,i}^{(0)} = 0, \quad (10)$$

$$\gamma_{k,k',j,r}^{(t+1)} = \gamma_{k,k',j,r}^{(t)} - \frac{\eta}{n_k m}\sum_{i=1}^{n_k}\ell_{k,i}'^{(t)}\sigma'(\langle\mathbf{w}_{k,j,r}^{(t)}, y_{k,i}\boldsymbol{\mu}_k\rangle)\|\boldsymbol{\mu}_k\|_2^2\mathbb{1}(k'=k), \quad (11)$$

$$\psi_{k,k',j,r,i}^{(t+1)} = \psi_{k,k',j,r,i}^{(t)} - \frac{\eta}{n_k m}\ell_{k,i}'^{(t)}\sigma'(\langle\mathbf{w}_{k,j,r}^{(t)}, \boldsymbol{\xi}_{k,i}\rangle)\|\boldsymbol{\xi}_{k,i}\|_2^2\mathbb{1}(y_{k,i}=j)\mathbb{1}(k'=k), \quad (12)$$

$$\phi_{k,k',j,r,i}^{(t+1)} = \phi_{k,k',j,r,i}^{(t)} + \frac{\eta}{n_k m}\ell_{k,i}'^{(t)}\sigma'(\langle\mathbf{w}_{k,j,r}^{(t)}, \boldsymbol{\xi}_{k,i}\rangle)\|\boldsymbol{\xi}_{k,i}\|_2^2\mathbb{1}(y_{k,i}=-j)\mathbb{1}(k'=k). \quad (13)$$

*Furthermore, the coefficients* $\overline{\gamma}_{k,j,r}^{(t)}$, $\overline{\psi}_{k,j,r,i}^{(t)}$, $\overline{\phi}_{k,j,r,i}^{(t)}$ *in decomposition (9), for all* $k \in [K]$, *are updated according to the following rules:*

$$\overline{\gamma}_{k,j,r}^{(t)} = \sum_{k''=1}^{K}\sum_{k'=1}^{K}\frac{n_{k'}}{n}\gamma_{k',k'',j,r}^{(t)}\mathbb{1}(\boldsymbol{\mu}_{k''}=\boldsymbol{\mu}_k), \overline{\psi}_{k,j,r,i}^{(t)} = \sum_{k'=1}^{K}\frac{n_{k'}}{n}\psi_{k',k,j,r,i}^{(t)}, \overline{\phi}_{k,j,r,i}^{(t)} = \sum_{k'=1}^{K}\frac{n_{k'}}{n}\phi_{k',k,j,r,i}^{(t)}. \quad (14)$$

The proof of Lemma 4.1 can be found in Appendix A. Lemma 4.1 provides us a new approach to track the dynamics of neural network in FL. In particular, we study the learning speed ratio between signal and noise, represented by Equations (10-14).

## 4.2 FEDAVG PROVABLY BENEFITS FROM COMMUNICATION

Before stating the main results, we demonstrate our setting and assumption for federated learning. First, without loss of generality, we set the same training size for all clients and $\ell_2$ norm of signal

vector is the same, namely $\overline{n} = n_1 = n_2 = \cdots = n_K$, and $\|\boldsymbol{\mu}^{(1)}\|_2 = \|\boldsymbol{\mu}^{(2)}\|_2 = \cdots = \|\boldsymbol{\mu}^{(C)}\|_2 = \|\boldsymbol{\mu}\|_2$. Therefore $n = \overline{n}K$. Our analysis will be made under the following set of assumptions:

**Assumption 4.2.** *Suppose that:*
*(1) The dimension $d$ is sufficiently large: $d = \tilde{\Omega}(\max\{n^2, \sigma_q^{-2}K^2\overline{n}\|\boldsymbol{\mu}\|_2^2\})$.*
*(2) The size of training sample $\overline{n}$ and width of CNNs $m$ adhere to $\overline{n} = \Omega(\text{polylog}(d))$, $m = \Omega(\text{polylog}(d))$. All network model share the same initialization.*
*(3) The learning rate $\eta$ satisfies $\eta \leq \tilde{O}(\frac{K}{E}\min\{\|\boldsymbol{\mu}\|_2^{-2}, \sigma_p^{-2}d^{-1}\})$. The standard deviation of Gaussian initialization $\sigma_0$ is chosen such that $\tilde{\Theta}(n)/(\sigma_p d) \leq \sigma_0 \leq \tilde{O}(1)\cdot\min\{(\sigma_p\sqrt{d})^{-1}, \|\boldsymbol{\mu}\|_2^{-1}\}$.*

Brief explanations for each assumption are provided: (1) The requirement for a high dimension in our assumptions specifically aims to ensure that learning occurs in an adequately over-parameterized setting. (2) It's necessary for the sample size and neural network width to be at least polylogarithmic in the dimension $d$. This condition guarantees that certain statistical properties of the training data and weight initialization are maintained. (3) The condition on $\eta$ and initialization $\sigma_0$ is to ensure that gradient descent can effectively minimize the training loss.

Finally, we introduce a critical quantity that plays an important role in our analysis. First, we define the local signal-to-noise ratio (SNR) for each client as $\text{SNR}_k = \|\boldsymbol{\mu}_k\|_2/(\sigma_p\sqrt{d})$ and the effective signal-to-noise ratio in FedAvg, represented as $\overline{\text{SNR}}_k = \sqrt{\sum_{k'=1}^{K}\frac{\langle\boldsymbol{\mu}_k,\boldsymbol{\mu}_{k'}\rangle}{\|\boldsymbol{\mu}_k\|_2^2}}\text{SNR}_k \triangleq \sqrt{\chi_k}\text{SNR}_k$, where we define $\chi_k = \sum_{k'=1}^{K}\frac{\langle\boldsymbol{\mu}_k,\boldsymbol{\mu}_{k'}\rangle}{\|\boldsymbol{\mu}_k\|_2^2}$. The signal-to-noise ratio serves as a measure of the relative learning speed between the signal and noise. Given the aforementioned assumptions and definitions, we present our main result on feature learning of FedAvg as follows:

**Theorem 4.3** (Convergence of FedAvg). *Suppose $\epsilon > 0$, and let $T = \tilde{\Theta}(\eta^{-1}Km\overline{n}\sigma_p^{-2}d^{-1} + \eta^{-1}\epsilon^{-1}m\overline{n}\sigma_p^{-2}K^2d^{-1})$. Under Assumption 4.2, then with probability at least $1 - d^{-1}$, there exists $0 \leq t \leq T$ such that the training loss of FedAvg converges to $\epsilon$, i.e., $L_{\mathcal{S}}(\overline{\mathbf{W}}^{(t)}) \leq \epsilon$.*

Theorem 4.3 demonstrates the global convergence of FedAvg on the training dataset by tracking the signal learning and noise memorization trajectory. We demonstrate the detailed proof sketch for convergence through feature learning theory in the next section and leave the complete proof in Appendix B. Based on results in Theorem 4.3, we illustrate the generalization of FedAvg through the following theorem:

**Theorem 4.4** (Generalization of FedAvg). *Under the same assumption as Theorem 4.3, there exists $0 \leq t \leq T$ such that if $\frac{\overline{n}\chi_k^2\|\boldsymbol{\mu}\|_2^4}{K\sigma_p^4d} = \Omega(1)$, the trained model by FedAvg achieves a small test loss: $L_{\mathcal{D}_k}^{0-1}(\overline{\mathbf{W}}^{(t)}) \leq \exp\left(-c\frac{\overline{n}\chi_k^2\|\boldsymbol{\mu}\|_2^4}{K\sigma_p^4d}\right)$. Here, $\chi_k = \sum_{k'=1}^{K}\frac{\langle\boldsymbol{\mu}_k,\boldsymbol{\mu}_{k'}\rangle}{\|\boldsymbol{\mu}_k\|_2^2}$. Otherwise, if $\frac{\overline{n}\|\boldsymbol{\mu}\|_2^4}{\sigma_p^4d} = O(1)$, the trained model by FedAvg achieves a large constant test error $L_{\mathcal{D}_k}^{0-1}(\overline{\mathbf{W}}^{(t)}) = \Omega(1)$.*

Theorem 4.4 further demonstrates the separation condition based on the effective SNR for the generalization of FedAvg. In this result, the signal-to-noise plays a significant role. When the number of sample $\overline{n}$ and effective SNR $\overline{\text{SNR}}_k$ is strong enough, FedAvg can achieve a small test error. Conversely, if these conditions are not met, FedAvg may not generalize effectively to the test samples.

Based on theorem 4.4, we draw a comparison between FedAvg and local training. Note that the SNR for FedAvg has an additional factor $\chi_k = \sum_{k'=1}^{K}\frac{\langle\boldsymbol{\mu}_k,\boldsymbol{\mu}_{k'}\rangle}{\|\boldsymbol{\mu}_k\|_2^2}$. This factor measures the overall similarity of input features across clients and can also be interpreted as a metric for data heterogeneity. In the context of local training, $\chi_k = 1$, which means the effective SNR simplifies to the vanilla SNR. Given the conditions $\frac{\overline{n}\chi_k^2\|\boldsymbol{\mu}_k\|_2^4}{K\sigma_p^4d} = \Omega(1)$ and $\frac{\overline{n}\|\boldsymbol{\mu}_k\|_2^4}{\sigma_p^4d} = O(1)$, FedAvg achieves a small test loss, while local training has a large test error.

Lastly, we interpret the effective SNR. In the case of the IID setting, we find that $\chi_k = K$ for $k \in [K]$, which can boost the performance of FedAvg mostly. On the other hand, when we consider the non-IID, the smallest value that $\chi_k$ can achieve is 1, when all the input features are different from each other. In this case, FedAvg performs worse than local training. These results point out the importance of communication in FedAvg, suggesting that weighted FedAvg can further improve the performance of FL, we examine this understanding in the experiment section.

# 5 PROOF SKETCH

In this section, we provide a proof sketch for Theorem 4.3 and Theorem 4.4. The key idea is to analyze the coefficients update function for $k \in [K]$. To achieve our optimization result, we employ a two-stage dynamic analysis. Next, we demonstrate the generalization analysis based on the expected test error.

## 5.1 ITERATIVE ANALYSIS OF COEFFICIENTS FOR CONVERGENCE

We adopt a two-stage dynamics analysis, inspired by Cao et al. (2022); Kou et al. (2023), to track the behavior of these coefficients in FL based on the behavior of the loss derivative.

**Stage 1** In the first stage, we adopt a small initialization for weights such that the neural network at initialization has constant level cross-entropy loss derivatives for all training data and clients, namely $\ell_{k,i}'^{(0)} = \ell'[y_{k,i}f(\mathbf{W}^{(0)}, \mathbf{x}_{k,i})] = \Theta(1)$ for all $i \in [\overline{n}]$ and $k \in [K]$. This is guaranteed based on assumption (4.2) regarding $\sigma_0$. Consequently, we can replace the loss derivative $\ell_{k,i}'^{(t)}$ by their constant upper and lower bounds in the analysis of the coefficients dynamics in Equations (10 - 14). This simplifies the analysis significantly, and then we present our main conclusion in Stage 1:

**Lemma 5.1.** *Under assumption 4.2, there exists $T_1 = \Theta(\eta^{-1}K\overline{n}m\sigma_p^{-2}d^{-1})$ such that*

- $\max_r \overline{\gamma}_{k,j,r}^{(T_1)} = \Theta(\overline{n} \cdot \overline{\mathrm{SNR}}_k^2)$ *for $j \in \{\pm 1\}$, and $k \in [K]$.*

- $|\overline{\psi}_{k,j,r,i}^{(t)}| = \Theta(1)$ *for some $j \in \{\pm 1\}$, $r \in [m]$, $i \in [n]$ and all $k \in [K]$.*

- $\max_{j,r} \overline{\phi}_{k,j,r,i} = O(1)$ *for all $i \in [n]$ and $k \in [K]$.*

The proof can be found in Appendix B.1. Lemmas 5.1 leverages the period of training when the derivatives of the loss function are of a constant order.

It's important to note that weight averaging plays a significant role in differentiating the learning speeds between signal learning and noise memorization during this initial stage. In each client's local update via gradient descent, the learning speeds for signal learning and noise memorization are determined by $\|\boldsymbol{\mu}\|_2$ and $\|\boldsymbol{\xi}\|_2$, respectively (Cao et al., 2022). Our theoretical analysis further reveals that, after the application of weight averaging, these learning speeds are approximately determined by $\chi_k\|\boldsymbol{\mu}_k\|_2$ and $\|\boldsymbol{\xi}\|_2$, respectively. This distinction arises because, during communication, clients with identical signal types can share their learning signal and preserve the signal magnitude. In contrast, the independence of noise vectors across clients means that their magnitude has a significant decrease. Consequently, communication in FedAvg slows down the learning speed of noise memorization, thus enabling FedAvg to focus more on signal learning.

**Stage 2** Building on the results from the first stage, we study the second stage of the training process. During this phase, the derivatives of the loss no longer remain constant. We demonstrate that the training error can be minimized to an arbitrarily small value. Importantly, the scale differences established during the first stage of learning continue to be maintained throughout the training process, and the convergence of the FedAvg in all clients can be guaranteed:

**Lemma 5.2.** *Suppose $\epsilon > 0$, let $T, T_1$ be defined in Theorem 4.3 and Lemma 5.1 respectively. Let $\overline{\mathbf{W}}^*$ be the collection of CNN parameters $\overline{\mathbf{w}}_{j,r}^* = \overline{\mathbf{w}}_{j,r}^{(0)} + \log(2/\epsilon)\sum_{k=1}^{K}(j \cdot \overline{n} \cdot \overline{\mathrm{SNR}}_k^2 \cdot \frac{\boldsymbol{\mu}_k}{\|\boldsymbol{\mu}_k\|_2^2} + \sum_{i=1}^{n} \mathbb{1}(j = y_{k,i})\frac{\boldsymbol{\xi}_{k,i}}{\|\boldsymbol{\xi}_{k,i}\|_2^2})$. Then under the same conditions as Theorem 4.3, for any $t \in [T_1, T]$, it holds that: $\frac{1}{t-T_1+1}\sum_{s=T_1}^{t} L_\mathcal{S}(\overline{\mathbf{W}}^{(s)}) \leq \frac{\|\overline{\mathbf{W}}^{(T_1)} - \overline{\mathbf{W}}^*\|_F^2}{\eta(t-T_1+1)} + \epsilon$.*

Lemma 5.2 provides a convergence result for FedAvg. It suggests that as the training time increases, the training loss approaches an arbitrarily small value.

## 5.2 TEST ERROR ANALYSIS FOR GENERALIZATION

Finally, we consider a new data point $(\mathbf{x}_k, y_k) \sim \mathcal{D}_k$ drawn from the distribution for client $k$. Theorem 4.4 indicates that FedAvg can achieve a low test error when both the sample size and effective SNR are sufficiently large. However, in the absence of these conditions, the test error achieved by FedAvg may exceed a certain constant. The proof can be found in Appendix B.3.

## 6 EXPERIMENTS

In this section, we first validate our theoretical findings through numerical simulations using synthetic data. Then we demonstrate the effectiveness of FedAvg on real-world datasets.

### 6.1 FEDERATED LEARNING ON SYNTHETIC DATA

In this section, we conduct experiments to substantiate our theoretical findings using synthetic data drawn from the two-level distribution in both IID and non-IID settings. For these experiments, the client number is set to be $K = 20$ and all of them are selected in each communication round. Each client is equipped with a two-layer ReLU CNN descried in Section 3.3, where width $m = 50$.

Then, we establish our synthetic dataset. The training data size is set to $\overline{n}_{train} = 100$ and the testing data size is set to $n_{test} = 2000$ with instance dimension to $d = 1000$ for each client. We further set the signal strength $\|\boldsymbol{\mu}\|_2 = 2$ and noise variance $\sigma_p = 1$ for all clients. We trained models for $E = 5$ epochs in the local training phase and conducted $R = 100$ communication rounds in total. All models are trained with the SGD optimizer whose learning rate is $\eta = 1$.

**Feature Learning in IID setting.** Under the IID setting, all clients share identical signal vectors, denoted as, $\boldsymbol{\mu}_1 = \boldsymbol{\mu}_2 = \cdots = \boldsymbol{\mu}_{20}$. Figure 1 displays the training loss, training accuracy, test loss, and test accuracy for both FedAvg and SingleSet (local training) on the dataset of the first client. Notably, both FedAvg and local training successfully achieve small training loss and high train accuracy, verifying results presented in Theorem 4.3. Furthermore, FedAvg displays superior test accuracy compared to local training, validating theoretical results in Theorem 4.4.

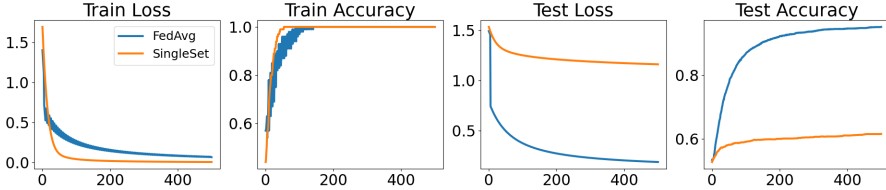

Figure 1: Convergence behavior comparison of train loss, train accuracy, test loss, and test accuracy on synthetic data. Both local training and FedAvg demonstrate convergence on the training set. FedAvg outperforms significantly on the testing set.

### 6.2 WEIGHTED FEDAVG ON REAL-WORLD DATASET

To verify our theoretical conclusion in FL with data heterogeneity issues, we conducted experiments in real-world datasets. In this case, the conventional aggregation strategy, which assigns model aggregation weights based on local data sizes, may not be optimal due to the diverse signals across clients. To address this, we enhanced collaboration between clients that exhibit similar signals. Particularly, we employ prototype similarity as a measure to approximate the signal similarity between clients. This prototype similarity is then utilized to determine the aggregation weights for each client, ensuring a more effective aggregation process.

**Weighed FedAvg** For each client $k$, we learn a well-trained model $f_k$ that is composed of a feature extractor $\mathbf{g}_k$ and a linear classifier $\mathbf{e}_k$. Supposing there are overall $M$ classes, a prototype of the $j$-th class in client $k$ is defined as $\mathbf{p}_k^j = \frac{1}{|\mathcal{S}_{k,j}|} \sum_{(\mathbf{X}, \mathbf{y}) \in \mathcal{S}_{k,j}} \mathbf{g}_k(\mathbf{X})$. We concatenate all class prototypes in client $k$ to get the client prototype $\mathbf{p}_k = \text{concat}_{i \in |M|} [\mathbf{p}_k^i]$. This client prototype is updated after

Table 1: Experimental Results for FL with Label Distribution Skew on CIFAR10 and CIFAR100.

| Methods | CIFAR10 | | CIFAR100 | |
|---|---|---|---|---|
| | Pathological | Dirichlet | Pathological | Dirichlet |
| SingleSet | 20.08±0.13 | 38.60±0.02 | 6.59±0.18 | 6.97±0.11 |
| FedAvg | 40.16±7.18 | 33.11±7.06 | 13.99±0.79 | 17.34±0.47 |
| FedPer | 38.09±5.73 | 36.52±7.13 | 29.10±2.42 | 17.63±0.93 |
| FedBN | 65.71±0.67 | 61.25±3.78 | 42.54±0.53 | 25.64±3.46 |
| Weighted FedAvg | 60.32±0.31 | 60.52±0.30 | 31.88±0.56 | 18.82±0.36 |
| + Personalized BN | **67.30±0.31** | **65.31±1.23** | **43.51±0.50** | **26.11±1.77** |

Table 2: Experimental Results for FL with Feature Skew on Digits

| Methods | Digits | | | | | |
|---|---|---|---|---|---|---|
| | MNIST | SVHN | USPS | SynthDigits | MNIST-M | Avg |
| SingleSet | 94.38±0.07 | 65.25±1.07 | 95.16±0.12 | 80.31±0.38 | 77.77±0.47 | 82.00±0.40 |
| FedAvg | 95.87±0.20 | 62.86±1.49 | 95.56±0.27 | 82.27±0.44 | 76.85±0.54 | 82.70±0.60 |
| FedPer | 96.21±0.02 | 67.61±0.04 | 96.53±0.02 | 83.88±0.02 | 76.85±0.54 | 81.89±0.03 |
| FedBN | **96.57±0.13** | 71.04±0.31 | **96.97±0.32** | 83.19±0.42 | 78.33±0.66 | 85.20±0.40 |
| Weighted FedAvg | 95.75±0.18 | 67.82±1.07 | 95.66±0.22 | 84.27±1.06 | 80.22±0.11 | 84.74±0.22 |
| + Personalized BN | 96.11±0.02 | **75.36±0.04** | 96.41±0.06 | **84.74±0.03** | **82.02±0.04** | **86.93±0.02** |

the local training phase of each involved client. Different from the normal model aggregation in FedAvg, we personalize the aggregation weights for each client model. Specifically, we determine the aggregation weight $s_{k,k'}$ of each involved client $k$ for client $k'$ based on the client prototypes, i.e., $s_{k,k'} = \frac{\exp(\text{sim}(\mathbf{p}_k, \mathbf{p}_{k'})/\tau)}{\sum_{i=1}^{K} \exp(\text{sim}(\mathbf{p}_k, \mathbf{p}_i)/\tau)}$, where $\tau > 0$ is a temperature parameter and $\text{sim}(\cdot)$ is the cosine similarity function. The default value of $\tau$ is set as 1. Then the server conducts personalized weighted aggregation to produce weight $\overline{\mathbf{W}}_k$ for $k \in [K]$ with the formula $\overline{\mathbf{W}}_k = \sum_{k'=1}^{K} s_{k,k'} \mathbf{W}_{k'}$.

**Experimental settings**  We conducted experiments on three image classification datasets: CIFAR10, CIFAR100 (Krizhevsky et al., 2009), and Digits. These were chosen to address FL scenarios with label distribution imbalance and feature shift data heterogeneity issues. For FL with label distribution skew using CIFAR10 and CIFAR100, clients possess distinct object category distributions. We adopted similar "Pathological" and "Dirichlet" data partition strategies similar to (Marfoq et al., 2022; Li et al., 2023b) to make the class distribution among clients distinct. In the "Pathological" partition, we sampled eight/twenty classes for each client. And, in the "Dirichlet" partition, we employed a symmetric Dirichlet distribution with the default parameter $\alpha = 0.8$ to divide the dataset into sub-datasets. Conversely, for FL with feature skew using the Office dataset, each client has a roughly similar category distribution but originates from diverse domains. We introduced two baselines FedBN (Li et al., 2021) and FedPer (Arivazhagan et al., 2019), which separately personalize the Batch Normalization (BN) layers and the Classification Head for each local model. For a fair comparison, we also personalize all BN layers for our Weighed FedAvg. For a more in-depth exploration of this experiment setting, please refer to Appendix C.

**Results**  The experimental outcomes for FL with label distribution skew are provided in Table 1, while the results for FL with feature skew are showcased in Table 2. The results confirm that FedAvg can outperform local training in almost all datasets and Weighted FedAvg can further enhance the performance by increasing the communication efficiency, aligning with our theoretical predictions.

## 7 CONCLUSION

This work establishes a unified theoretical framework for the analysis of convergence and generalization in Federated Learning by leveraging feature learning theory. Based on a signal-noise decomposition for weights, we can track the dynamics of signal learning and noise memorization, providing convergence results. Furthermore, we demonstrate the generalization gap between FedAvg and local training through an effective signal-to-noise ratio. This result reveals how FedAvg benefits from communication. As a pioneering study in feature learning within the context of FL, our theoretical framework is limited to examining FedAvg with gradient descent optimization and a certain data generalization model. Future work can extend our framework to consider stochastic gradient descent and more advanced settings in FL.

ACKNOWLEDGMENTS

This work was supported by JSPS KAKENHI Grant Number 24K20848 and RIKEN Incentive Research Project 100847-202301062011. We thank the anonymous reviewers for useful suggestions to improve the paper.

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

## A    COMPLETE PROOF FOR COEFFICIENT ITERATION

In this section, we provide the proof for Lemma 4.1. This lemma plays a crucial role by transforming the perspective from tracking weights to learning feature coefficients. The main idea combines weight decomposition with gradient update rules. Before presenting the proof for Lemma 4.1, we introduce several preliminary results concerning the inner product between noise vectors and signal vectors, both inter- and intra-client:

**Signal vector versus signal vector**    Between different clients, namely $k$ and $k'$, according to our assumption, $\boldsymbol{\mu}_k$ is either orthogonal to $\boldsymbol{\mu}_{k'}$ or identical to $\boldsymbol{\mu}_{k'}$.

**Noise vector versus Noise vector**    It is known that all vectors are drawn from a Gaussian distribution, specifically, $\boldsymbol{\xi}_{k,i} \sim \mathcal{N}(\mathbf{0}, \sigma_p^2(\mathbf{I} - \sum_{c=1}^{C} \boldsymbol{\mu}^{(c)}\boldsymbol{\mu}^{(c)\top}/\|\boldsymbol{\mu}^{(c)}\|_2^2))$.

We employ the following lemma to demonstrate that noise vectors are almost orthogonal to each other:

**Lemma A.1** (Cao et al. (2022)). *Suppose that $\delta > 0$, and $d \geq \log(4n/\delta)$, then with probability at least $1 - \delta$, we have*

$$\frac{1}{2}\sigma_p^2 d \leq \|\boldsymbol{\xi}_{k,i}\|_2^2 \leq \frac{3}{2}\sigma_p^2 d,$$

$$|\langle \boldsymbol{\xi}_{k,i}, \boldsymbol{\xi}_{k',i'} \rangle| \leq \sigma_p^2 \sqrt{\log(4n^2/\delta)d}.$$

**Remark A.2.** *The total number of samples in this work is $n_k K = n$. Thus the union bound is taking over $n$.*

**Signal vector versus Noise vector**   We consider the inner product between the noise vector and the signal vector. According to our setting of the noise vector, namely, $\boldsymbol{\xi}k, i \sim \mathcal{N}(\mathbf{0}, \sigma_p^2(\mathbf{I} - \sum_{c=1}^{C} \boldsymbol{\mu}^{(c)} \boldsymbol{\mu}^{(c)\top}/\|\boldsymbol{\mu}^{(c)}\|_2^2))$, the noise vector is strictly orthogonal to the signal vectors.

The above statements collectively demonstrate that noise vectors and signal vectors are (nearly) orthogonal to each other. They not only imply that the coefficient decomposition is guaranteed to be unique with a high probability but are also rooted in theoretical analysis. Now, we are ready to demonstrate the Lemma of coefficient iteration.

**Lemma A.3** (Restatement of Lemma 4.1). *The coefficients $\gamma_{k,k',j,r}^{(t)}, \psi_{k,k',j,r,i}^{(t)}, \phi_{k,k',j,r,i}^{(t)}$ in decomposition (8), for all $k, k' \in [K]$, follow the update rules governed by:*

$$\gamma_{k,k',j,r}^{(0)}, \psi_{k,k',j,r,i}^{(0)}, \phi_{k,k',j,r,i}^{(0)} = 0, \tag{15}$$

$$\gamma_{k,k',j,r}^{(t+1)} = \gamma_{k,k',j,r}^{(t)} - \frac{\eta}{n_k m} \sum_{i=1}^{n_k} \ell_{k,i}'^{(t)} \sigma'(\langle \mathbf{w}_{k,j,r}^{(t)}, y_{k,i}\boldsymbol{\mu}_k \rangle) \|\boldsymbol{\mu}_k\|_2^2 \mathbb{1}(k' = k), \tag{16}$$

$$\psi_{k,k',j,r,i}^{(t+1)} = \psi_{k,k',j,r,i}^{(t)} - \frac{\eta}{n_k m} \ell_{k,i}'^{(t)} \sigma'(\langle \mathbf{w}_{k,j,r}^{(t)}, \boldsymbol{\xi}_{k,i} \rangle) \|\boldsymbol{\xi}_{k,i}\|_2^2 \mathbb{1}(y_{k,i} = j)\mathbb{1}(k' = k), \tag{17}$$

$$\phi_{k,k',j,r,i}^{(t+1)} = \phi_{k,k',j,r,i}^{(t)} + \frac{\eta}{n_k m} \ell_{k,i}'^{(t)} \sigma'(\langle \mathbf{w}_{k,j,r}^{(t)}, \boldsymbol{\xi}_{k,i} \rangle) \|\boldsymbol{\xi}_{k,i}\|_2^2 \mathbb{1}(y_{k,i} = -j)\mathbb{1}(k' = k). \tag{18}$$

*Furthermore, the coefficients $\overline{\gamma}_{k,j,r}^{(t)}, \overline{\psi}_{k,j,r,i}^{(t)}, \overline{\phi}_{k,j,r,i}^{(t)}$ in decomposition (9), for all $k \in [K]$, are updated according to the following rules:*

$$\overline{\gamma}_{k,j,r}^{(t)} = \sum_{k''=1}^{K} \sum_{k'=1}^{K} \frac{n_{k'}}{n} \gamma_{k',k'',j,r}^{(t)} \mathbb{1}(\boldsymbol{\mu}_{k''} = \boldsymbol{\mu}_k), \overline{\psi}_{k,j,r,i}^{(t)} = \sum_{k'=1}^{K} \frac{n_{k'}}{n} \psi_{k',k,j,r,i}^{(t)}, \overline{\phi}_{k,j,r,i}^{(t)} = \sum_{k'=1}^{K} \frac{n_{k'}}{n} \phi_{k',k,j,r,i}^{(t)}. \tag{19}$$

*Proof of Lemma A.3.* First, we have proven that the vectors are both linearly independent and orthogonal, within each client and across different clients, with high probability. Consequently, this ensures that the decomposition, as expressed in Equation (9), remains linear independent as the dimensionality increases

Then, we demonstrate the dynamics concerning the gradient update for each client

$$\mathbf{w}_{k,j,r}^{(t+1)} = \mathbf{w}_{k,j,r}^{(t)} - \eta \cdot \nabla_{\mathbf{w}_{k,j,r}} L_{\mathcal{S}_k}(\mathbf{W}_k^{(t)})$$

$$= \mathbf{w}_{k,j,r}^{(t)} - \frac{\eta}{n_k m} \sum_{i=1}^{n_k} \ell_{k,i}'^{(t)} \cdot \sigma'(\langle \mathbf{w}_{k,j,r}^{(t)}, \boldsymbol{\xi}_{k,i} \rangle) \cdot j y_{k,i} \boldsymbol{\xi}_{k,i}$$

$$- \frac{\eta}{n_k m} \sum_{i=1}^{n_k} \ell_{k,i}'^{(t)} \cdot \sigma'(\langle \mathbf{w}_{k,j,r}^{(t)}, y_{k,i} \cdot \boldsymbol{\mu}_k \rangle) \cdot j \boldsymbol{\mu}_k.$$

At the same time, the weight decomposition can be expressed as:

$$\mathbf{w}_{k,j,r}^{(t)} = \sum_{k'=1}^{K} (\alpha_{k,k',j,r}^{(t)} \mathbf{w}_{k',j,r}^{(0)} + \gamma_{k,k',j,r}^{(t)} \|\boldsymbol{\mu}_{k'}\|_2^{-2} \boldsymbol{\mu}_{k'} + \sum_{i=1}^{n_{k'}} \rho_{k,k',j,r,i}^{(t)} \|\boldsymbol{\xi}_{k',i}\|_2^{-2} \boldsymbol{\xi}_{k',i}).$$

By compared with the coefficients of $\mathbf{w}_{k,j,r}^{(0)}$, $\boldsymbol{\mu}_k$, and $\boldsymbol{\xi}_{k,i}$, on above two equations, for all $k, k' \in [K]$, we arrive the following results:

$$\gamma_{k,k',j,r}^{(t+1)} = \gamma_{k,k',j,r}^{(t)} - \frac{\eta}{n_k m} \sum_{i=1}^{n_k} \ell_{k,i}'^{(t)} \sigma'(\langle \mathbf{w}_{k,j,r}^{(t)}, y_{k,i} \boldsymbol{\mu}_k \rangle) \|\boldsymbol{\mu}_k\|_2^2 \mathbb{1}(k' = k),$$

$$\psi_{k,k',j,r,i}^{(t+1)} = \psi_{k,k',j,r,i}^{(t)} - \frac{\eta}{n_k m} \ell_{k,i}'^{(t)} \sigma'(\langle \mathbf{w}_{k,j,r}^{(t)}, \boldsymbol{\xi}_{k,i} \rangle) \|\boldsymbol{\xi}_{k,i}\|_2^2 \mathbb{1}(y_{k,i} = j) \mathbb{1}(k' = k),$$

$$\phi_{k,k',j,r,i}^{(t+1)} = \phi_{k,k',j,r,i}^{(t)} + \frac{\eta}{n_k m} \ell_{k,i}'^{(t)} \sigma'(\langle \mathbf{w}_{k,j,r}^{(t)}, \boldsymbol{\xi}_{k,i} \rangle) \|\boldsymbol{\xi}_{k,i}\|_2^2 \mathbb{1}(y_{k,i} = -j) \mathbb{1}(k' = k),$$

$$\alpha_{k,k',j,r}^{(t+1)} = \alpha_{k,k',j,r}^{(t)}.$$

Note that the gradient descent update (4) does not change the coefficients of $\mathbf{w}_{k,j,r}^{(0)}$. However, as we will see later, the weight averaging operation will mix the initializations of weights among different clients. Furthermore, observe that $\ell_{k,i}'^{(t)} < 0$ by the definition of the cross-entropy loss. Therefore, the signs of $\psi_{k,k,j,r,i}^{(t+1)}$ and $\phi_{k,k,j,r,i}^{(t+1)}$ can be determined by $j$ and $y_{k,i}$.

Lastly, we illustrate the dynamics of weight averaging among clients. Examining the weight average update (5) and weight decomposition (9), we find that:

$$\overline{\mathbf{w}}_{j,r}^{(t)} = \sum_{k=1}^{K} \frac{n_k}{n} \mathbf{w}_{k,j,r}^{(t)},$$

$$\overline{\mathbf{w}}_{j,r}^{(t)} = \sum_{k=1}^{K} \left( \overline{\alpha}_{k,j,r}^{(t)} \mathbf{w}_{k,j,r}^{(0)} + \overline{\gamma}_{k,j,r}^{(t)} \|\boldsymbol{\mu}_k\|_2^{-2} \boldsymbol{\mu}_k + \sum_{i=1}^{n_k} (\overline{\psi}_{k,j,r,i}^{(t)} + \overline{\phi}_{k,j,r,i}^{(t)}) \|\boldsymbol{\xi}_{k,i}\|_2^{-2} \boldsymbol{\xi}_{k,i} \right).$$

By comparing the coefficients of $\mathbf{w}_{k,j,r}^{(0)}$, $\boldsymbol{\mu}_k$, and $\boldsymbol{\xi}_{k,i}$ in the above two equations, we derive the following results:

$$\overline{\alpha}_{k,j,r}^{(t)} = \sum_{k=1}^{K} \frac{n_k}{n} \alpha_{k,j,r}^{(t)},$$

$$\overline{\gamma}_{k,j,r}^{(t)} = \sum_{k''=1}^{K} \sum_{k'=1}^{K} \frac{n_{k'}}{n} \gamma_{k',k'',j,r}^{(t)} \mathbb{1}(\boldsymbol{\mu}_{k''} = \boldsymbol{\mu}_k),$$

$$\overline{\psi}_{k,j,r,i}^{(t)} = \sum_{k'=1}^{K} \frac{n_{k'}}{n} \psi_{k',k,j,r,i}^{(t)},$$

$$\overline{\phi}_{k,j,r,i}^{(t)} = \sum_{k'=1}^{K} \frac{n_{k'}}{n} \phi_{k',k,j,r,i}^{(t)}.$$

$\square$

# B    COMPLETE PROOF FOR COEFFICIENT DYNAMICS

## B.1    STAGE ONE: INCREASE OF SIGNAL LEARNING AND NOISE MEMORIZATION

According to the behavior of the loss derivative, we divide the entire training dynamics into two stages. In the first stage, the loss derivative is close to a constant. Thus, we can simplify the analysis of coefficient iteration dynamics by replacing the loss derivative with the corresponding constant value. Before we conduct the proof for the main lemma in stage one, we list several useful preliminary lemmas for random initialization.

**Lemma B.1** (Cao et al. (2022))**.** *Suppose that $\delta > 0$, then with probability at least $1 - \delta$, we have*

$$|\langle \mathbf{w}_{k,j,r}^{(0)}, \boldsymbol{\mu}^{(c)} \rangle| \leq \sqrt{\log(4mKC/\delta)} \sigma_0 \|\boldsymbol{\mu}^{(c)}\|_2,$$

$$|\langle \mathbf{w}_{k,j,r}^{(0)}, \boldsymbol{\xi}_{k',i} \rangle| \leq \sqrt{\log(4mKn/\delta)d} \sigma_0 \sigma_p.$$

With Lemma B.1 at hand, we are ready to demonstrate how the effect of initialization changes as we control the variance of the weights' initialization.

**Lemma B.2.** *Denote that:* $\beta = 2\max_{k,i,j,r}\{|\langle\mathbf{w}_{k,j,r}^{(0)},\boldsymbol{\mu}_k\rangle|, |\langle\mathbf{w}_{k,j,r}^{(0)},\boldsymbol{\xi}_{k,i}\rangle|\}$. *Suppose that* $\sigma_0 \leq [4\sqrt{\log(4KmnC/\delta)}]^{-1}\min\left\{\|\boldsymbol{\mu}\|_2^{-1}, (\sigma_p\sqrt{d})^{-1}\right\}$, *then with probability at least* $1 - \delta$, *we have* $\beta \leq 1$.

*Proof.* By Lemma B.1, with probability at least $1 - \delta$, we can set an upper bound for $\beta$ as follows:

$$\beta \leq 4\sqrt{\log(4KmnC/\delta)} \cdot \sigma_0 \cdot \max\{\|\boldsymbol{\mu}\|_2, \sigma_p\sqrt{d}\}.$$

According to the exact condition for $\sigma_0$, which is also a part of Assumption 4.2:

$$\sigma_0 \leq [4\sqrt{\log(4KmnC/\delta)}]^{-1}\min\left\{\|\boldsymbol{\mu}\|_2^{-1}, (\sigma_p\sqrt{d})^{-1}\right\}, \tag{20}$$

we conclude to verify the following inequality:

$$\beta \leq 4\sqrt{\log(4KmnC/\delta)} \cdot \sigma_0 \cdot \max\{\|\boldsymbol{\mu}\|_2, \sigma_p\sqrt{d}\} \leq 1. \tag{21}$$

$\square$

Next we bound the loss derivative in the first stage.

**Lemma B.3.** *Suppose that* $\gamma_{k,k',j,r}^{(t)} = O(1)$ *and* $\psi_{k,k',j,r,i}^{(t)} = O(1)$, $\phi_{k,k',j,r,i}^{(t)} = O(1)$ *with* $k, k' \in [K]$, *and* $i \in [n]$, *for* $t \in [0, T_1]$, *then we that*

$$C_1 \leq -\ell_{k,i}'^{(t)} \leq 1, \tag{22}$$

*where* $C_1$ *is a positive constant.*

*Proof.* According to the assumption that for $0 \leq t \leq T_1$, we have $\max_{k,k',r}\gamma_{k,k',j,r}^{(t)} = O(1)$ and $\max_{k,k',j,r}\psi_{k,k',j,r,i}^{(t)} = O(1)$ and $\max_{k,k',j,r}|\phi_{k,k',j,r,i}^{(t)}| = O(1)$ for all $i \in [n]$, the output function satisfies:

$$F_j(\mathbf{W}_j^{(t)}, \mathbf{x}) = \frac{1}{m}\sum_{r=1}^{m}\left[\sigma(\mathbf{w}_{k,j,r}^{\top}\mathbf{x}^{(1)}) + \sigma(\mathbf{w}_{k,j,r}^{\top}\mathbf{x}^{(2)})\right]$$

$$\overset{(a)}{\leq} \max\left\{\langle\mathbf{w}_{k,j,r}^{(0)},\boldsymbol{\mu}\rangle, \gamma_{k,j,r}^{(t)}, \langle\mathbf{w}_{k,j,r}^{(0)},\boldsymbol{\xi}\rangle, |\psi_{k,j,r,i}^{(t)}|, |\phi_{k,j,r,i}^{(t)}|\right\}$$

$$\overset{(b)}{=} \max\left\{\beta, \gamma_{k,j,r}^{(t)}, |\psi_{j,r,i}^{(t)}|, |\phi_{j,r,i}^{(t)}|\right\}$$

$$\overset{(c)}{=} O(1),$$

where (a) is by the weight decomposition (6), (b) is by the definition of $\beta$, and (c) is by Lemma B.2.

Next, we bound the loss derivative for $t \in [0, T_1]$ as follows:

$$-\ell_i'^{(t)} = \frac{\exp(-f_i y_i)}{1 + \exp(-f_i y_i)} = O(1). \tag{23}$$

Finally, we show the lower bound for $-\ell_i'^{(t)}$:

$$-\ell_i'^{(t)} = \frac{\exp(-f_i y_i)}{1 + \exp(-f_i y_i)}$$

$$= \frac{1}{1 + \exp(f_i y_i)}$$

$$\geq \frac{1}{1 + O(1)}.$$

Similarly, we can obtain

$$C_1 \leq -\ell_{k,i}'^{(t)}(\overline{\mathbf{W}}) \leq 1,$$

by the relation $\overline{\mathbf{w}}_{j,r}^{(t)} = \sum_{k=1}^{K}\frac{1}{K}\mathbf{w}_{k,j,r}^{(t)}$ and $K = \Theta(1)$. $\square$

With the above results at hand, we are ready to present the outcomes of the first stage.

**Lemma B.4** (Restatement of Lemma 5.1). *Under assumption 4.2, there exists total number of local update $T_1 = R_1 E = \Theta(\eta^{-1} K n m \sigma_p^{-2} d^{-1})$ such that*

- $\max_r \overline{\gamma}_{k,j,r}^{(T_1)} = \Theta(\overline{n} \cdot \overline{\mathrm{SNR}}_k^2)$ *for $j \in \{\pm 1\}$, and $k \in [K]$.*

- $|\overline{\psi}_{k,j,r,i}^{(t)}| = \Theta(1)$ *for some $j \in \{\pm 1\}$, $r \in [m]$, $i \in [n]$ and all $k \in [K]$.*

- $\max_{j,r} \overline{\phi}_{k,j,r,i} = O(1)$ *for all $i \in [n]$ and $k \in [K]$.*

*Proof of Lemma B.4.* We first consider noise memorization. Define that for each $i \in [n]$ and $k \in [K]$, $\Psi_{k,k',i}^{(t)} = \max_{j,r} |\rho_{k,k',j,r,i}^{(t)}| = \max_{j,r} \{\psi_{k,k',j,r,i}^{(t)}, -\phi_{k,k',j,r,i}^{(t)}\}$. Similarly, we define the coefficient after weight average $\overline{\Psi}_{k,k',i}^{(t)} = \max_{j,r} |\overline{\rho}_{k,k',j,r,i}^{(t)}| = \max_{j,r} \{\overline{\psi}_{k,k',j,r,i}^{(t)}, -\overline{\phi}_{k,k',j,r,i}^{(t)}\}$.

Clearly we have $\Psi_{k,k',i}^{(0)} = 0$ for all $i \in [n]$ and $k, k' \in [K]$ by definition. Then by Equation (12) and Equation (13) we have:

$$
\begin{aligned}
\Psi_{k,k,i}^{(t+1)} &= \Psi_{k,k,i}^{(t)} + \frac{\eta}{n_k m} \cdot |\ell_{k,i}'^{(t)}| \cdot \sigma'\Bigg( \langle \mathbf{w}_{k,j,r}^{(0)}, \boldsymbol{\xi}_{k,i} \rangle + \sum_{k'=1}^{K} \sum_{i'=1}^{n} \Psi_{k,k',i'}^{(t)} \cdot \frac{|\langle \boldsymbol{\xi}_{k',i'}, \boldsymbol{\xi}_{k,i} \rangle|}{\|\boldsymbol{\xi}_{k',i'}\|_2^2} \\
&\qquad + \sum_{k'=1}^{K} \sum_{i'=1}^{n} \Psi_{k,k',i'}^{(t)} \cdot \frac{|\langle \boldsymbol{\xi}_{k',i'}, \boldsymbol{\xi}_{k,i} \rangle|}{\|\boldsymbol{\xi}_{k',i'}\|_2^2} \Bigg) \cdot \|\boldsymbol{\xi}_{k,i}\|_2^2 \\
&\overset{(a)}{\leq} \Psi_{k,k,i}^{(t)} + \frac{\eta}{n_k m} \cdot \sigma'\Bigg( \langle \mathbf{w}_{k,j,r}^{(0)}, \boldsymbol{\xi}_{k,i} \rangle + 2 \cdot \sum_{k'=1}^{K} \sum_{i'=1}^{n} \Psi_{k,k',i}^{(t)} \cdot \frac{|\langle \boldsymbol{\xi}_{k',i'}, \boldsymbol{\xi}_{k,i} \rangle|}{\|\boldsymbol{\xi}_{k',i'}\|_2^2} \Bigg) \cdot \|\boldsymbol{\xi}_{k,i}\|_2^2 \\
&\overset{(b)}{\leq} \Psi_{k,k,i}^{(t)} + \frac{\eta}{n_k m} \cdot \|\boldsymbol{\xi}_{k,i}\|_2^2 \\
&\overset{(c)}{\leq} \Psi_{k,k,i}^{(t)} + \frac{\eta}{n_k m} \cdot 3/2\sigma_p^2 d,
\end{aligned}
\tag{24}
$$

where the first inequality (a) follows from $|\ell_{k,i}'^{(t)}| \leq 1$, as stated in Equation (22) within Lemma B.3; the second inequality (b) is due to the non-linear activation $\sigma(x) = \max\{0, x\}$; and the last inequality (c) is derived from Lemma A.1.

Before the first step of averaging weights, by taking a telescoping sum over $t = 0, 1, \ldots, E$, we obtain:

$$
\Psi_{k,k,i}^{(E)} \leq \frac{\eta E}{n_k m} \frac{3}{2} \sigma_p^2 d.
$$

At the same time, by Equations (17) and (18), we obtain $\Psi_{k,k',i}^{(t)} = 0$ with $k \neq k'$ for $i \in [n]$ and $t \in [E]$. After $t = E$ steps, we perform weight average operation in server. Then by Equation (19) we have:

$$
\overline{\Psi}_{k,i}^{(E)} \leq \frac{1}{K} \frac{\eta E}{n_k m} \frac{3}{2} \sigma_p^2 d.
$$

Note that noise vectors $\boldsymbol{\xi}_{k,i}$ are almost independent of each other. Thus, during the first weight averaging, the effect is directly reduced by $K$ times. Besides, every client will store this information after accepting the averaged weight from the server.

In the next $E$ gradient descent steps at each client, by applying (24), we further obtain:

$$
\begin{aligned}
\Psi_{k,k,i}^{(2E)} &\leq \frac{\eta E}{K n_k m} \frac{3}{2} \sigma_p^2 d + \frac{\eta E}{n_k m} \frac{3}{2} \sigma_p^2 d = \frac{K+1}{K} \cdot \frac{\eta E}{n_k m} \frac{3}{2} \sigma_p^2 d, \\
\Psi_{k,k',i}^{(2E)} &\leq \frac{\eta E}{K n_k m} \frac{3}{2} \sigma_p^2 d, \quad k' \neq k.
\end{aligned}
$$

In the second round of local updates, there are two sources of noise. The first originates from the last return of weight averaging, and the second comes from the gradient descent update in the local client. From this result, we find that the growth of noise memorization is akin to only $E$ steps of gradient descent update when $K$ is a large number.

Again, after gradient descent, we apply second weight average operation in server and obtain the following result:

$$\overline{\Psi}_{k,i}^{(2E)} \leq \frac{K+1}{K^2} \cdot \frac{\eta E}{n_k m} \frac{3}{2} \sigma_p^2 d + \frac{K-1}{K^2} \cdot \frac{\eta E}{n_k m} \frac{3}{2} \sigma_p^2 d = \frac{2K}{K^2} \cdot \frac{\eta E}{n_k m} \frac{3}{2} \sigma_p^2 d.$$

During the second weight averaging operation, we calculate the coefficient of noise memorization by averaging all weights. The first term $\frac{K+1}{K^2} \cdot \frac{\eta b}{n_k m} \frac{3}{2} \sigma_p^2 d$ results from averaging the coefficient on client $k$. Moreover, the second term originates from the other $K-1$ clients who stored the coefficient during the first weight averaging operation. Together, applying Equation (19) yields

$$\overline{\Psi}_{k,i}^{(2E)} \leq \frac{2}{K} \cdot \frac{\eta E}{n_k m} \frac{3}{2} \sigma_p^2 d,$$

which is equivalent to being divided by $K$ compared to the local update without weight aggregation. Similarly, we repeat the computation procedure and obtain the following results for the third round of local updates plus the weight averaging operation:

$$\Psi_{k,k,i}^{(3E)} \leq \frac{2}{K} \cdot \frac{\eta E}{n_k m} \frac{3}{2} \sigma_p^2 d + \frac{\eta E}{n_k m} \frac{3}{2} \sigma_p^2 d = \frac{K+2}{K} \cdot \frac{\eta E}{n_k m} \frac{3}{2} \sigma_p^2 d,$$

$$\Psi_{k,k',i}^{(3E)} \leq \frac{2}{K} \cdot \frac{\eta E}{n_k m} \frac{3}{2} \sigma_p^2 d, \quad k' \neq k,$$

$$\overline{\Psi}_{k,i}^{(3E)} \leq \frac{K+2}{K^2} \cdot \frac{\eta E}{n_k m} \frac{3}{2} \sigma_p^2 d + \frac{2(K-1)}{K^2} \cdot \frac{\eta E}{n_k m} \frac{3}{2} \sigma_p^2 d = \frac{3}{K} \cdot \frac{\eta E}{n_k m} \frac{3}{2} \sigma_p^2 d.$$

It is not hard to observe that after the third round of local updates and weight averaging, the noise memorization is equivalent to the noise from local training divided by $K$. Using the same technique, we can summarize that, given $R_1 \geq 1$ times of communication, at the end of stage one with $R_1 = T_1/E$ round of communication, the noise memorization has an upper bound:

$$\Psi_{k,k,i}^{(R_1 E)} \leq \frac{K+(R_1-1)}{K} \cdot \frac{\eta E}{n_k m} \frac{3}{2} \sigma_p^2 d,$$

$$\Psi_{k,k',i}^{(R_1 E)} \leq \frac{R_1-1}{K} \cdot \frac{\eta E}{n_k m} \frac{3}{2} \sigma_p^2 d, \quad k' \neq k,$$

$$\overline{\Psi}_{k,i}^{(R_1 E)} \leq \frac{R_1}{K} \cdot \frac{\eta E}{n_k m} \frac{3}{2} \sigma_p^2 d.$$

Note that $\frac{K+R_1-1}{K} > \frac{R_1}{K}$ given that $K > 1$.

On the other hand, we establish the lower bound for $\overline{\psi}_{k,j,r,i}^{(t)}$ and $\overline{\phi}_{k,j,r,i}^{(t)}$. We show for $y_{k,i} = j$ with gradient descent update before weight average, we have:

$$\langle \mathbf{w}_{k,j,r}^{(t)}, \boldsymbol{\xi}_{k,i} \rangle = \langle \mathbf{w}_{k,j,r}^{(0)}, \boldsymbol{\xi}_{k,i} \rangle + \sum_{k'=1}^{K} \sum_{i'=1}^{n_{k'}} \psi_{k,k',j,r,i'}^{(t)} \|\boldsymbol{\xi}_{k',i'}\|_2^{-2} \langle \boldsymbol{\xi}_{k,i}, \boldsymbol{\xi}_{k',i'} \rangle$$

$$+ \sum_{k'=1}^{K} \sum_{i'=1}^{n_{k'}} \phi_{k,k',j,r,i'}^{(t)} \|\boldsymbol{\xi}_{k',i'}\|_2^{-2} \langle \boldsymbol{\xi}_{k,i}, \boldsymbol{\xi}_{k',i'} \rangle$$

$$\overset{(a)}{\geq} \langle \mathbf{w}_{k,j,r}^{(0)}, \boldsymbol{\xi}_{k,i} \rangle + \psi_{k,k,j,r,i}^{(t)} - 4\sqrt{\frac{\log(4n^2/\delta)}{d}} \sum_{k'} \sum_{i' \neq i} (|\psi_{k,k',j,r,i'}^{(t)}| + |\phi_{k,k',j,r,i'}^{(t)}|)$$

$$\overset{(b)}{\geq} \langle \mathbf{w}_{k,j,r}^{(0)}, \boldsymbol{\xi}_{k,i} \rangle + \psi_{k,k,j,r,i}^{(t)} - 4C_2 n \sqrt{\frac{\log(4n^2/\delta)}{d}},$$

where $C_2$ is a positive constant, (a) is by Lemma A.1 and (b) is by $\phi_{k,k',j,r,i}^{(t)} = 0$ when $y_{k,i} = j$ and $\psi_{k,k',j,r,i} \leq C_2$.

Let $\Psi_{k,k',i}^{(t)} = \max_{j=y_{k,i},r} \left\{ \langle \mathbf{w}_{k,j,r}^{(0)}, \boldsymbol{\xi}_{k',i} \rangle + \psi_{k,k',j,r,i}^{(t)} - 4C_2 n \sqrt{\frac{\log(4n^2/\delta)}{d}} \right\}$. At initialization, it is easy to check that:

$$\Psi_{k,k',i}^{(0)} \geq \sigma_0 \sigma_p \sqrt{d}/4 - 4C_2 n \sqrt{\frac{\log(4n^2/\delta)}{d}} \overset{(c)}{>} 0.$$

where (a) is by the following condition:

$$\sigma_0 \geq C_3 n \sqrt{\log(4n^2/\delta)}/(\sigma_p d). \tag{25}$$

Then, we can compute the growth of $\Psi_{k,k,i}^{(t)}$ as follows:

$$
\begin{aligned}
\Psi_{k,k,i}^{(t+1)} &= \Psi_{k,k,i}^{(t)} - \frac{\eta}{n_k m} \ell_{k,i}'^{(t)} \cdot \sigma'\left( \langle \mathbf{w}_{k,j,r}^{(0)}, \boldsymbol{\xi}_{k,i} \rangle + \sum_{k'=1}^{K} \sum_{i'=1}^{n_{k'}} \Psi_{k,k',i}^{(t)} \cdot \frac{|\langle \boldsymbol{\xi}_{k',i'}, \boldsymbol{\xi}_{k,i} \rangle|}{\|\boldsymbol{\xi}_{k',i'}\|_2^2} \right) \cdot \|\boldsymbol{\xi}_{k,i}\|_2^2 \\
&\overset{(a)}{\geq} \Psi_{k,k,i}^{(t)} + \frac{\eta C_1}{n_k m} \cdot \sigma'\left( \langle \mathbf{w}_{k,j,r}^{(0)}, \boldsymbol{\xi}_{k,i} \rangle + \sum_{k'=1}^{K} \sum_{i'=1}^{n_{k'}} \Psi_{k,k',i}^{(t)} \cdot \frac{|\langle \boldsymbol{\xi}_{k',i'}, \boldsymbol{\xi}_{k,i} \rangle|}{\|\boldsymbol{\xi}_{k',i'}\|_2^2} \right) \cdot \|\boldsymbol{\xi}_{k,i}\|_2^2 \\
&\overset{(b)}{\geq} \Psi_{k,k,i}^{(t)} + \frac{\eta C_1}{n_k m} \cdot 1/2 \sigma_p^2 d,
\end{aligned}
$$

where (a) is by Lemma B.3 stating the lower bound of minus loss derivative, (b) is by Lemma A.1 and Equation (25). Before the first step of weight average, we take a telescoping sum over $t = 0, 1, \ldots, E$, then gives

$$\Psi_{k,k,i}^{(E)} \geq \frac{\eta E}{n_k m} \frac{C_1}{2} \sigma_p^2 d.$$

At the same time, we claim $\Psi_{k,k',i}^{(t)} \geq 0$ with $k \neq k'$ for $i \in [n]$ and $t \in [E]$ through Equations (18) and (17). After $t = E$ steps, we perform weight average operation in server. Then by Equation (19) we have:

$$\overline{\Psi}_{k,i}^{(E)} \geq \frac{1}{K} \frac{\eta E}{n_k m} \frac{C_1}{2} \sigma_p^2 d.$$

The next $E$ gradient descent steps yields:

$$\Psi_{k,k,i}^{(2E)} \geq \frac{\eta E}{K n_k m} \frac{C_1}{2} \sigma_p^2 d + \frac{\eta E}{n_k m} \frac{C_1}{2} \sigma_p^2 d = \frac{K+1}{K} \frac{\eta E}{n_k m} \frac{C_1}{2} \sigma_p^2 d,$$

$$\Psi_{k,k',i}^{(2E)} \geq \frac{\eta E}{K n_k m} \frac{C_1}{2} \sigma_p^2 d, \quad k' \neq k.$$

In the second round of local update, there are two sources. The first source is from the previous weight average update, and the second is from the gradient descent update in the local client. From the result, we find the growth of noise memorization is close to only $E$ steps of gradient descent update when $K$ is a large number. Similarly, after gradient descent, we apply second weight average operation in server and obtain the following result:

$$\overline{\Psi}_{k,i}^{(2E)} \geq \frac{K+1}{K^2} \cdot \frac{\eta E}{n_k m} \frac{C_1}{2} \sigma_p^2 d + \frac{K-1}{K^2} \cdot \frac{\eta E}{n_k m} \frac{C_1}{2} \sigma_p^2 d = \frac{2}{K} \cdot \frac{\eta E}{n_k m} \frac{C_1}{2} \sigma_p^2 d.$$

It is not hard to observe that after the third round of local updates and weight averaging, the noise memorization is equivalent to local training divided by $K$. Using the same technique, we can summarize that, given $R_1$ times of communication, at the end of stage one with $R_1 = T_1/E \in \mathbb{Z}^+$ round of communication, the noise memorization has lower bounds:

$$\Psi_{k,k,i}^{(R_1 E)} \geq \frac{K + (R_1 - 1)}{K} \cdot \frac{\eta E}{n_k m} \frac{C_1}{2} \sigma_p^2 d,$$

$$\Psi_{k,k',i}^{(R_1 E)} \geq \frac{R_1 - 1}{K} \cdot \frac{\eta E}{n_k m} \frac{C_1}{2} \sigma_p^2 d, \quad k' \neq k,$$

$$\overline{\Psi}_{k,i}^{(R_1 E)} \geq \frac{R_1}{K} \cdot \frac{\eta E}{n_k m} \frac{C_1}{2} \sigma_p^2 d.$$

Finally, we confirm that

$$\max_{j=y_{k,i},r} \overline{\psi}_{k,j,r,i}^{(T_1)} \geq \frac{\eta T_1}{n_k m K} \frac{C_1}{2} \sigma_p^2 d - \langle \mathbf{w}_{k,j,r}^{(0)}, \boldsymbol{\xi}_{k,i} \rangle + 4C_2 n \sqrt{\frac{\log(4n^2/\delta)}{d}}$$

$$\overset{(a)}{\geq} \frac{\eta T_1}{n_k m K} \frac{C_1}{2} \sigma_p^2 d - \langle \mathbf{w}_{k,j,r}^{(0)}, \boldsymbol{\xi}_{k,i} \rangle + C_3$$

$$\overset{(b)}{\geq} C_4,$$

where the inequality (a) is by $4C_2 n \sqrt{\frac{\log(4n^2/\delta)}{d}} \leq C_3$ which holds when $d \geq C_5 \log(4n^2/\delta) n^2$ and the inequality (b) is by taking the value of $T_1$ and Lemma B.2.

A the same time, we calculate the growth of $\gamma_{k,k',j,r}^{(t)}$ and $\overline{\gamma}_{k,j,r}^{(t)}$. Using the iteration equation for the coefficient of signal learning under local gradient descent in the first round, we have:

$$\gamma_{k,k,j,r}^{(t+1)} = \gamma_{k,k,j,r}^{(t)} - \frac{\eta}{n_k m} \cdot \sum_{i=1}^{n_k} \ell_{k,i}'^{(t)} \sigma'(\langle \mathbf{w}_{k,j,r}^{(t)}, y_{k,i} \boldsymbol{\mu}_k \rangle) \|\boldsymbol{\mu}_k\|_2^2.$$

The next step is to provide an upper bound for signal learning at the first round of local updates.

$$\gamma_{k,k,j,r}^{(t+1)} \overset{(a)}{\leq} \gamma_{k,k,j,r}^{(t)} + \frac{\eta}{m} \cdot \|\boldsymbol{\mu}_k\|_2^2.$$

where (a) is by Lemma B.3. Before the first step of weight average, taking a telescoping sum over $t = 0, 1, \ldots, E$ then yields:

$$\gamma_{k,k,j,r}^{(E)} \leq \frac{\eta E}{m} \|\boldsymbol{\mu}_k\|_2^2.$$

After $t = E$ steps, we perform the weight averaging operation on the server through Equation (19). Then, we have:

$$\overline{\gamma}_{k,j,r}^{(E)} \leq \frac{1}{K} \sum_{k'=1}^{K} \frac{\langle \boldsymbol{\mu}_k, \boldsymbol{\mu}_{k'} \rangle}{\|\boldsymbol{\mu}_k\|_2^2} \frac{\eta E}{m} \|\boldsymbol{\mu}_k\|_2^2.$$

Note that $\sum_{k'=1}^{K} \frac{\langle \boldsymbol{\mu}_k, \boldsymbol{\mu}_{k'} \rangle}{\|\boldsymbol{\mu}_k\|_2^2}$ counts the total similarity between signal vectors among clients. In the case of i.i.d., this sum approaches the maximum value of 1. In the extreme non-i.i.d. case, where no client shares the same class, this sum equals $\frac{1}{K}$. For ease of derivation, we denote this sum as $\chi_k \triangleq \sum_{k'=1}^{K} \frac{\langle \boldsymbol{\mu}_k, \boldsymbol{\mu}_{k'} \rangle}{\|\boldsymbol{\mu}_k\|_2^2}$. In the subsequent $E$ gradient descent steps on each client, by applying Equation (24), we further have:

$$\gamma_{k,k,j,r}^{(2E)} \leq \frac{\eta E}{m} \sum_{k'=1}^{K} \frac{\langle \boldsymbol{\mu}_k, \boldsymbol{\mu}_{k'} \rangle}{\|\boldsymbol{\mu}_k\|_2^2} \|\boldsymbol{\mu}_k\|_2^2 + \frac{\eta E}{m} \|\boldsymbol{\mu}_k\|_2^2$$

$$= \left( \frac{1}{K} \sum_{k'=1}^{K} \frac{\langle \boldsymbol{\mu}_k, \boldsymbol{\mu}_{k'} \rangle}{\|\boldsymbol{\mu}_k\|_2^2} + 1 \right) \cdot \frac{\eta E}{m} \|\boldsymbol{\mu}_k\|_2^2$$

$$= (\frac{\chi_k}{K} + 1) \cdot \frac{\eta E}{m} \|\boldsymbol{\mu}_k\|_2^2.$$

In the second round of update, there are two sources. The first one arises from the last round of weight averaging, and the second comes from the gradient descent update in the local client. Again, after performing gradient descent, we apply a second weight averaging operation on the server and obtain the following result:

$$\overline{\gamma}_{k,j,r}^{(2E)} \leq \left( \frac{\chi_k}{K}(\frac{\chi_k}{K} + 1) + (1 - \frac{\chi_k}{K})\frac{\chi_k}{K} \right) \frac{\eta E}{m} \|\boldsymbol{\mu}_k\|_2^2 = 2\frac{\chi_k}{K} \frac{\eta E}{m} \|\boldsymbol{\mu}_k\|_2^2.$$

Here, the first term $\frac{\chi_k}{K}(\frac{\chi_k}{K} + 1) \cdot \frac{\eta E}{m} \|\boldsymbol{\mu}_k\|_2^2$ results from averaging the coefficient across $\chi_k$ clients who share the same signal vector. Moreover, the second term $(1 - \frac{\chi_k}{K})\frac{\chi_k}{K} \frac{\eta E}{m} \|\boldsymbol{\mu}_k\|_2^2$ originates from other $K - \chi_k$ clients who retained the coefficient during the first weight averaging operation.

Similarly, we repeat the computation procedure and obtain the following results for the third round of local update plus the weight averaging operation:

$$\gamma_{k,k,j,r}^{(3E)} \leq 2\frac{\chi_k}{K} \cdot \frac{\eta E}{m}\|\boldsymbol{\mu}_k\|_2^2 + \frac{\eta E}{m}\|\boldsymbol{\mu}_k\|_2^2 = (2\frac{\chi_k}{K}+1) \cdot \frac{\eta E}{m}\|\boldsymbol{\mu}_k\|_2^2,$$

$$\overline{\gamma}_{k,j,r}^{(3E)} \leq \frac{\chi_k}{K}(2\frac{\chi_k}{K}+1) \cdot \frac{\eta E}{m}\|\boldsymbol{\mu}_k\|_2^2 + (2\frac{\chi_k}{K})(1-\frac{\chi_k}{K}) \cdot \frac{\eta E}{m}\|\boldsymbol{\mu}_k\|_2^2 = 3\frac{\chi_k}{K} \cdot \frac{\eta E}{m}\|\boldsymbol{\mu}_k\|_2^2.$$

We repeat the computation process and find that, given $R_1$ times of communication, at the end of stage one, with $R_1 = T_1/E$ rounds of communication, the noise memorization has an upper bound:

$$\gamma_{k,j,r}^{(R_1E)} \leq ((R_1-1)\frac{\chi_k}{K}+1) \cdot \frac{\eta E}{m}\|\boldsymbol{\mu}_k\|_2^2,$$

$$\overline{\gamma}_{k,j,r}^{(R_1E)} \leq R_1\frac{\chi_k}{K} \cdot \frac{\eta E}{m}\|\boldsymbol{\mu}_k\|_2^2.$$

Lastly, we provide the lower bound for signal learning in the first stage. It is known that

$$\langle \mathbf{w}_{k,j,r}^{(t)}, \boldsymbol{\mu}_k \rangle = \langle \mathbf{w}_{k,j,r}^{(0)}, \boldsymbol{\mu}_k \rangle + \sum_{k'=1}^{K}\sum_{i=1}^{n_{k'}} \psi_{k,k',j,r,i}^{(t)}\|\boldsymbol{\xi}_{k',i}\|_2^{-2}\langle \boldsymbol{\xi}_{k',i}, \boldsymbol{\mu}_k \rangle$$

$$+ \sum_{k'=1}^{K}\sum_{i=1}^{n_{k'}} \phi_{k,k',j,r,i}^{(t)}\|\boldsymbol{\xi}_{k',i}\|_2^{-2}\langle \boldsymbol{\xi}_{k',i}, \boldsymbol{\mu}_k \rangle$$

$$\overset{(a)}{=} \langle \mathbf{w}_{k,j,r}^{(0)}, \boldsymbol{\mu}_k \rangle,$$

where (a) is by the orthogonal relation between feature vector and noise vector.

It is known that there exists some $r$ for which $\langle \mathbf{w}_{k,j,r}^{(0)}, y_i\boldsymbol{\mu}_k \rangle > 0$. Thus, for the first round of gradient update, we have:

$$\gamma_{k,k,j,r}^{(t+1)} = \gamma_{k,k,j,r}^{(t)} - \frac{\eta}{n_k m} \cdot \sum_{i=1}^{n_k} \ell_i^{\prime(t)} \cdot \sigma'(\langle \mathbf{w}_{k,j,r}^{(t)}, y_i\boldsymbol{\mu}_k \rangle) \cdot \|\boldsymbol{\mu}_k\|_2^2$$

$$\overset{(a)}{\geq} \gamma_{k,k,j,r}^{(t)} + \frac{\eta}{m}C_1 \cdot \|\boldsymbol{\mu}_k\|_2^2,$$

where (a) follows from the fact that $C_1 \leq -\ell_{k,i}' \leq 1$. Before the first step of weight averaging, taking a telescoping sum over $t = 0, 1, \ldots, E$ yields:

$$\gamma_{k,k,j,r}^{(E)} \geq \frac{\eta C_1 E}{m}\|\boldsymbol{\mu}_k\|_2^2.$$

After $t = E$ steps, we perform the weight averaging operation on the server. Then, we have:

$$\overline{\gamma}_{k,k,j,r}^{(E)} \geq \frac{1}{K}\sum_{k'=1}^{K}\frac{\langle \boldsymbol{\mu}_k, \boldsymbol{\mu}_{k'} \rangle}{\|\boldsymbol{\mu}_k\|_2^2}\frac{\eta C_1 E}{m}\|\boldsymbol{\mu}_k\|_2^2.$$

Recall that $\chi_k \triangleq \sum_{k'=1}^{K}\frac{\langle \boldsymbol{\mu}_k, \boldsymbol{\mu}_{k'} \rangle}{\|\boldsymbol{\mu}_k\|_2^2}$ counts the total similarity between the signal vectors among clients. In the next $E$ gradient descent steps on each client, by applying Equation (24) we further have:

$$\gamma_{k,k,j,r}^{(2E)} \geq \frac{\eta C_1 E}{mK}\sum_{k'=1}^{K}\frac{\langle \boldsymbol{\mu}_k, \boldsymbol{\mu}_{k'} \rangle}{\|\boldsymbol{\mu}_k\|_2^2}\|\boldsymbol{\mu}_k\|_2^2 + \frac{\eta C_1 E}{m}\|\boldsymbol{\mu}_k\|_2^2$$

$$= (\frac{\chi_k}{K}+1) \cdot \frac{\eta C_1 E}{m}\|\boldsymbol{\mu}_k\|_2^2.$$

Again, after gradient descent, we apply the second weight averaging operation on the server and obtain the following result:

$$\overline{\gamma}_{k,j,r}^{(2E)} \geq \left(\frac{\chi_k}{K}(\frac{\chi_k}{K}+1) + (1-\frac{\chi_k}{K})\frac{\chi_k}{K}\right)\frac{\eta C_1 E}{m}\|\boldsymbol{\mu}_k\|_2^2$$

$$= 2\frac{\chi_k}{K}\frac{\eta C_1 E}{m}\|\boldsymbol{\mu}_k\|_2^2.$$

During the second weight averaging operation, we calculate the coefficient of noise memorization by averaging all weights. Similarly, we repeat the computation procedure and obtain the following results for the $R_1$-th round of local update plus weight average operation:

$$\gamma_{k,k,j,r}^{(R_1 E)} \geq \left((R_1 - 1)\frac{\chi_k}{K} + 1\right) \cdot \frac{\eta C_1 E}{m}\|\boldsymbol{\mu}_k\|_2^2,$$

$$\overline{\gamma}_{k,j,r}^{(R_1 E)} \geq R_1 \frac{\chi_k}{K} \cdot \frac{\eta C_1 E}{m}\|\boldsymbol{\mu}_k\|_2^2.$$

Finally, taking the value of $T_1$ into the above inequality, we have:

$$\overline{\gamma}_{k,j,r}^{(T_1)} = \frac{C_1 \eta \chi_k}{m} \cdot \|\boldsymbol{\mu}_k\|_2^2 \cdot \frac{Cm\overline{n}}{\eta \sigma_q^2 d} = \Theta(\overline{n} \cdot \overline{\text{SNR}}_k^2).$$

$\square$

### B.2 STAGE TWO: CONVERGENCE ANALYSIS AND FEATURE LEARNING SCALE

In this section, we demonstrate how the learning scale of signal learning and noise memorization at convergence completes the characterization of feature learning initiated in the first stage.

#### B.2.1 FEATURE LEARNING SCALE AT CONVERGENCE

In the second stage, we aim to prove the following scaling behavior of feature learning

$$0 \leq \overline{\psi}_{k,j,r,i}^{(t)} \leq \psi_{k,k,j,r,i}^{(t)} \leq \log(T^*), \tag{26}$$

$$0 \geq \phi_{k,j,r,i}^{(t)} \geq \overline{\phi}_{k,k,j,r,i}^{(t)} \geq -\log(T^*), \tag{27}$$

where $t = RE$ for an integer $R$ and $T^* > T_1$.

The proof is by induction. Before proceeding with the main lemma, it is necessary to introduce several technical lemmas that are fundamental to our argument.

**Lemma B.5.** *Under Condition 4.2, suppose Equation (26) and Equation (27) hold at iteration $t$. Then*

$$\overline{\psi}_{k,j,r,i}^{(t)} - 8n\sqrt{\frac{\log(4n^2/\delta)}{d}}\log(T^*) \leq \langle \overline{\mathbf{w}}_{j,r}^{(t)} - \overline{\mathbf{w}}_{j,r}^{(0)}, \boldsymbol{\xi}_{k,i}\rangle \leq \overline{\psi}_{k,j,r,i}^{(t)} + 8n\sqrt{\frac{\log(4n^2/\delta)}{d}}\log(T^*),$$

$$\overline{\phi}_{k,j,r,i}^{(t)} - 8n\sqrt{\frac{\log(4n^2/\delta)}{d}}\log(T^*) \leq \langle \overline{\mathbf{w}}_{j,r}^{(t)} - \overline{\mathbf{w}}_{j,r}^{(0)}, \boldsymbol{\xi}_{k,i}\rangle \leq \overline{\phi}_{k,j,r,i}^{(t)} + 8n\sqrt{\frac{\log(4n^2/\delta)}{d}}\log(T^*),$$

*for all $r \in [m]$, $j \in \{\pm 1\}$, $i \in [n]$ and $k \in [K]$. Here we define $\overline{\mathbf{w}}_{j,r}^{(0)} = \sum_{k=1}^{K} \frac{n_k}{n}\mathbf{w}_{k,j,r}^{(0)}$.*

*Proof of Lemma B.5.* It is known that, when $y_{k,i} = j$:

$$\langle \overline{\mathbf{w}}_{j,r}^{(t)} - \overline{\mathbf{w}}_{j,r}^{(0)}, \boldsymbol{\xi}_{k,i}\rangle = \sum_{k'=1}^{K}\sum_{i'=1}^{n_{k'}} \overline{\psi}_{k',j,r,i'}^{(t)}\|\boldsymbol{\xi}_{k',i'}\|_2^{-2} \cdot \langle \boldsymbol{\xi}_{k',i'}, \boldsymbol{\xi}_{k,i}\rangle$$

$$+ \sum_{k'=1}^{K}\sum_{i'=1}^{n_{k'}} \overline{\phi}_{k',j,r,i'}^{(t)}\|\boldsymbol{\xi}_{k',i'}\|_2^{-2} \cdot \langle \boldsymbol{\xi}_{k',i'}, \boldsymbol{\xi}_{k,i}\rangle$$

$$\overset{(a)}{\leq} 4\sqrt{\frac{\log(4n^2/\delta)}{d}}\sum_{k'=1}^{K}\sum_{i'\neq i}|\overline{\psi}_{k',j,r,i'}^{(t)}| + 4\sqrt{\frac{\log(4n^2/\delta)}{d}}\sum_{k'=1}^{K}\sum_{i'\neq i}|\overline{\phi}_{k',j,r,i'}^{(t)}| + \overline{\psi}_{k,j,r,i}^{(t)}$$

$$\overset{(b)}{\leq} \overline{\psi}_{k,j,r,i}^{(t)} + 8n\sqrt{\frac{\log(4n^2/\delta)}{d}}\log(T^*),$$

where the first inequality (a) is by Lemma A.1 and the last inequality is by $|\overline{\psi}_{k,j,r,i'}^{(t)}|, |\overline{\phi}_{k,j,r,i'}^{(t)}| \leq \log(T^*)$ in Equation (26). Similarly, we can show that when $y_i \neq j$:

$$\langle \overline{\mathbf{w}}_{j,r}^{(t)} - \overline{\mathbf{w}}_{j,r}^{(0)}, \boldsymbol{\xi}_{k,i} \rangle = \sum_{k'=1}^{K} \sum_{i'=1}^{n_{k'}} \overline{\psi}_{k',j,r,i'}^{(t)} \|\boldsymbol{\xi}_{k',i'}\|_2^{-2} \cdot \langle \boldsymbol{\xi}_{k',i'}, \boldsymbol{\xi}_{k,i} \rangle$$

$$+ \sum_{k'=1}^{K} \sum_{i'=1}^{n_{k'}} \overline{\phi}_{k',j,r,i'}^{(t)} \|\boldsymbol{\xi}_{k',i'}\|_2^{-2} \cdot \langle \boldsymbol{\xi}_{k',i'}, \boldsymbol{\xi}_{k,i} \rangle$$

$$\overset{(a)}{\leq} 4\sqrt{\frac{\log(4n^2/\delta)}{d}} \sum_{k'=1}^{K} \sum_{i' \neq i} |\overline{\psi}_{k',j,r,i'}^{(t)}| + 4\sqrt{\frac{\log(4n^2/\delta)}{d}} \sum_{k'=1}^{K} \sum_{i' \neq i} |\overline{\phi}_{k',j,r,i'}^{(t)}| + \overline{\phi}_{k,j,r,i'}^{(t)}$$

$$\overset{(b)}{\leq} \overline{\phi}_{k,j,r,i}^{(t)} + 8n\sqrt{\frac{\log(4n^2/\delta)}{d}} \log(T^*),$$

where the first inequality is by Lemma A.1 and the second inequality is by $|\overline{\psi}_{k,j,r,i'}^{(t)}|, |\overline{\phi}_{k,j,r,i'}^{(t)}| \leq \log(T^*)$ in equation 26, which completes the proof. □

**Lemma B.6.** *Under Condition 4.2, suppose Equation (26) and Equation (27) hold at iteration t. Then we have:*

$$\psi_{k,k,j,r,i}^{(t)} - 8n\sqrt{\frac{\log(4n^2/\delta)}{d}} \log(T^*) \leq \langle \mathbf{w}_{k,j,r}^{(t)} - \overline{\mathbf{w}}_{j,r}^{(0)}, \boldsymbol{\xi}_{k,i} \rangle \leq \psi_{k,k,j,r,i}^{(t)} + 8n\sqrt{\frac{\log(4n^2/\delta)}{d}} \log(T^*),$$

$$\phi_{k,k,j,r,i}^{(t)} - 8n\sqrt{\frac{\log(4n^2/\delta)}{d}} \log(T^*) \leq \langle \mathbf{w}_{k,j,r}^{(t)} - \overline{\mathbf{w}}_{j,r}^{(0)}, \boldsymbol{\xi}_{k,i} \rangle \leq \phi_{k,k,j,r,i}^{(t)} + 8n\sqrt{\frac{\log(4n^2/\delta)}{d}} \log(T^*),$$

*for all $r \in [m]$, $j \in \{\pm 1\}$, $i \in [n]$ and $k \in [K]$. Here we define $\overline{\mathbf{w}}_{j,r}^{(0)} = \sum_{k=1}^{K} \frac{1}{K} \mathbf{w}_{k,j,r}^{(0)}$.*

*Proof of Lemma B.6.* It is known that, when $y_{k,i} = j$:

$$\langle \mathbf{w}_{k,j,r}^{(t)} - \overline{\mathbf{w}}_{j,r}^{(0)}, \boldsymbol{\xi}_{k,i} \rangle = \sum_{k'=1}^{K} \sum_{i'=1}^{n_{k'}} \psi_{k',k,j,r,i'}^{(t)} \|\boldsymbol{\xi}_{k',i'}\|_2^{-2} \cdot \langle \boldsymbol{\xi}_{k',i'}, \boldsymbol{\xi}_{k,i} \rangle$$

$$+ \sum_{k'=1}^{K} \sum_{i'=1}^{n_{k'}} \phi_{k',k,j,r,i'}^{(t)} \|\boldsymbol{\xi}_{k',i'}\|_2^{-2} \cdot \langle \boldsymbol{\xi}_{k',i'}, \boldsymbol{\xi}_{k,i} \rangle$$

$$\overset{(a)}{\leq} 4\sqrt{\frac{\log(4n^2/\delta)}{d}} \sum_{k'=1}^{K} \sum_{i' \neq i} |\psi_{k,k',j,r,i'}^{(t)}| + 4\sqrt{\frac{\log(4n^2/\delta)}{d}} \sum_{k'=1}^{K} \sum_{i' \neq i} |\phi_{k,k',j,r,i'}^{(t)}| + \psi_{k,k,j,r,i}^{(t)}$$

$$\overset{(b)}{\leq} \psi_{k,k,j,r,i}^{(t)} + 8n\sqrt{\frac{\log(4n^2/\delta)}{d}} \log(T^*),$$

where the first inequality (a) is by Lemma A.1 and the last inequality is by $|\psi_{k,k',j,r,i'}^{(t)}|, |\phi_{k,k',j,r,i'}^{(t)}| \leq \log(T^*)$ in Equation (26). Similarly, we can show that when $y_i \neq j$:

$$\langle \mathbf{w}_{k,j,r}^{(t)} - \overline{\mathbf{w}}_{j,r}^{(0)}, \boldsymbol{\xi}_{k,i} \rangle = \sum_{k'=1}^{K} \sum_{i'=1}^{n_{k'}} \psi_{k,k',j,r,i'}^{(t)} \|\boldsymbol{\xi}_{k',i'}\|_2^{-2} \cdot \langle \boldsymbol{\xi}_{k',i'}, \boldsymbol{\xi}_{k,i} \rangle$$

$$+ \sum_{k'=1}^{K} \sum_{i'=1}^{n_{k'}} \phi_{k,k',j,r,i'}^{(t)} \|\boldsymbol{\xi}_{k',i'}\|_2^{-2} \cdot \langle \boldsymbol{\xi}_{k',i'}, \boldsymbol{\xi}_{k,i} \rangle$$

$$\overset{(a)}{\leq} 4\sqrt{\frac{\log(4n^2/\delta)}{d}} \sum_{i' \neq i} |\psi_{k,j,r,i'}^{(t)}| + 4\sqrt{\frac{\log(4n^2/\delta)}{d}} \sum_{k'=1}^{K} \sum_{i' \neq i} |\phi_{k,k',j,r,i'}^{(t)}| + \phi_{k,k,j,r,i}^{(t)}$$

$$\overset{(b)}{\leq} \phi_{k,k,j,r,i}^{(t)} + 8n\sqrt{\frac{\log(4n^2/\delta)}{d}} \log(T^*),$$

where the first inequality is by Lemma A.1 and the second inequality is by $|\psi_{k,k',j,r,i'}^{(t)}|, |\phi_{k,k',j,r,i'}^{(t)}| \leq \log(T^*)$ in equation 26, which completes the proof. □

**Lemma B.7.** *Under Condition 4.2. For any $t > E$, it holds that*

$$\langle \overline{\mathbf{w}}_{j,r}^{(t)} - \overline{\mathbf{w}}_{j,r}^{(0)}, \boldsymbol{\mu}_k \rangle = j \cdot \overline{\gamma}_{k,j,r}^{(t)}$$

$$\langle \mathbf{w}_{k,j,r}^{(t)} - \overline{\mathbf{w}}_{j,r}^{(0)}, \boldsymbol{\mu}_k \rangle = j \cdot \gamma_{k,k,j,r}^{(t)}$$

*for all $r \in [m]$, $j \in \{\pm 1\}$, and $k \in [K]$.*

*Proof of Lemma B.7.* For any time $t > E$, we have that

$$\langle \mathbf{w}_{k,j,r}^{(t)} - \overline{\mathbf{w}}_{j,r}^{(0)}, \boldsymbol{\mu}_k \rangle = j \cdot \gamma_{k,k,j,r}^{(t)} + \sum_{k'=1}^{K} \sum_{i'=1} \psi_{k,k',j,r,i'}^{(t)} \|\boldsymbol{\xi}_{k',i'}\|_2^{-2} \cdot \langle \boldsymbol{\xi}_{k',i'}, \boldsymbol{\mu} \rangle$$

$$+ \sum_{k'=1}^{K} \sum_{i'=1} \phi_{k,k',j,r,i'}^{(t)} \|\boldsymbol{\xi}_{k',i'}\|_2^{-2} \cdot \langle \boldsymbol{\xi}_{k',i'}, \boldsymbol{\mu} \rangle$$

$$= j \cdot \gamma_{k,k,j,r}^{(t)},$$

where the equation is by our orthogonal assumption between feature vector and noise vector.
Similarly,

$$\langle \overline{\mathbf{w}}_{k,j,r}^{(t)} - \overline{\mathbf{w}}_{j,r}^{(0)}, \boldsymbol{\mu}_k \rangle = j \cdot \overline{\gamma}_{k,j,r}^{(t)} + \sum_{k'=1}^{K} \sum_{i'=1} \overline{\psi}_{k,j,r,i'}^{(t)} \|\boldsymbol{\xi}_{k',i'}\|_2^{-2} \cdot \langle \boldsymbol{\xi}_{k',i'}, \boldsymbol{\mu} \rangle$$

$$+ \sum_{k'=1}^{K} \sum_{i'=1} \overline{\phi}_{k',j,r,i'}^{(t)} \|\boldsymbol{\xi}_{k',i'}\|_2^{-2} \cdot \langle \boldsymbol{\xi}_{k',i'}, \boldsymbol{\mu} \rangle$$

$$= j \cdot \overline{\gamma}_{k,j,r}^{(t)}.$$

$\square$

**Lemma B.8.** *Under Condition 4.2, for $0 \le t \le T^*$, where $T^* = \eta^{-1}\text{poly}(\epsilon^{-1}, \|\boldsymbol{\mu}\|_2^{-1}, d^{-1}\sigma_p^{-2}, \sigma_0^{-1}, n, m, d) = R^*E$, we have that*

$$0 \le \overline{\psi}_{k,j,r,i}^{(t)} \le \psi_{k,k,j,r,i}^{(t)} \le \log(T^*)$$

$$0 \ge \phi_{k,j,r,i}^{(t)} \ge \overline{\phi}_{k,j,r,i}^{(t)} \ge -\log(T^*),$$

$$0 \le \overline{\gamma}_{k,j,r}^{(t)} \le \gamma_{k,k,j,r}^{(t)} \le \overline{n}\chi_k\text{SNR}_k^2\log(T^*),$$

*for all $r \in [m]$, $j \in \{\pm 1\}$, $k \in [K]$ and $i \in [n]$.*

*Proof of Lemma B.8.* The proof relies on induction. At $t = 0$, the results are straightforward, given that all coefficients are zero. We assume that there is a time $R_aE \le T^*$ for which the Lemma B.8 is valid for all moments $0 \le t \le (R_a - 1)E$. Our goal is to demonstrate that the result also stands true for $t = R_aE$.

We first prove that Equation (27) holds for $t = R_aE$, i.e., $\overline{\phi}_{k,j,r,i}^{(t)} \ge -\beta - 16n\sqrt{\frac{\log(4n^2/\delta)}{d}}\log(T^*)$ for $t = R_aE$, $r \in [m]$, $j \in \{\pm 1\}$, $k \in [K]$, and $i \in [n]$. Notice that $\overline{\phi}_{k,j,r,i}^{(t)} = 0, \forall j = y_{k,i}$. Therefore, we only need to consider the case that $j \ne y_{k,i}$.

When $\overline{\phi}_{k,j,r,i}^{((R_a-1)E)} \le -0.5\beta - 8n\sqrt{\frac{\log(4n^2/\delta)}{d}}\log(T^*)$, by Lemma B.5 we have that

$$\langle \overline{\mathbf{w}}_{j,r}^{((R_a-1)E)}, \boldsymbol{\xi}_{k,i} \rangle \le \overline{\phi}_{k,j,r,i}^{((R_a-1)E)} + \langle \overline{\mathbf{w}}_{j,r}^{(0)}, \boldsymbol{\xi}_{k,i} \rangle + 8n\sqrt{\frac{\log(4n^2/\delta)}{d}}\log(T^*) \le 0,$$

Then we have $E$ steps of local update, and we find that, for $\tau \in [E]$:

$$\langle \mathbf{w}_{k,j,r}^{((R_a-1)E)+\tau}, \boldsymbol{\xi}_{k,i} \rangle \le \langle \overline{\mathbf{w}}_{j,r}^{((R_a-1)E)}, \boldsymbol{\xi}_{k,i} \rangle.$$

Therefore, we have,

$$\overline{\phi}_{k,j,r,i}^{(R_aE)} = \overline{\phi}_{k,j,r,i}^{((R_a-1)E)} + \frac{1}{K}\sum_{\tau=1}^{E}\frac{\eta}{n_k m}\ell_i^{\prime((R_a-1)E)}\cdot\sigma'(\langle\mathbf{w}_{k,j,r}^{((R_a-1)E+\tau)},\boldsymbol{\xi}_{k,i}\rangle)\cdot\mathbb{1}(y_{k,i}=-j)\|\boldsymbol{\xi}_{k,i}\|_2^2$$

$$= \overline{\phi}_{k,j,r,i}^{((R_a-1)E)} \overset{(a)}{\geq} -\beta - 16n\sqrt{\frac{\log(4n^2/\delta)}{d}}\log(T^*),$$

where the last inequality (a) is by induction hypothesis.

When $\overline{\phi}_{k,j,r,i}^{((R_a-1)E)} \geq -0.5\beta - 8n\sqrt{\frac{\log(4n^2/\delta)}{d}}\log(T^*)$, we have that

$$\overline{\phi}_{k,j,r,i}^{(R_aE)} = \overline{\phi}_{k,j,r,i}^{((R_a-1)E)} + \frac{\eta}{n_k m K}\cdot\sum_{\tau=1}^{E}\ell_i^{\prime((R_a-1)E+\tau)}\cdot\sigma'(\langle\mathbf{w}_{k,j,r}^{((R_a-1)E+\tau)},\boldsymbol{\xi}_{k,i}\rangle)\cdot\mathbb{1}(y_i=-j)\|\boldsymbol{\xi}_{k,i}\|_2^2$$

$$\overset{(a)}{\geq} -0.5\beta - 8n\sqrt{\frac{\log(4n^2/\delta)}{d}}\log(T^*) - O\left(\frac{\eta E\sigma_p^2 d}{n_k m K}\right)\sigma'\left(0.5\beta + 8n\sqrt{\frac{\log(4n^2/\delta)}{d}}\log(T^*)\right)$$

$$\overset{(b)}{\geq} -0.5\beta - 8n\sqrt{\frac{\log(4n^2/\delta)}{d}}\log(T^*) - O\left(\frac{\eta E\sigma_p^2 d}{n_k m K}\right)$$

$$\overset{(c)}{\geq} -\beta - 16n\sqrt{\frac{\log(4n^2/\delta)}{d}}\log(T^*),$$

where we use $\ell_{k,i}^{\prime(t)} \leq 1$, and $\|\boldsymbol{\xi}_{k,i}\|_2 = O(\sigma_p^2 d)$ by Lemma A.1, and Lemma B.6 in the first inequality (a), the second inequality (b) is by $0.5\beta + 8n\sqrt{\frac{\log(4n^2/\delta)}{d}}\log(T^*) \leq 1$, and the last inequality (c) is by $\eta = O\big(n_k m K/(\sigma_p^2 dE)\big)$ in Assumption 4.2.

Next we prove Equation (26) holds for $t = R_aE$. Direct computation leads to $\overline{\psi}_{k,j,r,i}^{(t)}$,

$$\overline{\psi}_{k,j,r,i}^{(RE)} = \overline{\psi}_{k,j,r,i}^{(E(R-1))} - \frac{\eta}{n_k m K}\sum_{\tau=1}^{E}\ell_{k,i}^{\prime(ER-E+\tau)}\cdot\sigma'(\langle\mathbf{w}_{k,j,r}^{(RE-E+\tau)},\boldsymbol{\xi}_{k,i}\rangle)\cdot\mathbb{1}(y_{k,i}=j)\|\boldsymbol{\xi}_{k,i}\|_2^2.$$

Let $t_b = R_bE$ to be the last time $t < T^*$ that $\overline{\psi}_{k,j,r,i}^{(t_b)} \leq 0.5\log(T^*)$. Then we have that

$$\overline{\psi}_{k,j,r,i}^{(R_aE)} = \overline{\psi}_{k,j,r,i}^{(t_b)} - \frac{\eta}{n_k m K}\cdot\sum_{\tau=1}^{E}\ell_i^{\prime(t_b+\tau)}\cdot\sigma'(\langle\mathbf{w}_{k,j,r}^{(t_b+\tau)},\boldsymbol{\xi}_{k,i}\rangle)\cdot\mathbb{1}(y_{k,i}=j)\|\boldsymbol{\xi}_{k,i}\|_2^2$$

$$-\sum_{\tau=1}^{E}\sum_{R_b\leq R<R_a}\frac{\eta}{n_k m K}\cdot\ell_i^{\prime(RE+\tau)}\cdot\sigma'(\langle\mathbf{w}_{k,j,r}^{(RE+\tau)},\boldsymbol{\xi}_{k,i}\rangle)\cdot\mathbb{1}(y_{k,i}=j)\|\boldsymbol{\xi}_{k,i}\|_2^2$$

$$\overset{(a)}{\leq} 2\frac{\eta E\sigma_p^2 d}{nmK} + \sum_{R_b\leq R<T_a}\frac{\eta E}{n_k m K}\cdot\exp(-\sigma(\langle\mathbf{w}_{k,j,r}^{(t)},\boldsymbol{\xi}_{k,i}\rangle)+1)\cdot\sigma'(\langle\mathbf{w}_{k,j,r}^{(t)},\boldsymbol{\xi}_{k,i}\rangle)\cdot\|\boldsymbol{\xi}_{k,i}\|_2^2$$

$$\overset{(c)}{\leq} 0.25\log(T^*) + 0.25T^*\exp(-\log(T^*))\log(T^*)$$

$$\leq 0.5\log(T^*),$$

where the first inequality (a) is by Lemmas B.6 and A.1, the second inequality (b) is by $\eta \leq n_k m K/(E\sigma_p^2 d)\log(T^*)$, $\overline{\psi}_{k,j,r,i}^{(t)} > 0.5\log(T^*)$ and $\langle\mathbf{w}_{k,j,r}^{(0)},\boldsymbol{\xi}_{k,i}\rangle \geq -0.5\beta$ due to the definition of $t_b$ and $\beta$, (c) is by $\beta \leq 0.1\log(T^*)$ and $8n\sqrt{\frac{\log(4n^2/\delta)}{d}}\log(T^*) \leq 0.1\log(T^*)$.

Finally, we can prove that $0 \leq \overline{\gamma}_{k,j,r}^{(t)} \leq \gamma_{k,k,j,r}^{(t)} \leq \overline{n}\chi_k\mathrm{SNR}_k^2\log(T^*)$. Direct computation yields:

$$\overline{\gamma}_{k,j,r}^{(t+1)} = \overline{\gamma}_{k,j,r}^{(t)} - \sum_{\tau=1}^{E}\frac{\eta\chi_k}{n_k m K}\cdot\sum_{i=1}^{n_k}\ell_{k,i}^{\prime(t+\tau)}\cdot\sigma'(\langle\mathbf{w}_{k,j,r}^{(t+\tau)},y_{k,i}\cdot\boldsymbol{\mu}_k\rangle)\|\boldsymbol{\mu}_k\|_2^2.$$

Let $T_c = R_c E$ to be the last time $t < T^*$ that $\overline{\gamma}_{k,j,r}^{(t)} \leq 0.5 \log(T^*) \cdot \overline{n}\overline{\text{SNR}}_k^2$. Then we have that

$$
\begin{aligned}
\overline{\gamma}_{k,j,r}^{(T_a)} &= \overline{\gamma}_{k,j,r}^{(T_c)} - \frac{\eta\chi_k}{n_k m K} \sum_{\tau=1}^{E} \sum_{i=1}^{n_k} \ell_{k,i}'^{(T_c+\tau)} \cdot \sigma'(\langle \mathbf{w}_{k,j,r}^{(T_c+\tau)}, y_{k,i}\boldsymbol{\mu}_k \rangle)\|\boldsymbol{\mu}_k\|_2^2 \\
&\quad - \sum_{R_c < R < R_a} \frac{\eta\chi_k}{n_k m K} \sum_{\tau=1}^{E} \sum_{i=1}^{n_k} \ell_{k,i}'^{(RE+\tau)} \cdot \sigma'(\langle \mathbf{w}_{k,j,r}^{(RE+\tau)}, y_{k,i}\boldsymbol{\mu}_k \rangle)\|\boldsymbol{\mu}_k\|_2^2 \\
&\overset{(a)}{\leq} \overline{\gamma}_{k,j,r}^{(T_c)} - \frac{\eta\chi_k}{n_k m} \sum_{\tau=1}^{E} \sum_{i=1}^{n_k} \ell_i'^{(T_c+\tau)} \cdot \sigma'(\langle \mathbf{w}_{k,j,r}^{(T_c+\tau)}, y_{k,i}\boldsymbol{\mu}_k \rangle)\|\boldsymbol{\mu}_k\|_2^2 \\
&\quad + \sum_{R_c < R < R_a} \frac{\eta\chi_k}{n_k m} \sum_{\tau=1}^{E} \exp(-\sigma(\langle \mathbf{w}_{k,j,r}^{(RE+\tau)}, y_{k,i}\boldsymbol{\mu}_k \rangle) + 1) \cdot \sigma'(\langle \mathbf{w}_{j,r}^{(RE+\tau)}, y_{k,i}\boldsymbol{\mu}_k \rangle)\|\boldsymbol{\mu}_k\|_2^2 \\
&\overset{(b)}{\leq} \overline{\gamma}_{k,j,r}^{(t_c)} + 0.25 \cdot \overline{n}\chi_k \text{SNR}_k^2 \cdot \log(T^*) + 0.25 T^* \exp(-\log(T^*)\overline{n}\chi_k \text{SNR}_k^2) \log(T^*)\overline{n}\chi_k \text{SNR}_k^2,
\end{aligned}
$$

where the first inequality (a) is by the sign of $\ell_{k,i}'$, the second inequality (b) is by Lemma B.3, property of ReLU activation, $\eta \leq mK/(E\|\boldsymbol{\mu}_k\|_2^2)$. Furthermore, we have used the following inequality,

$$
\begin{aligned}
\langle \overline{\mathbf{w}}_{j,r}^{(t)}, \boldsymbol{\mu}_k \rangle &\overset{(a)}{\geq} \langle \overline{\mathbf{w}}_{j,r}^{(0)}, \boldsymbol{\mu}_k \rangle + \overline{\gamma}_{k,j,r}^{(t)} \\
&\overset{(b)}{\geq} -0.5\beta + 0.5\overline{n}\chi_k \text{SNR}_k^2 \log(T^*) \\
&\overset{(c)}{\geq} 0.25\overline{n}\chi_k \text{SNR}_k^2 \log(T^*),
\end{aligned}
$$

where the first inequality (a) is by Lemma B.7, the second inequality (b) is by $\overline{\gamma}_{k,j,r}^{(t)} > 0.5 \log(T^*)\overline{n}\chi_k \text{SNR}_k^2$ and $\langle \overline{\mathbf{w}}_{j,r}^{(0)}, \boldsymbol{\mu}_k \rangle \geq -0.5\beta$ due to the definition of $T_c$ and $\beta$, the last inequality (c) is by $\beta \leq 0.1\overline{n}\chi_k \text{SNR}_k^2 \log(T^*)$. Similarly, for $T_c < t < T_a$ and $y_i = j$, we can also upper bound $\langle \overline{\mathbf{w}}_{j,r}^{(t)}, \boldsymbol{\mu}_k \rangle$ as follows,

$$
\begin{aligned}
\langle \overline{\mathbf{w}}_{j,r}^{(t)}, \boldsymbol{\mu}_k \rangle &\overset{(a)}{\leq} \langle \overline{\mathbf{w}}_{j,r}^{(0)}, \boldsymbol{\mu}_k \rangle + \overline{\gamma}_{k,j,r}^{(t)} \\
&\overset{(b)}{\leq} 0.5\beta + 0.5\overline{n}\chi_k \text{SNR}_k^2 \log(T^*) \\
&\overset{(c)}{\leq} 0.5\overline{n}\chi_k \text{SNR}_k^2 \log(T^*),
\end{aligned}
$$

where the first inequality (a) is by Lemma B.7, the second inequality (b) is by induction hypothesis $\overline{\gamma}_{k,j,r}^{(t)} \leq \overline{n}\chi_k \text{SNR}_k^2 \log(T^*)$, the last inequality (c) is by $\beta \leq 0.1\overline{n}\chi_k \text{SNR}_k^2 \log(T^*)$. $\qquad\square$

### B.2.2 CONVERGENCE ANALYSIS

In this section, our proof is highly inspired by Cao et al. (2022) and Kou et al. (2023).

**Lemma B.9.** *We choose the solution of FedAvg $\overline{\mathbf{W}}^*$ as follows:*

$$
\overline{\mathbf{w}}_{j,r}^* = \overline{\mathbf{w}}_{j,r}^{(0)} + \log(2/\epsilon) \sum_{k=1}^{K} \left( j \cdot \overline{n} \cdot \overline{\text{SNR}}_k^2 \cdot \frac{\boldsymbol{\mu}_k}{\|\boldsymbol{\mu}_k\|_2^2} + \sum_{i=1}^{n_k} \mathbb{1}(j = y_{k,i}) \frac{\boldsymbol{\xi}_{k,i}}{\|\boldsymbol{\xi}_{k,i}\|_2^2} \right).
$$

*Under the same conditions as Theorem 4.3, we have that*

$$
\|\overline{\mathbf{W}}^{(T_1)} - \mathbf{W}^*\|_F \leq \tilde{O}\left( m^{1/2}\overline{n}^{1/2} K \sigma_q^{-1} d^{-1/2} \right).
$$

*Proof of Lemma B.9.* Recall that in the first stage and at time step $T_1$ we know that:

$$
\overline{\mathbf{w}}_{j,r}^{(T_1)} = \overline{\mathbf{w}}_{j,r}^{(0)} + \sum_{k=1}^{K} (j \cdot \overline{\gamma}_{k,j,r}^{(T_1)} \cdot \frac{\boldsymbol{\mu}_k}{\|\boldsymbol{\mu}_k\|_2^2} + \sum_{i=1}^{n_k} \overline{\psi}_{k,j,r,i}^{(T_1)} \cdot \frac{\boldsymbol{\xi}_{k,i}}{\|\boldsymbol{\xi}_{k,i}\|_2^2} + \sum_{i=1}^{n_k} \overline{\phi}_{k,j,r,i}^{(T_1)} \cdot \frac{\boldsymbol{\xi}_{k,i}}{\|\boldsymbol{\xi}_{k,i}\|_2^2}).
$$

Then the distance between the two weights can be calculated as follows:

$$\|\overline{\mathbf{W}}^{(T_1)} - \overline{\mathbf{W}}^*\|_F \overset{(a)}{\leq} \|\overline{\mathbf{W}}^{(T_1)} - \overline{\mathbf{W}}^{(0)}\|_F + \|\overline{\mathbf{W}}^{(0)} - \overline{\mathbf{W}}^*\|_F$$

$$\overset{(a)}{\leq} \sum_k \sqrt{m} \max_{j,r} \frac{\overline{\gamma}_{k,j,r}^{(T_1)}}{\|\boldsymbol{\mu}_k\|_2} + \sum_{k,i} \sqrt{m} \max_{j,r} \frac{|\overline{\psi}_{k,j,r,i}^{(T_1)}|}{\|\boldsymbol{\xi}_{k,i}\|_2} + \sum_{k,i} \sqrt{m} \max_{j,r} \frac{|\overline{\phi}_{k,j,r,i}^{(T_1)}|}{\|\boldsymbol{\xi}_{k,i}\|_2}$$

$$+ O(m^{1/2} \log(1/\epsilon)) \sum_{k=1}^K \left( \overline{n} \cdot \overline{\mathrm{SNR}}_k^2 \|\boldsymbol{\mu}_k\|_2^{-1} + \sqrt{\overline{n}} \max_i \|\boldsymbol{\xi}_{k,i}\|_2^{-1} \right)$$

$$\overset{(b)}{\leq} O(m^{1/2} \|\boldsymbol{\mu}_k\|_2^{-1} \overline{n} \sum_{k=1}^K \mathrm{SNR}_k^2) + O(K\overline{n} m^{1/2} \sigma_p^{-1} d^{-1/2})$$

$$+ O(m^{1/2} \log(1/\epsilon))(\overline{n} \sum_{k=1}^K \overline{\mathrm{SNR}}_k^2 \cdot \|\boldsymbol{\mu}\|_2^{-1} + K\overline{n}^{1/2} \sigma_q^{-1} d^{-1/2})$$

$$\overset{(c)}{\leq} \tilde{O}(m^{1/2}(\overline{n} \sum_{k=1}^K \overline{\mathrm{SNR}}_k^2 \cdot \|\boldsymbol{\mu}\|_2^{-1} + \sqrt{\overline{n}} K \sigma_q^{-1} d^{-1/2})) \overset{(d)}{\leq} \tilde{O}(m^{1/2} \sqrt{\overline{n}} K \sigma_q^{-1} d^{-1/2}),$$

where the first inequality (a) is by triangle inequality, the second inequality (b) is by the decomposition of $\overline{\mathbf{W}}^{(T_1)}$ and the definition of $\overline{\mathbf{W}}^*$, the third inequality (c) is by Lemma B.4, and the last inequality is by direct derivation, the last inequality (d) is by that $\sigma_q^2 d \geq K^2 \overline{n} \|\boldsymbol{\mu}\|_2^2$ in Assumption 4.2. $\qquad\qquad\square$

**Lemma B.10.** *Under the same conditions as Theorem 4.3, we have that $y_{k,i} \langle \nabla f(\overline{\mathbf{W}}^{(t)}, \mathbf{x}_{k,i}), \overline{\mathbf{W}}^* \rangle \geq \log(2/\epsilon)$ for all $i \in [n]$, $k \in [K]$ and $T_1 \leq t \leq T^*$.*

*Proof of Lemma B.10.* Recall that $f(\overline{\mathbf{W}}^{(t)}, \mathbf{x}_{k,i}) = (1/m) \sum_{j,r} j \cdot [\sigma(\langle \overline{\mathbf{w}}_{j,r}, y_{k,i}\boldsymbol{\mu}_k \rangle) + \sigma(\langle \overline{\mathbf{w}}_{j,r}, \boldsymbol{\xi}_{k,i} \rangle)]$, so we have

$$y_{k,i} \langle \nabla f(\overline{\mathbf{W}}^{(t)}, \mathbf{x}_{k,i}), \overline{\mathbf{W}}^* \rangle$$

$$= \frac{1}{m} \sum_{j,r} \sigma'(\langle \overline{\mathbf{w}}_{j,r}^{(t)}, y_{k,i}\boldsymbol{\mu}_k \rangle) \langle \boldsymbol{\mu}_k, j\overline{\mathbf{w}}_{j,r}^* \rangle + \frac{1}{m} \sum_{j,r} \sigma'(\langle \overline{\mathbf{w}}_{j,r}^{(t)}, \boldsymbol{\xi}_{k,i} \rangle) \langle y_{k,i}\boldsymbol{\xi}_{k,i}, j\overline{\mathbf{w}}_{j,r}^* \rangle$$

$$= \frac{1}{m} \sum_{j,r} \sigma'(\langle \overline{\mathbf{w}}_{j,r}^{(t)}, y_{k,i}\boldsymbol{\mu}_k \rangle) 2\log(2/\epsilon) \cdot \overline{n} \cdot \overline{\mathrm{SNR}}_k^2 + \frac{1}{m} \sum_{j,r} \sigma'(\langle \overline{\mathbf{w}}_{j,r}^{(t)}, y_{k,i}\boldsymbol{\mu}_k \rangle) \langle \boldsymbol{\mu}_k, j\overline{\mathbf{w}}_{j,r}^{(0)} \rangle$$

$$+ \frac{1}{m} \sum_{j,r} \sigma'(\langle \overline{\mathbf{w}}_{j,r}^{(t)}, \boldsymbol{\xi}_i \rangle) \langle y_{k,i}\boldsymbol{\xi}_i, j\overline{\mathbf{w}}_{j,r}^{(0)} \rangle + \frac{1}{m} \sum_{i'} \sum_{j,r} \sigma'(\langle \overline{\mathbf{w}}_{j,r}^{(t)}, \boldsymbol{\xi}_{k,i} \rangle) \langle y_i\boldsymbol{\xi}_{k,i}, j\boldsymbol{\xi}_{k,i'} \rangle$$

$$\overset{(a)}{\geq} \frac{1}{m} \sum_{j,r} \sigma'(\langle \overline{\mathbf{w}}_{j,r}^{(t)}, y_{k,i}\boldsymbol{\mu}_k \rangle) 2\log(2/\epsilon)\overline{n} \cdot \overline{\mathrm{SNR}}_k^2 - \frac{1}{m} \sum_{j,r} \sigma'(\langle \overline{\mathbf{w}}_{j,r}^{(t)}, y_{k,i}\boldsymbol{\mu}_k \rangle) \tilde{O}(\sigma_0 \|\boldsymbol{\mu}_k\|_2)$$

$$- \frac{1}{m} \sum_{j,r} \sigma'(\langle \overline{\mathbf{w}}_{j,r}^{(t)}, \boldsymbol{\xi}_{k,i} \rangle) \tilde{O}(\sigma_0 \sigma_p \sqrt{d}) + \frac{1}{m} \sum_{j=y_{k,i},r} \sigma'(\langle \overline{\mathbf{w}}_{j,r}^{(t)}, \boldsymbol{\xi}_{k,i} \rangle) 2\log(2/\epsilon)$$

$$- \frac{1}{m} \sum_{i' \neq i} \sum_{j,r} \sigma'(\langle \overline{\mathbf{w}}_{j,r}^{(t)}, \boldsymbol{\xi}_{k,i} \rangle) \langle \boldsymbol{\xi}_{k,i}, \boldsymbol{\xi}_{k,i'} \rangle \cdot \|\boldsymbol{\xi}_{k,i}\|_2^{-2}$$

$$\overset{(b)}{\geq} \frac{1}{m} \sum_{j,r} \sigma'(\langle \overline{\mathbf{w}}_{j,r}^{(t)}, y_{k,i}\boldsymbol{\mu}_k \rangle) 2\log(2/\epsilon)\overline{n} \cdot \overline{\mathrm{SNR}}_k^2 - \frac{1}{m} \sum_{j,r} \sigma'(\langle \mathbf{w}_{j,r}^{(t)}, y_i\boldsymbol{\mu}_k \rangle) \tilde{O}(\sigma_0 \|\boldsymbol{\mu}_k\|_2)$$

$$- \frac{1}{m} \sum_{j,r} \sigma'(\langle \overline{\mathbf{w}}_{j,r}^{(t)}, \boldsymbol{\xi}_{k,i} \rangle) \tilde{O}(\sigma_0 \sigma_p \sqrt{d}) + \frac{1}{m} \sum_{j=\tilde{y}_i,r} \sigma'(\langle \overline{\mathbf{w}}_{j,r}^{(t)}, \boldsymbol{\xi}_{k,i} \rangle) 2\log(2/\epsilon)$$

$$- \frac{1}{m} \sum_{i' \neq i} \sum_{j,r} \sigma'(\langle \overline{\mathbf{w}}_{j,r}^{(t)}, \boldsymbol{\xi}_{k,i} \rangle) \tilde{O}(1/\sqrt{d}), \tag{28}$$

where the first inequality (a) is by Lemma B.1, and the definition of $\overline{\mathbf{w}}_{j,r}^*$, and the second inequality (b) is by Lemma A.1, and we have used the following inequality:

$$
\begin{aligned}
\langle \overline{\mathbf{w}}_{j,r}^{(t)}, \boldsymbol{\xi}_{k,i} \rangle &= \sum_{k'=1}^{K} \sum_{i'=1}^{n_{k'}} \overline{\psi}_{k',j,r,i'}^{(t)} \|\boldsymbol{\xi}_{k',i'}\|_2^{-2} \langle \boldsymbol{\xi}_{k,i}, \boldsymbol{\xi}_{k',i'} \rangle + \sum_{k'=1}^{K} \sum_{i'=1}^{n_{k'}} \overline{\phi}_{k',j,r,i'}^{(t)} \|\boldsymbol{\xi}_{k',i'}\|_2^{-2} \langle \boldsymbol{\xi}_{k,i}, \boldsymbol{\xi}_{k',i'} \rangle \\
&\overset{(a)}{\geq} \langle \overline{\mathbf{w}}_{j,r}^{(0)}, \boldsymbol{\xi}_{k,i} \rangle + \overline{\psi}_{k,j,r,i}^{(t)} - 8\sqrt{\frac{\log(4n^2/\delta)}{d}} \sum_{k'=1}^{K} \sum_{i' \neq i} |\overline{\psi}_{k',j,r,i'}^{(t)}| \\
&\quad - 8\sqrt{\frac{\log(4n^2/\delta)}{d}} \sum_{k'=1}^{K} \sum_{i' \neq i} |\overline{\phi}_{k',j,r,i'}^{(t)}| \\
&\overset{(b)}{\geq} \langle \overline{\mathbf{w}}_{j,r}^{(0)}, \boldsymbol{\xi}_{k,i} \rangle + \overline{\psi}_{k,j,r,i}^{(t)} - 8C_2 n \sqrt{\frac{\log(4n^2/\delta)}{d}} \\
&\overset{(c)}{\geq} 2 - 2\sqrt{\log(8mn/\delta)} \cdot \sigma_0 \sigma_p \sqrt{d} - 8C_2 n \sqrt{\frac{\log(4n^2/\delta)}{d}} \\
&\overset{(d)}{\geq} 1,
\end{aligned}
\tag{29}
$$

where $C_2$ is a positive constant, (a) is by Lemma A.1 and (b) is by $\overline{\phi}_{k,j,r,i}^{(t)} = 0$ when $y_{k,i} = j$ and $\overline{\psi}_{k,j,r,i} = O(1)$ for all $i \in [\overline{n}]$, (c) is by Lemmas B.1, and (d) is by conditions on $d$ and $\sigma_0$ stated in Assumption 4.2.

Plugging Equation (29) into Equation (28) gives:

$$
\begin{aligned}
y_i \langle \nabla f(\overline{\mathbf{W}}^{(t)}, \mathbf{x}_{k,i}), \overline{\mathbf{W}}^* \rangle &\geq 2\log(2/\epsilon) - \tilde{O}(\sigma_0 \|\boldsymbol{\mu}_k\|_2) - \tilde{O}(\sigma_0 \sigma_p \sqrt{d}) \\
&\overset{(a)}{\geq} \log(2/\epsilon),
\end{aligned}
$$

where the last inequality (a) is by $\sigma_0 \leq \tilde{O}(1) \min\{(\sigma_p \sqrt{d})^{-1}, \|\boldsymbol{\mu}_k\|_2^{-1}\}$ in Assumption 4.2. $\qquad \square$

**Lemma B.11** (Cao et al. (2022)). *Under Assumption 4.2, for $0 \leq t \leq T^*$, the following result holds for all $k \in [K]$.*

$$
\|\nabla L_{\mathcal{S}_k}(\overline{\mathbf{W}}^{(t)})\|_F^2 \leq O\left( \max\left\{ \|\boldsymbol{\mu}_k\|_2^2, \sigma_p^2 d \right\} \right) L_{\mathcal{S}_k}(\overline{\mathbf{W}}^{(t)}).
$$

**Lemma B.12** (Cao et al. (2022)). *Under the same conditions as Theorem 4.3, we have that*

$$
\|\overline{\mathbf{W}}^{(t)} - \overline{\mathbf{W}}^*\|_F^2 - \|\overline{\mathbf{W}}^{(t+1)} - \overline{\mathbf{W}}^*\|_F^2 \geq \eta L_{\mathcal{S}}(\mathbf{W}^{(t)}) - \eta\epsilon
$$

*for all $T_1 \leq t \leq T^*$.*

**Lemma B.13.** *Under the same conditions as Theorem 4.3, let $T = T_1 + \left\lfloor \frac{\|\overline{\mathbf{W}}^{(T_1)} - \overline{\mathbf{W}}^*\|_F^2}{2\eta\epsilon} \right\rfloor = T_1 + \tilde{O}(m\eta^{-1}\epsilon^{-1}\overline{n}K^2\sigma_q^{-2}d^{-1})$. Then we have $L_{\mathcal{S}}(\overline{\mathbf{W}}^{(t)}) \leq \epsilon$ for some $T_1 \leq t \leq T$.*

*Proof of Lemma B.13.* By Lemma B.12, for any $t \in [T_1, T]$, we have that

$$
\|\overline{\mathbf{W}}^{(s)} - \overline{\mathbf{W}}^*\|_F^2 - \|\overline{\mathbf{W}}^{(s+1)} - \overline{\mathbf{W}}^*\|_F^2 \geq \eta L_{\mathcal{S}}(\overline{\mathbf{W}}^{(s)}) - \eta\epsilon
$$

holds for $s \leq t$. Taking a summation, we obtain that

$$
\sum_{s=T_1}^{t} L_{\mathcal{S}}(\overline{\mathbf{W}}^{(s)}) \leq \frac{\|\overline{\mathbf{W}}^{(T_1)} - \overline{\mathbf{W}}^*\|_F^2 + \eta\epsilon(t - T_1 + 1)}{\eta}
\tag{30}
$$

for all $T_1 \leq t \leq T$. Dividing $(t - T_1 + 1)$ on both side of Equation (30) gives that

$$
\frac{1}{t - T_1 + 1} \sum_{s=T_1}^{t} L_{\mathcal{S}}(\overline{\mathbf{W}}^{(s)}) \leq \frac{\|\overline{\mathbf{W}}^{(T_1)} - \overline{\mathbf{W}}^*\|_F^2}{\eta(t - T_1 + 1)} + \epsilon.
$$

Then we can take $t = T$ and have that

$$\frac{1}{T - T_1 + 1} \sum_{s=T_1}^{T} L_{\mathcal{S}}(\overline{\mathbf{W}}^{(s)}) \leq \frac{\|\overline{\mathbf{W}}^{(T_1)} - \overline{\mathbf{W}}^*\|_F^2}{\eta(T - T_1 + 1)} + \epsilon$$

$$\leq \frac{m^{1/2} \overline{n}^{1/2} K \sigma_q^{-1} d^{-1/2}}{\eta(T - T_1 + 1)} + \epsilon < 2\epsilon,$$

where the second inequity is by Lemma B.9. Therefore, our choice that $T = T_1 + \left\lfloor \frac{\|\mathbf{W}^{(T_1)} - \overline{\mathbf{W}}^*\|_F^2}{2\eta\epsilon} \right\rfloor$. Because the mean is smaller than $\epsilon$, we can conclude that there exist $T_1 \leq t \leq T$ such that $L_{\mathcal{S}}(\overline{\mathbf{W}}^{(t)}) < \epsilon$. $\qquad\square$

## B.3 POPULATION LOSS

Consider a new data point $(\mathbf{x}_k, y) \sim \mathcal{D}_k$ drawn from the distribution of client $k$. Moreover, by the signal-noise decomposition, the learned neural network has parameter:

$$\overline{\mathbf{w}}_{j,r}^* = \overline{\mathbf{w}}_{j,r}^{(0)} + \log(2/\epsilon) \sum_{k=1}^{K} \left( j \cdot \overline{n} \cdot \overline{\text{SNR}}_k^2 \cdot \frac{\boldsymbol{\mu}_k}{\|\boldsymbol{\mu}_k\|_2^2} + \sum_{i=1}^{n_k} \mathbb{1}(j = y_{k,i}) \frac{\boldsymbol{\xi}_{k,i}}{\|\boldsymbol{\xi}_{k,i}\|_2^2} \right),$$

for $j \in \{\pm 1\}$ and $r \in [m]$. The calculation of the test error follows the results established in Theorems E.1 and E.3 from Kou et al. (2023).

**Lemma B.14** (Restatement of Theorem 4.4). *Let $T$ be defined in Theorem 4.3. Under the same conditions as Theorem 4.3, there exists $0 \leq t \leq T$ such that if $\frac{\overline{n}\chi_k^2\|\boldsymbol{\mu}\|_2^4}{K\sigma_p^4 d} = \Omega(1)$ then the test error satisfies $L_{\mathcal{D}_k}(\overline{\mathbf{W}}^{(t)}) \leq \exp\left(-c\frac{\overline{n}\chi_k^2\|\boldsymbol{\mu}\|_2^4}{K\sigma_p^4 d}\right).$*

*Proof of Lemma B.14.* Recall that the test error is define as $P_{(\mathbf{x},y)\sim\mathcal{D}_k}(y \neq f(\overline{\mathbf{W}}^{(t)}, \mathbf{x}))$ It is equivalent to calculate that

$$P_{(\mathbf{x},y)\sim\mathcal{D}_k}(y \neq f(\overline{\mathbf{W}}^{(t)}, \mathbf{x})) = P_{(\mathbf{x},y)\sim\mathcal{D}_k}(yf(\overline{\mathbf{W}}^{(t)}, \mathbf{x}) < 0).$$

It is essential to calculate the output function of neural network which follows:

$$yf(\overline{\mathbf{W}}^{(t)}, \mathbf{x}) = \frac{1}{m} \sum_{j,r} \sigma(\langle \overline{\mathbf{w}}_{j,r}^{(t)}, y\boldsymbol{\mu}_k \rangle) + \frac{1}{m} \sum_{j,r} \sigma(\langle \overline{\mathbf{w}}_{j,r}^{(t)}, \boldsymbol{\xi}_{k,i} \rangle).$$

We first calculate the signal learning part:

$$\langle \overline{\mathbf{w}}_{j,r}^{(t)}, y\boldsymbol{\mu}_k \rangle = \langle \overline{\mathbf{w}}_{j,r}^{(0)}, y\boldsymbol{\mu}_k \rangle + jy\overline{\gamma}_{k,j,r}^{(t)} + \sum_{k=1}^{K} \sum_{i=1}^{n} (\overline{\psi}_{k,j,r,i}^{(t)} + \overline{\phi}_{k,j,r,i}^{(t)})\langle \boldsymbol{\xi}_{k,i}, y\boldsymbol{\mu}_k \rangle \|\boldsymbol{\xi}_{k,i}\|_2^{-2}$$

$$= \Theta(yj\overline{\gamma}_{k,j,r}^{(t)}).$$

Next, we calculate the noise vector part. It is known that $\langle \overline{\mathbf{w}}_{j,r}^{(t)}, \boldsymbol{\xi}_{k,i} \rangle$ is a Gaussian distribution with mean zero. Let $\tilde{\mathbf{w}}_{j,r}^{(t)} = \overline{\mathbf{w}}_{j,r}^{(t)} - \sum_{k=1}^{K} j \cdot \overline{\gamma}_{k,j,r}^{(t)} \cdot \frac{\boldsymbol{\mu}_k}{\|\boldsymbol{\mu}_k\|_2^2}$, then we have that $\langle \tilde{\mathbf{w}}_{j,r}^{(t)}, \boldsymbol{\xi}_{k,i} \rangle = \langle \overline{\mathbf{w}}_{j,r}^{(t)}, \boldsymbol{\xi}_{k,i} \rangle$ and

$$\|\tilde{\mathbf{w}}_{j,r}^{(t)}\|_2 \leq \Theta(1/(\sigma_p\sqrt{dn})) \sum_{k=1}^{K} \sum_{i=1}^{n} \overline{\psi}_{k,j,r,i}. \tag{31}$$

As a result, by the condition $\frac{\overline{\gamma}_{k,j,r}^{(t)}}{\sum_{k=1}^{K} \sum_{i=1}^{n_k} \overline{\psi}_{k,j,r,i}^{(t)}} = \Theta(\frac{\overline{n}\chi_k\|\boldsymbol{\mu}\|_2^2}{n\sigma_q^2 d})$, we have that:

$$P_{(\mathbf{x},y)\sim\mathcal{D}_k}(yf(\overline{\mathbf{W}}^{(t)}, \mathbf{x}) < 0) \leq P_{(\mathbf{x},y)\sim\mathcal{D}_k}\left[ \sum_r \sigma(\langle \overline{\mathbf{w}}_{j,r}^{(t)}, y\boldsymbol{\mu}_k \rangle) \geq \sum_r \sigma(\langle \overline{\mathbf{w}}_{j,r}^{(t)}, \boldsymbol{\xi}_{k,i} \rangle) \right]$$

$$\leq \exp\left(-c\frac{\overline{n}\chi_k^2\|\boldsymbol{\mu}\|_2^4}{K\sigma_p^4 d}\right).$$

$\qquad\square$

## C    SUPPLEMENTS FOR EXPERIMENTS

### C.1    EXPERIMENTAL DETAILS FOR WEIGHTED FEDAVG ON REAL-WORLD DATASET

Following the footsteps of the prior works (Acar et al., 2021; Shamsian et al., 2021), A ConvNet LeCun et al. (1998) with two convolutional layers and three fully-connected layers is adopted for experiments on CIFAR10 and CIFAR100 datasets. The real-world dataset comprises five distinct sub-datasets exhibiting feature shift: SVHN (Netzer et al., 2011), USPS (Hull, 1994), SynthDigits (Ganin & Lempitsky, 2015), MNIST-M (Ganin & Lempitsky, 2015), and MNIST (Wang et al., 2014), where each domain serves as a client.

As for the Digits dataset, we utilized another ConvNet LeCun et al. (1998) similar to Li et al. (2021) with four convolutional layers and three fc layers. For all experiments in this part, models were trained with 300 communication rounds and all tasks were optimized with the SGD optimizer. For experiments on CIFAR10/CIFAR100, we set the learning rate 0.03 and randomly sampled half of the clients participating in each communication round. For experiments on Digits, the learning rate is set to 0.01 and all clients are involved in each communication round. Algorithms were implemented on PyTorch Paszke et al. (2019) with an RTX 3090 Ti GPU. The learned models are tested on the private test data of each client. To be specific, we run five times trials and report their mean and std. Codes are available at https://anonymous.4open.science/r/fed-feature-learning-31E9/.

### C.2    FEATURE LEARNING PROCESS IN FL

We provide an experimental evidence to validate our theoretical analysis of feature learning for FL, as shown in Figure 2. In this experiment, we engage two clients under the IID setting. Both the data and neural networks are synthesized in accordance with our theoretical framework. Both data and neural networks are synthetic by our theoretical setting. In the plot, we illustrate the learning trajectory for client 1 for client 1 in terms of signal learning represented by $\max_{j,r} \gamma_{j,r}$ and noise memorization represented by $\max_{j,r} \rho_{j,r}$. We observe that, with the help of communication, the signal learning keeps increasing without any degradation. In contrast, noise memorization exhibits a marked decline with at each weight averaging. This observed behavior further reinforces our theoretical analysis.

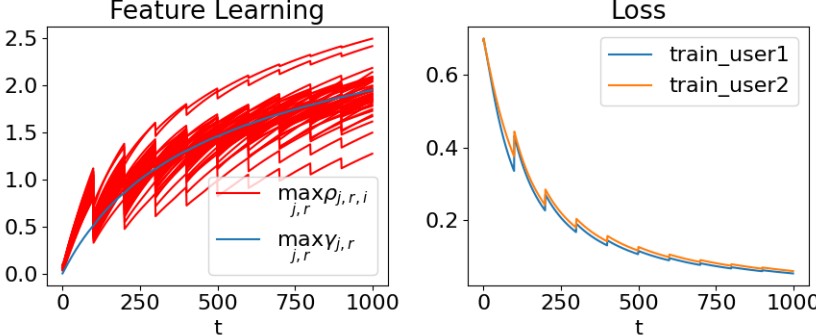

Figure 2: Feature learning trajectory of FedAvg on synthetic data. The signal learning (blue curve) keeps increasing without any degradation. In contrast, noise memorizations (red curves) exhibit a marked decline with at each weight averaging.

### C.3    ADDITIONAL EXPERIMENT WITH HIGH DIMENSION SETTING

We further examine our theoretical result by experimental simulation with a high dimension setting. The training data size is set to $\bar{n}_{train} = 100$ and the testing data size is set to $n_{test} = 2000$ with instance dimension to $d = 8000$ for each client. The result is shown in Figure 3. These additional experiments confirm that Federated Averaging (FedAvg) can indeed achieve better results than local training methods, aligning with our theoretical predictions.

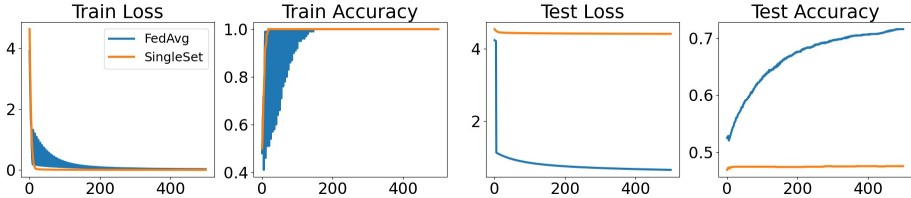

Figure 3: Convergence behavior comparison of train loss, train accuracy, test loss, and test accuracy on synthetic data. Both local training and FedAvg demonstrate convergence on the training set. FedAvg outperforms significantly on the testing set.

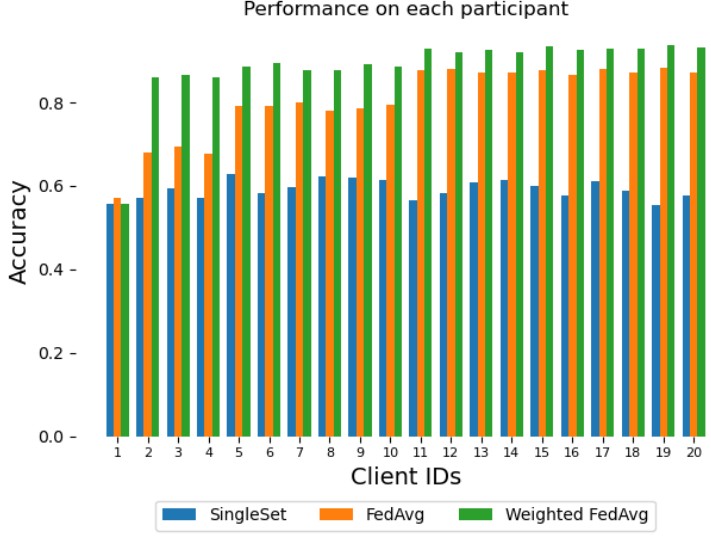

Figure 4: Comparison results on each client under non-IID setting.

### C.4 FEATURE LEARNING IN NON-IID SETTING

To simulate the non-IID scenario, we split 20 clients into four groups. Within each group, the signal vectors are identical. However, signal vectors from different groups are orthogonal to each other. Specifically, the client groupings are as follows: $\{k\}_{k=1}, \{k\}_{k\in[2,4]}, \{k\}_{k\in[5,10]}$, and $\{k\}_{k\in[11,20]}$. Besides, we implement the Weighted FedAvg algorithm, where each client only communicates with those clients in the same group. The test accuracy on each client is demonstrated in Figure 4. The findings indicate that as the number of members in a group increases, the performance of both FedAvg and Weighted FedAvg improves. This aligns with our theoretical insights regarding the generalization of Federated Learning (FL).

## D ADDITIONAL DISCUSSION

### D.1 NAME OF EMPLOYED MODEL AS CNN

Our choice to label the model as a CNN is based on the specific characteristics of the data model we used. We adopt a two-patch model, where $\mathbf{x}_{k,i} = [\mathbf{x}_{k,i}^{(1)}, \mathbf{x}_{k,i}^{(2)}] = [y_{k,i}\boldsymbol{\mu}_k, \boldsymbol{\xi}_{k,i}]$. We implement a convolution operation using a single weight across these two patches, akin to applying a filter in traditional CNNs. This is concisely represented as follows:

$$f = \frac{1}{m}\sum_{r=1}^{m}[\sigma(\mathbf{w}_{k,+1,r}^{\top}\mathbf{x}_{k,i}^{(1)}) + \sigma(\mathbf{w}_{k,+1,r}^{\top}\mathbf{x}_{k,i}^{(2)})] - \frac{1}{m}\sum_{r=1}^{m}[\sigma(\mathbf{w}_{k,-1,r}^{\top}\mathbf{x}_{k,i}^{(1)}) + \sigma(\mathbf{w}_{k,-1,r}^{\top}\mathbf{x}_{k,i}^{(2)})].$$

Here the filter $\mathbf{w}_{k,j,r}$ operates on two patch, with $k \in [K]$ denoting the index of client, $j \in \{+1, -1\}$ corresponding to the weights value at the second layer, and $r \in [m]$ as the filter index. Our approach alongside fixing the second layer is consistent with the related works [Cao et al. (2022); Kou et al. (2023)], where similar network structures applying weights across multiple data patches are identified as CNNs.

## D.2 RELATION BETWEEN CONVERGENCE AND DATA HETEROGENEITY

The effect of data heterogeneity on convergence is a critical aspect of our analysis and highlights the significant influence on convergence rates. As demonstrated in Theorem 4.4, we establish condition related to the signal-to-noise ratio (SNR):

$$\frac{\overline{n}\chi_k^2\|\boldsymbol{\mu}_k\|_2^4}{K\sigma_p^4 d} = \Omega(1)$$

with the effective SNR for each climent $k$ given by $\overline{\mathrm{SNR}}_k = \left(\sum_{k'=1}^{K} \frac{\langle\boldsymbol{\mu}_k, \boldsymbol{\mu}_{k'}\rangle}{\|\boldsymbol{\mu}_k\|_2^2}\right) \mathrm{SNR}_k$. In scenarios with increased data heterogeneity, the term $\chi_k = \sum_{k'=1}^{K} \frac{\langle\boldsymbol{\mu}_k, \boldsymbol{\mu}_{k'}\rangle}{\|\boldsymbol{\mu}_k\|_2^2}$ tends to decrease. To maintain a consistent SNR level in such cases, it becomes necessary to reduce $\sigma_p^2 d$, which, according to the convergence time in Theorem 4.3: $T = \tilde{\Theta}(\eta^{-1}Km\overline{n}\sigma_p^{-2}d^{-1} + \eta^{-1}\epsilon^{-1}m\overline{n}\sigma_p^{-2}K^2d^{-1})$ results in slower convergence. On the contrary, **reducing data heterogeneity will shorten the convergence time**.

## D.3 EFFECT OF THE NUMBER OF LOCAL EPOCHS ON THE CONVERGENCE AND GENERALIZATION OF FEDAVG

In our analysis, the term $T$ in Theorem 4.3 represents the total number of local updates, defined as $T = RE$, where $R$ is the number of commutation rounds and $E$ is the number of local epochs.

Under our assumption on the learning rate, where $\eta \leq \tilde{O}(\frac{K}{E}\min\{\|\boldsymbol{\mu}\|_2^{-2}, \sigma_p^{-2}d^{-1}\})$, **increasing $E$ may lead to slower convergence**. This observation aligns with the empirical results reported in McMahan et al. (2017). The impact of varying $E$ on the required number of communication rounds to achieve adequate convergence is demonstrated in Table 2 of McMahan et al. (2017), which compares the communication rounds needed for the same training accuracy on MNIST.

Besides, it is important to note that **a smaller $E$ does not necessarily imply better generalization**. To explore this relationship further, we conducted additional experiments of FedAvg on CIFAR 10. From Table 3 we can see that the generalization performance of FedAvg is not closely related to the number of local epochs.

Table 3: Relation between local epochs and test accuracy.

| $E/R$ | 1 | 20 | 40 | 60 | 80 |
|---|---|---|---|---|---|
| 5 | 48.18 | 66.28 | 66.24 | 64.98 | 64.55 |
| 10 | 54.17 | 64.6 | 64.48 | 63.44 | 64.5 |
| 15 | 54.34 | 64.1 | 63.91 | 63.44 | 64.33 |

