# OpenReview forum: "Understanding Convergence and Generalization in Federated Learning through Feature Learning Theory"
_ICLR.cc/2024/Conference — ICLR 2024 poster_

### Official Review · Reviewer_aCq5 · 2023-10-23

**Soundness:** 3 good
**Presentation:** 3 good
**Contribution:** 3 good
**Rating:** 6
**Confidence:** 3

**Summary:**

This paper studies the optimization dynamics and generalization behavior of FedAvg several neural networks in federated learning. Inspired by feature learning theory, they prove that FedAvg converges with gradient descent optimization under a certain data-generating model.

**Strengths:**

The paper is generally well-written, making me easy to follow the logic flow. The motivation is strong, and the theoretical results are intuitive. Moreover, the authors honestly discuss the limitations of the paper in conclusion.

**Weaknesses:**

The contribution of this paper is still a bit unclear to me, probably since I am not familiar with federated learning: It seems that the optimization dynamics turns to be *explicit* in the setting of this paper (a certain data-generating model, homogeneous two-layer CNNs and gradient descent optimization), and the results seem *direct* under the classical analysis of gradient descent. I am wonder whether this setting is representative enough, could the authors elaborate the technical innovations on this point?

Another thing that I am concerned is that the current result are strongly related to the neural networks. In my understanding, federated learning is a topic *independent* of neural networks. Could the authors discuss the probabilities that the theory in this paper can go beyond just using two-layer CNNs to be individual models?

(minor) Some symbols and equations should be defined and clearly explained. The authors should check Section 3 and 4 to make sure that all symbols are defined in their occurances.

**Questions:**

- It would be better if the authors explain the rationale behind the two-patch feature generation model again *just below* its definition. Moreover, should the signal patch and the noise patch necessarily have same dimension?
- What is the expression of the distribution $P(\mu^{(1)},\mu^{(2)},\cdots,\mu^{(C)})$?
- Page 3: $F_{-1}(W_{+1}, x)$ should be $F_{-1}(W_{-1}, x)$
- Page 4: $yf(\bar{W}^{(t)}, x<0)$ should be $yf(\bar{W}^{(t)}, x)<0$
- What is $w_{k',j,r}$ in eq.(6) and (8)? It seems that the right-hand-side is independent of $k'$.
- (minor) Theorem 4.3 is a "best-iterate" result. Could it be turned into a "last-iterate" or "average-iterate" guarantee?
- The discussion below Theorem 4.4 could be made more accessible. Specifically, when justifying the superiority of FedAvg compared to local training, could the authors elaborate more, rather than just discussing on two equations on SNR?

---

> ### Author Response · Authors · 2023-11-20
> **Response to Reviewer aCq5 [Part I]**
>
> Thank you for your thoughtful review and the constructive comments on our submission. We are pleased that you found the paper well-structured and the strong motivation and theoretical results. We appreciate your feedback and would like to address your concerns as follows:
>
> ___
>
> > **Weakness 1 The contribution of this paper is still a bit unclear to me, probably since I am not familiar with federated learning: It seems that the optimization dynamics turns to be explicit in the setting of this paper (a certain data-generating model, homogeneous two-layer CNNs and gradient descent optimization), and the results seem direct under the classical analysis of gradient descent. I am wonder whether this setting is representative enough, could the authors elaborate the technical innovations on this point?**
>
> While our setting might initially appear explicit with homogeneous two-layer CNNs and gradient descent optimization, the complexity and challenges of our analysis are significantly increased by the inherent non-convexity of two-layer neural networks and the unique dynamics of federated learning. We elaborate on our main contributions as follows:
>
> 1. **Impact of Weight Averaging:**
>
> Unlike single network models, federated learning involves weight averaging across multiple neural networks. This introduces additional complexity, as the training dynamics of each network are influenced by this averaging process. This will lead to the dynamic correlation between neural networks after weight averaging. Besides, despite the fact that weight averaging is a linear combination of weight, its impact on the nonlinear dynamics of individual clients is substantial. We address this by employing a global weight decomposition approach, enabling us to effectively track the dynamics of feature learning within this federated framework.
>
> 2. **Impact of Data heterogeneity**
>
> In federated learning, unlike in single network scenarios, the design of a data model distribution that encompasses both iid and non-iid cases is a significant challenge. To capture the nuances of data heterogeneity and its impact on generalization, we introduce an effective signal-to-noise ratio. This innovative approach allows us to characterize the influence of data diversity on the learning process more accurately.
>
>
> ___
>
> > **Weakness 2: Another thing that I am concerned is that the current result are strongly related to the neural networks. In my understanding, federated learning is a topic independent of neural networks. Could the authors discuss the probabilities that the theory in this paper can go beyond just using two-layer CNNs to be individual models?**
>
> While federated learning is a broad topic that encompasses a variety of models, our research specifically addresses the challenges and dynamics associated with neural networks, which are increasingly prevalent in modern federated learning applications. The urgency to analyze federated learning within the neural network paradigm stems from both their widespread usage and the complexity of their optimization and generalization characteristics, particularly given their non-convex nature.
> The main motivation for our work is the lack of existing studies that comprehensively reveal the generalization gap between Federated Averaging (FedAvg) and local training within the neural network framework. Our focus on two-layer CNNs is driven by the need to tackle these hard-to-analyze aspects and to provide foundational insights into how federated learning operates in neural network settings.
>
> Regarding the possibility of extending our results to more complex neural network models, we recognize this as a challenging yet intriguing direction for future research. Currently, most studies on feature learning, including our references [1,2,3], concentrate on two-layer CNNs due to their tractable analysis framework. Expanding this research to encompass more intricate neural network architectures presents a valuable opportunity for advancing the field and is an exciting prospect for subsequent work.
>
> ___
>
> > **Weakness 3: (minor) Some symbols and equations should be defined and clearly explained. The authors should check Section 3 and 4 to make sure that all symbols are defined in their occurances.**
>
> We appreciate the reviewer's attention to detail in highlighting the need for clearer definitions and explanations of certain symbols and equations in our paper. In these sections, we have now ensured that:
>
> - All symbols are defined clearly at their first point of use.
>
> - Equations are explained with adequate context and clarity to facilitate better understanding

---

> > ### Author Response · Authors · 2023-11-20
> > **Response to Reviewer aCq5 [Part II]**
> >
> > > **Question 1:It would be better if the authors explain the rationale behind the two-patch feature generation model again just below its definition. Moreover, should the signal patch and the noise patch necessarily have the same dimension?**
> >
> > Thank you for your suggestion regarding the explanation of the two-patch feature generation model. We have revised our manuscript to include a more detailed explanation of this model just below its definition.
> >
> > Regarding the dimensionality of the signal and noise patches, they are indeed designed to have the same dimensions in our model. This design choice is particularly relevant for the convolutional neural network framework we employ, where a single weight is used in relation to both signal and noise patches. Having equal dimensions for both types of patches simplifies the convolution operation and aligns with the structure of typical CNNs.
> >
> > However, it is possible to introduce more patches to adjust the dimensionality ratio between signal and noise patches to explore different model configurations.
> >
> > ___
> >
> > > **Question 2: What is the expression of the distribution P**
> >
> > $P$ is defined as a discrete distribution that assigns probabilities to each element in the range from 1 to $C$, where
> > $C$ represents the number of classes or distinct data categories.$P$ is a discrete distribution assigning probabilities to each element from $1$ to $C$. It governs the likelihood of each class or category appearing within the data set of a client. We have added this information in our updated manuscript.
> >
> > ___
> >
> > > **Question 3 & 4: Page 3: W_{+1,-1}  should be; Page 4: should be <0**
> >
> > Thanks for pointing them out, we have modified them.
> >
> > ___
> >
> > > **Question 5 What is  in eq.(6) and (8)? It seems that the right-hand-side is independent of k'**
> >
> > To address the concerns regarding Equation (6) and Equation (8), we offer the following clarification:
> >
> > The rewritten form of Equation (6) is as follows:
> >
> > $ \mathbf{w}\_{k,j,r}^{(t)}  =  \sum\_{k'=1}^K (
> >    \alpha^{(t)}\_{k',j,r}   \mathbf{w}\_{k',j,r}^{(0)} + \gamma\_{k',j,r}^{(t)}   \| \boldsymbol{\mu}\_{k'} \|^{-2}\_2   \boldsymbol{\mu}\_{k'}  +  \sum_{i=1}^{n\_{k'}} \rho\_{k',j,r,i}^{(t)}  \|  {\boldsymbol{\xi}}_{k',i} \|^{-2}_2    {\boldsymbol{\xi}}\_{k',i}) $
> >
> > for all $k \in [K]$. The right-hand side of this equation involves a summation over all $k’ \in [K]$, which includes the specific $k$ on the left hand side. This formula represents the iteration considering weight averaging in our model. Crucially, in each iteration, the variable $\gamma^{(t)}\_{k,j,r}$ and $\rho^{(t)}\_{k,j,r,i}$ will be differ for each $k$, leading to distinct growth rates for $\mathbf{w}\_{k,j,r}$ across different clients.
> >
> > ___
> >
> > > **Question 6: (minor) Theorem 4.3 is a "best-iterate" result. Could it be turned into a "last-iterate" or "average-iterate" guarantee?**
> >
> > Our belief is that with detailed analysis, Theorem 4.3 can be extended to a last-iterate guarantee. Specifically, in the second stage of the optimization dynamics, although the loss derivative is non-constant, our preliminary observations suggest that the iterative process of weight adjustment tends to approach the optimal solution progressively. This behavior is indicative of the potential for a last-iterate guarantee.
> >
> >
> > ___
> >
> > > **Question 7: The discussion below Theorem 4.4 could be made more accessible. Specifically, when justifying the superiority of FedAvg compared to local training, could the authors elaborate more, rather than just discussing on two equations on SNR?**
> >
> > We appreciate your suggestion to provide a more comprehensive explanation of the advantages of Federated Averaging (FedAvg) compared to local training. In response, we have expanded our discussion in the manuscript to include a new perspective on this matter.
> >
> > In the added section, we present the viewpoint that FedAvg can, under certain conditions, be equivalent to centralized training, where all datasets are accessible for model training. This equivalence is significant because it suggests that FedAvg has the potential to achieve the same, if not better, results than local training methods. Centralized training typically results in better generalization due to the utilization of a more diverse and comprehensive dataset. In contrast, local training is often limited by the quantity and variability of data available to each client.
> >
> > ___
> >
> > **References**
> >
> > [1] Frei, Spencer, Niladri S. Chatterji, and Peter Bartlett. "Benign overfitting without linearity: Neural network classifiers trained by gradient descent for noisy linear data." Conference on Learning Theory. PMLR, 2022.
> >
> > [2] Cao, Yuan, et al. "Benign overfitting in two-layer convolutional neural networks." Advances in neural information processing systems 35 (2022): 25237-25250.
> >
> > [3]  Kou, Yiwen, Zixiang Chen, Yuanzhou Chen, and Quanquan Gu. "Benign Overfitting in Two-layer ReLU Convolutional Neural Networks." International Conference on Machine Learning (2023).

---

> > > ### Comment · Reviewer_aCq5 · 2023-11-21
> > >
> > > Thank you for the response. The authors' response has resolved my main concerns. Given that the authors will make revision as promised, I will no longer stand on the way of acceptance.

---

> > > > ### Author Response · Authors · 2023-11-21
> > > >
> > > > Thank you for your positive feedback and support towards the acceptance of our paper! We have updated the manuscript reflecting the changes discussed, and we're grateful for your constructive guidance in this process.

---

### Official Review · Reviewer_bJ2V · 2023-10-28

**Soundness:** 3 good
**Presentation:** 3 good
**Contribution:** 3 good
**Rating:** 6
**Confidence:** 4

**Summary:**

This paper studies a scenario where each
client employs a two-layer CNN for local training
on their own data by tracking the trajectory of signal learning and noise
memorization in FLs. The paper shows
that FedAvg can achieve near-zero test error by effectively increasing the signal-to-noise ratio in feature learning, while pure local training without communication
achieves a large constant test error. Inspired by the theoretical results, the paper proposes a heuristic weighted FedAvg which leverages the similarity of local representations into mixing weights.

**Strengths:**

The paper presents a fresh perspective and innovative techniques. It successfully demonstrates fruitful results, offering valuable insights into the generalization of FedAvg.

**Weaknesses:**

1. My primary concern with this work is its inability to effectively account for the influence of local training rounds (i.e., $E$ in the paper) on generalization performance. The paper only demonstrates that when $E=\infty$, which signifies no server-worker communication, the generalization performance is inferior compared to the scenario with $E<\infty$. This can be partly attributed to the reduced amount of data utilized in client collaboration. It will be more exciting to see any trend within the regime $1\leq E<\infty$.

2. The paper establishes that training convergence is independent of data heterogeneity, a finding that appears to contrast with a substantial body of research and empirical observations in FL. It would be greatly beneficial if the authors could provide further elucidation regarding why this particular model exhibits this behavior.

I am open to raising my score if the two points can be adequately addressed.

3. A minor point relates to the clarity of the writing. The paper's notations appear somewhat redundant and disorganized. For instance, the notations $\mu^{(1)}, \dots, \mu^{(C)}$ are not used beyond Section 3.2, and the definition of $\sigma_p$, which plays a pivotal role in signal-to-noise ratios, is missing. In Equation (4), it might be more appropriate to replace $n$ with $n_k$. I recommend that the authors thoroughly review the manuscript to enhance its overall clarity and coherence.

**Questions:**

NA


=================================

I raised my score after the rebuttal.

---

> ### Author Response · Authors · 2023-11-20
> **Response to Reviewer bJ2V**
>
> Thank you for your insightful feedback and constructive comments on our submission. We appreciate the opportunity to clarify your concerns and questions as follows:
>
> ___
>
> > **Weakness 1: My primary concern with this work is its inability to effectively account for the influence of local training rounds (i.e., E iin the paper) on generalization performance. The paper only demonstrates that when  which signifies no server-worker communication, the generalization performance is inferior compared to the scenario with . This can be partly attributed to the reduced amount of data utilized in client collaboration. It will be more exciting to see any trend within the regime**
>
> We are a bit confused about this comment “The paper only demonstrates that when $E= \infty$ …” as we never do that. Our analysis actually focuses on the regime where  $ 0 < E < \infty $. This means that during the communication rounds, the number of local updates $E$ is finite. In particular, we use $T = E R$ to count the total number of local update steps, where $R$ is the number of communication rounds. In summary, all of our theoretical analyses are based on  $ 0 < E < \infty $ instead of $E= \infty$.
>
> ___
>
> > **Weakness 2: The paper establishes that training convergence is independent of data heterogeneity, a finding that appears to contrast with a substantial body of research and empirical observations in FL. It would be greatly beneficial if the authors could provide further elucidation regarding why this particular model exhibits this behavior.**
>
> The effect of data heterogeneity on convergence is a critical aspect of our analysis and highlights the significant influence on convergence rates. As demonstrated in Theorem 4.4,  we establish condition related to the signal-to-noise ratio (SNR):
>
> $ \overline{n} \cdot \frac{\|\boldsymbol{\mu}\_k \|^2\_2}{\sigma^2\_p} \cdot \overline{\mathrm{SNR}}^2\_k =  \Omega( 1 )$
> with the effective SNR for each climent $k$ given by
>  $ \overline{\mathrm{SNR}}\_{k} = \left(\sum\_{k'=1}^K \frac{\langle \boldsymbol{\mu}\_k, \boldsymbol{\mu}\_{k'}  \rangle}{ \| \boldsymbol{\mu}_k \|^2\_2} \right) \mathrm{SNR}_k   $
>
> In scenarios with increased data heterogeneity, the term $\sum\_{k'=1}^K \frac{\langle \boldsymbol{\mu}\_k, \boldsymbol{\mu}\_{k'}  \rangle}{ \| \boldsymbol{\mu}_k \|^2_2}$ tends to decrease. To maintain a consistent SNR level in such cases, it becomes necessary to reduce $\sigma^2_p d$, which, according to the convergence time in Theorem 4.3:
>  $T = \tilde{\Theta}( \eta^{-1} K mn\sigma\_0 ^{-1} \sigma^{-2}\_p d^{-1} +  \eta^{-1}\epsilon^{-1} mn \sigma^{-2}\_p K d^{-1})$
>
> results in slower convergence.  On the contrary, if we reduce the data heterogeneity, the convergence time will become shorter.
>
> ___
>
> > **Weakness 3: A minor point relates to the clarity of the writing. The paper's notations appear somewhat redundant and disorganized. For instance, the notations  are not used beyond Section 3.2, and the definition of  which plays a pivotal role in signal-to-noise ratios, is missing. In Equation (4), it might be more appropriate to replace with. I recommend that the authors thoroughly review the manuscript to enhance its overall clarity and coherence.**
>
> We greatly appreciate your feedback regarding the clarity and organization of notations in our manuscript.
>
> 1. We would like to clarity in section 4.2, we have used $|\mu^{(C)}|$;
> We have introduced a clear definition of $\sigma_p$, which represents the strength of the noise vector. The definition is now presented as follows:
>
> 2.  Noise vector is distributed as  $\boldsymbol{\xi}\_{k,i} \sim \mathcal{N}(\mathbf{0}, {\sigma^2\_p}\mathbf{I}-\sum\_{c=1}^C \boldsymbol{\mu}^{(c)} {\boldsymbol{\mu}^{(c)}}^\top/\| \boldsymbol{\mu}^{(c)} \|\_2^{2})$,
>
> 3. Thanks for your suggestion, we have replaced the term $n_k$ in Eq (4).
> We have conducted a thorough review of the manuscript to ensure that all notations are used consistently and defined clearly at their first point of use.

---

> > ### Comment · Reviewer_bJ2V · 2023-11-20
> >
> > Thanks for your response.
> >
> > Just for clarification, for weakness 1, you compared FedAvg ($0<E<\infty$) with pure local training (i.e., no client-server communication) in the last paragraph of Section 4. The latter can somewhat interpreted as FedAvg but with $E=\infty$. It is clear and natural that FedAvg outperforms pure local training in generalization due to more accessible data (or equivalently, stochastic gradients). However, it is more interesting to me to see how generalization performance varies with $E\in(0,\infty)$. Any discussion regarding this is valued.
> >
> > The responses to other points make sense to me. I will reevaluate after further clarifications are given.

---

> > > ### Author Response · Authors · 2023-11-21
> > >
> > > Thank you for your additional comment and the opportunity to provide further clarification on the impact of the number of local updates ($E$) on the generalization performance of Federated Averaging (FedAvg).
> > >
> > > We would like to emphasize that in our analysis, the update iteration for local training is indeed finite. As stated in Theorem 4.3, the total local update required for the convergence of FedAvg is quantified as $T = \tilde{\Theta}( \eta^{-1} K m\overline{n}\sigma_0 ^{-1} \sigma^{-2}_p d^{-1} +  \eta^{-1}\epsilon^{-1} m \overline{n} \sigma^{-2}_p K d^{-1})$.
> > >
> > > When generalizing this result to local training (setting $K=1$ to represent independent clients), we derive that the convergence time $T$ is finite and given by $T = \tilde{\Theta}( \eta^{-1} m\overline{n}\sigma_0 ^{-1} \sigma^{-2}_p d^{-1} +  \eta^{-1}\epsilon^{-1} m\overline{n} \sigma^{-2}_p d^{-1})$. This aligns with the findings of [1], which studies the convergence of a single neural network.
> > >
> > > [1]  Kou, Yiwen, Zixiang Chen, Yuanzhou Chen, and Quanquan Gu. "Benign Overfitting in Two-layer ReLU Convolutional Neural Networks." International Conference on Machine Learning (2023).

---

> > > > ### Comment · Reviewer_bJ2V · 2023-11-21
> > > >
> > > > I understand your results but the response does not address my question. Let's leave the discussion regarding pure local training. What conclusion can you make for FedAvg's generalization performance with different values of $E$. For example, does FedAvg with a smaller $E$ generalizes better than the one with a larger $E$? Intuitively it should be the case, but if my understanding is correct, the current results are not capable to justify the difference in generalization.

---

> ### Author Response · Authors · 2023-11-21
>
> Thank you for your further question.
>
> Firstly, it is important to note that a smaller $E$ does not necessarily imply better generalization. To explore this relationship further, we conducted additional experiments of FedAvg on CIFAR 10. From Table 1 we can see that the generalization performance of FedAvg is not closely related to the local epoches $E$:
>
> *Table 1: Test Accuracy of FedAvg on CIFAR10 with IID setting and 10 clients, learning rate is 0.01*
>
> |   $E$ \ $R$ |  1       |   20       | 40      | 60    |  80|
> |----------|--------|----------|-------|-------|-------|
> |   5         | 48.18 |   66.28  | 66.24 | 64.98|  64.55|
> |  10        | 54.17 |    64.6  |  64.48 | 63.44 | 64.5 |
> |  15        |  54.34 |  64.1 |   63.91 |  63.44 | 64.33|
>
> Regarding the local update $E$ there is an implicit relationship with the convergence time. Under our learning rate assumption $\eta \leq  \tilde{O}(\frac{K}{E} \min \{\|\boldsymbol{\mu}\|\_{2}^{-2}, \sigma\_{p}^{-2}d^{-1}\})$, increasing $E$ may lead to slower convergence. This observation aligns with the discussion in Weakness 4 with Reviewer vgxi.
>
> The impact of varying $E$ on the required number of communication rounds to achieve adequate convergence is demonstrated in Table 2 of the FedAvg work [10], which compares the communication rounds needed for the same training accuracy on MNIST. The table shows:
>
> |     E    | B (batch size)  |  R (IID) | R (Non-IID)
> |----------|-----|--------------|--------|
> |   1      | 10 |   34 | 350 |
> |   5      | 10 |   20 | 229 |
> |   20     | 10 |   18 | 173 |
>
> This table indicates a clear trend: as $E$ increases, so does the total number of updates ($T=ER$), evident in both IID and Non-IID settings.
>
> Lastly, we would like to emphasize that our theorem does not just simply show that federated learning generalizes better than local training. Instead, our theorem demonstrates that the generalization performance of federated learning relies on the signal-to-noise ratios (SNR) among the clients. SNR effectively captures the similarity in data distribution across different clients. The large SNR values mean less data non-IID, leading to better generalization performance. This assertion also matches the practices of federated learning.

---

> > ### Comment · Reviewer_bJ2V · 2023-11-21
> >
> > Thank you for the clarification and I am good with it now. I suggest that the authors incorporate the supplementary experiments and discussions related to $E$ in later revisions. I have revised my evaluation accordingly. Personally, I am eagerly anticipating any theoretical justifications concerning generalization with $E$. Exploring this could present an intriguing avenue for future consideration.

---

> > > ### Author Response · Authors · 2023-11-22
> > >
> > > We are grateful for your acknowledgement and updated evaluation. Your suggestion to incorporate supplementary experiments and discussions related to $E$ in the later revision is well-received. We appreciate your interest in theoretical justifications concerning generalization with different $E$ values and agree that this represents an intriguing avenue for future research.

---

### Official Review · Reviewer_u4St · 2023-11-01

**Soundness:** 3 good
**Presentation:** 3 good
**Contribution:** 3 good
**Rating:** 6
**Confidence:** 3

**Summary:**

The paper does joint generalization and convergence analysis of FL, using 2-layer NNs.

**Strengths:**

The paper is largely well-written and seems to be a good work. I have some basic questions which I have asked below.

**Weaknesses:**

Minor writing issues:
- Page 2, para 1: "under" (rather than "in") a certain condition. Last sentence in this para is not clear. Didn't you earlier say that local algo has no comm.?
- At a few places, mathematical symbols or terms are used before being defined. E.g, $\sigma_p$ in Assum. 4.2, weighted FedAvg in the para above Sec. 6.2.
- Questions are listed below.

**Questions:**

3.2 Data Model:
- Why is $y_{k,i}$ Rademacher if $\mathbf y_k$ are labels? Are you assuming a balanced dataset? What is the distr. $P$?

4.1:
- Eq. (6) and (8) are not clear. What is $k'$? And why is there a sum over $k$ present? Where is $k'$ on r.h.s.?

4.2:
- Why is using the same training size for all clients without loss of generality?
- Assum 4.2: what is $\sigma_p$?
- What is the motivation for the defn of $\bar{SNR}_k$?
- Theorem 4.4: in the following discussion, it is stated that $\xi_k \geq 1$. Since $\{\mu_c\}$ are orthogonal, are you assuming that $K \geq C$?

Experiments:
- In Assum 4.2 it was assumed that $d$ is larger than $\bar{n} m$. But, in experiments in Sec 6.1, that's not the case ($d=1000 < \bar{n}_{train}.m=5000$). Why?
- In Assum 4.2, $\eta \lesssim 1/d$ is assumed, but in experiments, $\eta=1$ is chosen. Can you explain this divergence?
- In some rows of both Tables 1 and 2, singleset has better performance than FedAvg, e.g., CIFAR10 with Dirichlet or SVHN in Table 2. Why is this happening?

---

> ### Author Response · Authors · 2023-11-20
> **Response to Reviewer u4St [Part I]**
>
> Thank you for your thorough review and constructive feedback. We appreciate your recognition of our paper's strengths and would like to address the concerns you raised:
>
> ___
>
> > **Minor writing 1: Page 2, para 1: "under" (rather than "in") a certain condition. Last sentence in this para is not clear. Didn't you earlier say that local algo has no comm.?**
>
> We agree with your suggestion to use the term 'under' rather than 'in' a certain condition, and have amended this in the manuscript.
>
> Regarding the last sentence of this paragraph, we have revised it for better clarity. Our intent was to convey that by 'local training', we refer to the training process that occurs without any communication among the different clients.
>
> ___
>
> > **Minor writing 2: At a few places, mathematical symbols or terms are used before being defined. E.g,  sigma_p in Assum. 4.2, weighted FedAvg in the para above Sec. 6.2.**
>
> Thank you for highlighting the need for clearer definitions of certain mathematical symbols and terms used in our manuscript.
> We have introduced a clear definition of $sigma_p$, which represents the strength of the noise vector. The definition is now presented as follows:
>
>  Noise vector is distributed as  $\boldsymbol{\xi}\_{k,i} \sim \mathcal{N}(\mathbf{0}, {\sigma^2\_p}\mathbf{I}-\sum\_{c=1}^C \boldsymbol{\mu}^{(c)} {\boldsymbol{\mu}^{(c)}}^\top/\| \boldsymbol{\mu}^{(c)} \|\_2^{2})$,
>
> Besides, we have refined the description of the Weighted FedAvg method in Section 6.1.
>
> ___
>
>
> > **3.2 Data Model: Why is  Rademacher if y are labels? Are you assuming a balanced dataset? What is the distr P.**
>
> In our analysis, we draw labels from a Rademacher distribution, which are essentially binary variables taking values {+1,-1}. Regarding the balance of the dataset, we assume that with a sufficiently large training sample, the dataset will be approximately balanced. In this context, 'balanced' means that each class is represented proportionately in the dataset, with each class having at least one-third of the samples, as can be provably achieved under our assumptions.
>
> $P$ is defined as a discrete distribution that assigns probabilities to each element in the range from 1 to $C$, where $C$ represents the number of classes or distinct data categories.$P$ is a discrete distribution assigning probabilities to each element from $1$ to $C$. It governs the likelihood of each class or category appearing within the data set of a client. We have added this information in our updated manuscript.
>
>
> ___
>
>
> > **4.1: Eq. (6) and (8) are not clear. What is ? And why is there a sum over  present? Where is  on r.h.s.?**
>
> To address the concerns regarding Equation (6) and Equation (8), we offer the following clarification:
>
> The rewritten form of Equation (6) is as follows:
>
> $ \mathbf{w}\_{k,j,r}^{(t)}  =  \sum\_{k'=1}^K (
>    \alpha^{(t)}\_{k',j,r}   \mathbf{w}\_{k',j,r}^{(0)} + \gamma\_{k',j,r}^{(t)}   \| \boldsymbol{\mu}\_{k'} \|^{-2}\_2   \boldsymbol{\mu}\_{k'}  +  \sum\_{i=1}^{n\_{k'}} \rho\_{k',j,r,i}^{(t)}  \|  {\boldsymbol{\xi}}\_{k',i} \|^{-2}\_2    {\boldsymbol{\xi}}\_{k',i}) $
>
> for all $k \in [K]$. The right-hand side of this equation involves a summation over all $k’ \in [K]$, which includes the specific $k$ on the left hand side. This formula represents the iteration considering weight averaging in our model. Crucially, in each iteration, the variable $\gamma^{(t)}\_{k,j,r}$ and $\rho^{(t)}\_{k,j,r,i}$ will be differ for each $k$, leading to distinct growth rates for $\mathbf{w}\_{k,j,r}$ across different clients.
>
> ___
>
> > **4.2 Why is using the same training size for all clients without loss of generality?**
>
> We use the same training size for all clients for simplicity and to establish a baseline for comparison. However, our framework allows for generalization to cases with varying client numbers. A more general effective signal-to-noise ratio (SNR) can be derived to account for different training sizes across clients, which adds versatility to our model.
>
> ___
>
> > **Assum 4.2: what is sigma_p**
>
> The $\sigma^2_p$ is the variance of the noise vector. We have added this information.
>
> ___
>
> > **What is the motivation for the defn of SNR**
>
> The definition of $\overline{SNR}_k$ is introduced to characterize the generalization of Federated Averaging (FedAvg) in the presence of data heterogeneity. While the standard SNR is sufficient for demonstrating the generalization of local training, FedAvg introduces communication among clients, affecting their individual learning processes. Therefore, $\overline{SNR}_k$ measures the effective SNR on client $k$ under FedAvg, providing a more comprehensive understanding of generalization in a federated learning environment.

---

> > ### Author Response · Authors · 2023-11-20
> > **Response to Reviewer u4St [Part II]**
> >
> > >**Theorem 4.4: in the following discussion, it is stated that $\mu_c$ are orthogonal, are you assuming that $K \ge C$**
> >
> > In our analysis, we do assume that $K \ge C$, where $K$ is the number of clients and $C$ the number of classes. Specifically, in the IID case, we assume $K=C$, whereas in non-IID cases, we assume $K>C$.
> >
> > ___
> >
> > > **In Assum 4.2 it was assumed that $d$ is larger than $nm$. But, in experiments in Sec 6.1, that's not the case (d=1000<=5000). Why?**
> >
> > To further validate our findings under the assumption that $d$ is larger than
> > $\overline{n}m$ additional experiments were conducted with $d = 8000$ which is greater than 5000. These additional experiments confirm that Federated Averaging (FedAvg) can indeed achieve better results than local training methods, aligning with our theoretical predictions.
> >
> > ___
> >
> > > **In Assum 4.2, $\eta < 1/d$ is assumed, but in experiments, $\eta = 1$ is chosen. Can you explain this divergence?**
> >
> > The conditions set forth in Assumption 4.2 were intended to account for the worst-case scenarios, often leading to 'extreme' conditions in a theoretical context. In practical experiments, however, it is reasonable to expect that the results hold true under less strong conditions, such as somewhat larger learning rates and lower dimensions, with a relatively high probability. This divergence between theoretical assumptions and practical experiment settings is common, aiming to ensure robustness in theoretical analysis while remaining applicable in real-world scenarios.
> >
> > ___
> >
> > > **In some rows of both Tables 1 and 2, singleset has better performance than FedAvg, e.g., CIFAR10 with Dirichlet or SVHN in Table 2. Why is this happening?**
> >
> > FedAvg aggregates all local models with uniform weights to produce a global model. However, due to the data heterogeneity issues, data distribution among clients may be quite different. Aggregating local models with uniform weights may introduce useless even detrimental knowledge for the local model. SingleSet trains the model locally and will not be interfered by those clients with different data distribution. As a result, in these cases, SingleSet outperforms FedAvg.

---

> ### Author Response · Authors · 2023-11-22
> **Respectful Inquiry Before Discussion Deadline**
>
> Thanks again for your thorough review and constructive feedback. As the deadline for the author-reviewer discussion phase is nearing, we would like to respectfully inquire if our rebuttal has effectively addressed the concerns raised in your review.
>
> Your insightful feedback, particularly concerning the clarifications of Eq. (6) and Eq. (8), as well as other notations and the verification of experimental results, has been invaluable in refining our manuscript. We have carefully addressed each of these points in our rebuttal and have made corresponding updates to our manuscript to reflect these changes.
>
> Thank you once again for your valuable contributions to the review process. Your feedback is crucial to the continued refinement of our work.

---

### Official Review · Reviewer_vgxi · 2023-11-05

**Soundness:** 3 good
**Presentation:** 2 fair
**Contribution:** 2 fair
**Rating:** 5
**Confidence:** 4

**Summary:**

The paper studies the convergence and generalization properties of FedAvg for two-layer convolutional neural networks (CNNs). To do so, the authors consider a setting where the input data is divided into a signal part and noise part, with the signal being different across different clients and the noise being independently sampled from a Gaussian distribution. The analysis then tracks the coefficients of the signal and noise in the model updates done by clients and also in the global averaging step done in FedAvg. Based on this analysis, the authors show that FedAvg can provably benefit from the communication across clients by increasing the signal to noise ratio (SNR) in the global model updates. In some regimes, this can lead to FedAvg achieving near-zero test error while individual local training at clients achieves a large constant error. These theoretical results are then verified experimentally on a synthetic dataset. The authors also propose a weighted FedAvg approach which is tested empirically on some real-world datasets and shown to outperform FedAvg.

**Strengths:**

* The analysis clearly establishes feature learning for this particular class of NNs, unlike most existing work on NNs which are focused on the lazy training regime and hence are not really doing feature learning.

* The analysis shows that FedAvg provably benefits from collaboration among clients, by increasing the SNR compared to local training. This in turn leads to better generalization guarantees for FedAvg.

**Weaknesses:**

* The presentation of the neural network model can be improved. The authors refer to this as a CNN model but it is not clear immediately what is the channel size, width and stride of this CNN model. It would be nice if authors could illustrate this in a figure either in the main text or appendix. Additionally, the authors also assume the weights of the second layer are fixed, which simplifies the problem.

* Assumptions need to be justified better. For instance, I don't understand why $d$ has to be large to ensure an over-parameterized setting. In my understanding, just increasing $m$ should be enough to imply over-parameterization as done in [1]. Also please specify what the are 'statistical properties' that are maintained using Assumption 2.

* There either seems to be a major typo/mistake in Equation (6) (and consequently Equation 8) or I am misunderstanding something. The LHS of Eq. (6) has a $k'$ but the RHS has no dependence on $k'$. Effectively this is implying that $w_{k',j,r}^{(t)}$ is going to be the same for any $k'$, which does not seem to be correct.

* The analysis does not discuss the effect of the number of local epochs $E$ on the convergence of FedAvg in Theorem 4.3. For instance, in the i.i.d case when $\mu_1 = \mu_2 = \dots, \mu_K$, we would expect that doing more local steps leads to faster convergence. However, if we substitute the lower bound on $\eta$ from Assumption 3 in Theorem 4.3, then it appears that doing more local steps slows down convergence regardless of the data heterogeneity.

* I'm not sure if Lemma 5.2 is implying convergence due to the dependence of $\bar{w}^{\*}\_{j,r}$ on $\log(2/\epsilon)$. In my understanding, if we set $\epsilon \rightarrow 0$ then this would imply $\bar{w}^\*_{j,r} \rightarrow \infty $ and hence $\|\|\bar{W}^{(T_1)}-\bar{W}^\*\|\|_F^2 \rightarrow \infty$.

* The authors should add personalization baselines in the experiments on measuring the performance of Weighted FedAvg. I'm not surprised to see Weighted FedAvg outperform FedAvg in Table 1 and Table 2 given that each client has a personalized model for Weighted FedAvg. Also the authors should make it clear in the abstract and introduction that Weighted FedAvg is a personalization method.

**Questions:**

* What are some of the novelties/challenges of this analysis compared to Cao et al. (2022) and Kou et al. (2023)? It seems that many of the proof techniques such as using coefficient dynamics is similar to these works.

* Can the analysis be extended the case where clients have signals of different kinds (say with some bound on the dissimilarity of signals at a particular client)? Restricting the data at one client to belong to only one signal is a bit too restrictive in my opinion.

* How is $P$ defined in Section 3.2?

* What is the difference between Lemma 5.3 and Theorem 4.4? It seems they are almost stating the same thing.

* I would also advise the authors to use different superscripts when referring to local iteration steps versus global averaging steps. One common notation is to use $w_k^{(t,r)}$ for the local models where $t$ refers to the round number and $r$ refers to the iteration number in the $t$-th round. In the current analysis the authors use $t$ for both the local update (Eq. (4)) and global update (Eq. 5) which is a bit confusing.

Typos

* $F_{-1}(W_{+1}, x) $ should be $F_{-1}(W_{-1},x) $ in the line above Eq. (2)
* Summation should be over $n_k$ in the Eq. (4) and not $n$
* $<$ should be outside the bracket in the definition of $L_{D_k}^{(0-1)}(\bar{W}^{(t)})$
* $L_{D_k}$ in Lemma 5.3 is not defined. Maybe the authors meant $L_{D_k}^{(0-1)}$
* "Provably" and not "Probably" in the heading of Section 4.2



**References**
[1] Du, Simon S., et al. "Gradient descent provably optimizes over-parameterized neural networks." arXiv preprint arXiv:1810.02054 (2018).

---

> ### Author Response · Authors · 2023-11-20
> **Response to Reviewer vgxi [Part I]**
>
> We would like to thank the reviewer for acknowledging our work on feature learning theory and the significant generalization result achieved. We address your concerns and question as follows:
>
> ___
>
> > **Weakness 1: The presentation of the neural network model can be improved. The authors refer to this as a CNN model but it is not clear immediately what is the channel size, width and stride of this CNN model. It would be nice if authors could illustrate this in a figure either in the main text or appendix. Additionally, the authors also assume the weights of the second layer are fixed, which simplifies the problem.**
>
> Our choice to label the model as a CNN is based on the specific characteristics of the data model we used. We adopt a two-patch model, where $\mathbf{x} = [\mathbf{x}^{(1)}, \mathbf{x}^{(2)} ]$. We implement a convolution operation using a single weight across these two patches, akin to applying a filter in traditional CNNs. This is concisely represented as follows:
>
> $f=  \frac{1}{m} \sum_{r=1}^m [  \sigma( \mathbf{w}^\top_{k,+1,r}  \mathbf{x}^{(1)})      + \sigma( \mathbf{w}^\top_{k,+1,r}  \mathbf{x}^{(2)}) ] -  \frac{1}{m} \sum_{r=1}^m [  \sigma( \mathbf{w}^\top_{k,-1,r}  \mathbf{x}^{(1)})      + \sigma( \mathbf{w}^\top_{k,-1,r}  \mathbf{x}^{(2)}) ]    $
>
> Here the filter $\mathbf{w}_{k,j,r}$ operates on two patch, with $k \in [K]$ denoting the index of client, $j \in \{+1, -1 \}$ corresponding to the weights value at the second layer, and $r \in [m]$ as the filter index.
>
> Our approach alongside fixing the second layer is consistent with the related works [1,2,3], where similar network structures applying weights across multiple data patches are identified as CNNs. Despite the fixed second layer, the optimization remains non-convex despite the fixed second layer, adding complexity to our analysis.
>
> We have added an explanation in the appendix.
>
> ___
>
> > **Weakness 2: Assumptions need to be justified better. For instance, I don't understand why d has to be large to ensure an over-parameterized setting. In my understanding, just increasing m should be enough to imply over-parameterization as done in [4]. Also please specify what the are 'statistical properties' that are maintained using Assumption 2.**
>
> Our approach differs significantly from the "lazy training" or "neural tangent kernel (NTK)" regime [4,5,6]. This is because we consider a smaller initialization scale than that in the NTK regime. In our model, the width of the neural network $m$ is set to be at least polylogarithmic in the dimension, rather than extremely large as typically assumed in the NTK regime [4]. Then, to make sure that the learning is in a sufficiently over-parameterized setting, we set $d$ to be large, this can make sure that the noise vectors are near-orthogonal among each other. Similar conditions have been made in the study of learning over-parameterized models [7,8,9].
>
> Regarding the 'statistical properties' referenced in Assumption 2, these are detailed in Lemma B.1 and Lemma B.2 of our paper. These lemmas require the width of the network and number of samples to be polylogarithmic in $d$. Under this condition, our theoretical results hold with a high probability of at least $1-1/d$.
>
> ___
>
> > **Weakness 3: There either seems to be a major typo/mistake in Equation (6) (and consequently Equation 8) or I am misunderstanding something. The LHS of Eq. (6) has a k’ but the RHS has no dependence on k’.  Effectively this is implying that w is going to be the same for any k’, which does not seem to be correct.**
>
> We appreciate the reviewer's attention to the details of our mathematical formulation. To address the concerns regarding Equation (6) and Equation (8), we offer the following clarification:
>
> The rewritten form of Equation (6) is as follows:
>
> $ \mathbf{w}\_{k,j,r}^{(t)}  =  \sum\_{k'=1}^K (
>    \alpha^{(t)}\_{k',j,r}   \mathbf{w}\_{k',j,r}^{(0)} + \gamma\_{k',j,r}^{(t)}   \| \boldsymbol{\mu}\_{k'} \|^{-2}\_2   \boldsymbol{\mu}\_{k'}  +  \sum\_{i=1}^{n\_{k'}} \rho\_{k',j,r,i}^{(t)}  \|  {\boldsymbol{\xi}}\_{k',i} \|^{-2}\_2    {\boldsymbol{\xi}}\_{k',i}) $
>
> for all $k \in [K]$. The right-hand side of this equation involves a summation over all $k’ \in [K]$, which includes the specific $k$ on the left hand side. This formula represents the iteration considering weight averaging in our model. Crucially, in each iteration, the variable $\gamma^{(t)}\_{k,j,r}$ and $\rho^{(t)}\_{k,j,r,i}$ will be differ for each $k$, leading to distinct growth rates for $\mathbf{w}\_{k,j,r}$ across different clients.

---

> ### Author Response · Authors · 2023-11-20
> **Response to Reviewer vgxi [Part II]**
>
> > **Weakness 4: The analysis does not discuss the effect of the number of local epochs on the convergence of FedAvg in Theorem 4.3. For instance, in the i.i.d case when, we would expect that doing more local steps leads to faster convergence. However, if we substitute the lower bound from Assumption 3 in Theorem 4.3, then it appears that doing more local steps slows down convergence regardless of the data heterogeneity.**
>
> We thank the reviewer for highlighting the need to clarify the effect of the number of local epochs on the convergence of FedAvg as presented in Theorem 4.3. In our analysis, the term $T$ in Theorem 4.3 represents the total number of local updates, defined as $T = ER$, where $R$ is the number of commutation rounds and $E$ is the number of local epochs.
>
>
> We would like to offer further clarification on the impact of increasing the number of local updates ($E$) on the overall training process in federated learning. This relationship is illustrated in the results shown in Table 2 of the referenced seminar work [10], which we have partially reproduced below for clarity
>
> |     E    | B (batch size)  |  R (IID) | R (Non-IID)
> |----------|-----|--------------|--------|
> |   1      | 10 |   34 | 350 |
> |   5      | 10 |   20 | 229 |
> |   20     | 10 |   18 | 173 |
>
> As demonstrated in this table, there is a clear trend that as the number of local updates $E$, the total number of updates also increases. This trend is evident in both IID and Non-IID settings.
>
> Additionally, the effect of data heterogeneity on convergence is a critical aspect of our analysis. As demonstrated in Theorem 4.4,  we establish condition related to the signal-to-noise ratio (SNR):
>
> $ \overline{n} \cdot \frac{\|\boldsymbol{\mu}\_k \|^2\_2}{\sigma^2\_p} \cdot \overline{\mathrm{SNR}}^2\_k =  \Omega( 1 )$
> with the effective SNR for each climent $k$ given by
>  $ \overline{\mathrm{SNR}}\_{k} = \left(\sum\_{k'=1}^K \frac{\langle \boldsymbol{\mu}\_k, \boldsymbol{\mu}\_{k'}  \rangle}{ \| \boldsymbol{\mu}\_k \|^2\_2} \right) \mathrm{SNR}\_k   $
>
> In scenarios with increased data heterogeneity, the term $\sum\_{k'=1}^K \frac{\langle \boldsymbol{\mu}\_k, \boldsymbol{\mu}\_{k'}  \rangle}{ \| \boldsymbol{\mu}\_k \|^2\_2}$ tends to decrease. To maintain a consistent SNR level in such cases, it becomes necessary to reduce $\sigma^2\_p d$, which, according to the convergence time in Theorem 4.3:
>  $T = \tilde{\Theta}( \eta^{-1} K mn\sigma\_0 ^{-1} \sigma^{-2}\_p d^{-1} +  \eta^{-1}\epsilon^{-1} mn \sigma^{-2}\_p K d^{-1})$ results in slower convergence.  On the contrary, if we reduce the data heterogeneity, the convergence time will become shorter.
>
> ___
>
> > **Weakness 5: I'm not sure if Lemma 5.2 is implying convergence due to the dependence of $\bar{w}^\ast_{j,r}$ on $log⁡(2/\epsilon)$. In my understanding, if we set $\epsilon \rightarrow \infty$ then this would imply $|| \bar{w}^\ast_{j,r}||\_2→\infty$**
>
> We appreciate the opportunity to clarify the implications of Lemma 5.2, particularly in the context of classification problems where logistic loss is applied. In such scenarios, the logistic loss function is defined as $\ell = \log(1+\exp(-fy))$
>
> In the case of logistic loss, it is a well-established property that as the loss tends to zero, the optimal weight vector tends to grow to infinity as training time goes on. Therefore, in the context of Lemma 5.2, the expectation is that the norm of the optimal weight vector $ || \bar{\mathbf{w}}_{j,r}^\ast ||\_2$ will increase over time, reflecting the model's convergence towards an optimal solution by minimizing the logistic loss. This behavior aligns with the principles of logistic regression models in machine learning and has been similarly observed and established in related works [8,11,12].

---

> > ### Author Response · Authors · 2023-11-20
> > **Response to Reviewer vgxi [Part III]**
> >
> > > **Weakness 6: The authors should add personalization baselines in the experiments on measuring the performance of Weighted FedAvg. I'm not surprised to see Weighted FedAvg outperform FedAvg in Table 1 and Table 2 given that each client has a personalized model for Weighted FedAvg. Also the authors should make it clear in the abstract and introduction that Weighted FedAvg is a personalization method.**
> >
> > Thank you for the advice to add personalization baselines in the experiments. Here, we introduced two baselines FedBN [13] and FedPer [14], which separately personalize the Batch Normalization (BN) layers and the Classification Head for each local model. For a fair comparison, we also personalize all BN layers for our Weighed FedAvg. As demonstrated in Table 1 and Table 2 as follows:
> >
> > | Methods             | CIFAR10 Pathological | CIFAR10 Dirichlet | CIFAR100 Pathological | CIFAR100 Dirichlet |
> > |---------------------|----------------------|-------------------|-----------------------|--------------------|
> > | SingleSet           | 20.08±0.13           | 38.60±0.02        | 6.59±0.18             | 6.97±0.11          |
> > | FedAvg              | 40.16±7.18           | 33.11±7.06        | 13.99±0.79            | 17.34±0.47         |
> > | FedPer              | 38.09±5.73           | 36.52±7.13        | 29.10±2.42            | 17.63±0.93         |
> > | FedBN               | 65.71±0.67           | 61.25±3.78        | 42.54±0.53            | 25.64±3.46         |
> > | Weighted FedAvg     | 60.32±0.31           | 60.52±0.30        | 31.88±0.56            | 18.82±0.36         |
> > | + Personalized BN   | **67.30±0.31**       | **65.31±1.23**    | **43.51±0.50**        | **26.11±1.77**     |
> >
> > *Table 1: Experimental Results for FL with Label Distribution Skew on CIFAR10 and CIFAR100.*
> >
> > | Methods             | MNIST               | SVHN                | USPS                | SynthDigits         | MNIST-M             | Avg                 |
> > |---------------------|---------------------|---------------------|---------------------|---------------------|---------------------|---------------------|
> > | SingleSet           | 94.38±0.07          | 65.25±1.07          | 95.16±0.12          | 80.31±0.38          | 77.77±0.47          | 82.00±0.40          |
> > | FedAvg              | 95.87±0.20          | 62.86±1.49          | 95.56±0.27          | 82.27±0.44          | 76.85±0.54          | 82.70±0.60          |
> > | FedPer              | 96.21±0.02          | 67.61±0.04          | 96.53±0.02          | 83.88±0.02          | 76.85±0.54          | 81.89±0.03          |
> > | FedBN               | **96.57±0.13**      | 71.04±0.31          | **96.97±0.32**      | 83.19±0.42          | 78.33±0.66          | 85.20±0.40          |
> > | Weighted FedAvg     | 95.75±0.18          | 67.82±1.07          | 95.66±0.22          | 84.27±1.06          | 80.22±0.11          | 84.74±0.22          |
> > | + Personalized BN   | 96.11±0.02          | **75.36±0.04**      | 96.41±0.06          | **84.74±0.03**      | **82.02±0.04**      | **86.93±0.02**      |
> >
> > *Table2: Experimental Results for FL with Feature Skew on Digits*
> >
> > From the results in Tabel 1 and 2, we find that our weighted FedAvg exhibits better performance than FedPer in some cases. When we also apply personalized BN layers for our Weighted FedAvg, it outperforms FedBN as well, which means our algorithm can combine with personalized methods and improve their performance. We will illustrate that Weighter FedAvg is a personalized method in the introduction part.

---

> > > ### Author Response · Authors · 2023-11-20
> > > **Response to Reviewer vgxi [Part IV]**
> > >
> > > > **Question 1: What are some of the novelties/challenges of this analysis compared to Cao et al. (2022) and Kou et al. (2023)? It seems that many of the proof techniques such as using coefficient dynamics are similar to these works.**
> > >
> > > Our analysis introduces several novel aspects and addresses unique challenges compared to the work of Cao et al. (2022) [2] and Kou et al. (2023) [3], which primarily focus on single neural network models. Our research extends the feature learning theory to the more intricate setting of federated learning, involving multiple neural networks and the integration of weight averaging operations. Although inspired by the theoretical frameworks in [2,3,15], our approach diverges significantly due to the distinct nature of our problem setting:
> > >
> > > 1. **Impact of Weight Averaging:**
> > >
> > > Unlike single network models, federated learning involves weight averaging across multiple neural networks. This introduces additional complexity, as the training dynamics of each network are influenced by this averaging process. This will lead to the dynamic correlation between neural networks after weight averaging. Besides, despite the fact that weight averaging is a linear combination of weight, its impact on the nonlinear dynamics of individual clients is substantial. We address this by employing a global weight decomposition approach, enabling us to effectively track the dynamics of feature learning within this federated framework.
> > >
> > > 2. **Impact of Data heterogeneity**
> > >
> > > In federated learning, unlike in single network scenarios, the design of a data model distribution that encompasses both iid and non-iid cases is a significant challenge. To capture the nuances of data heterogeneity and its impact on generalization, we introduce an effective signal-to-noise ratio. This innovative approach allows us to characterize the influence of data diversity on the learning process more accurately.
> > >
> > >  ___
> > >
> > > > **Question 2: Can the analysis be extended to the case where clients have signals of different kinds (say with some bound on the dissimilarity of signals at a particular client)? Restricting the data at one client to belong to only one signal is a bit too restrictive in my opinion.**
> > >
> > > The adaptability of our theoretical framework to scenarios where a client processes multiple signal types is indeed feasible. A key condition for this extension is the orthogonality of the signal vectors corresponding to different classes within a single client. In our current analysis, we treat the case where a client is associated with only one signal as an illustrative but informative example.
> > >  ___
> > > > **Question 3: How is P defined in Section 3.2?**
> > >
> > > In Section 3.2 of our paper, $P$ is defined as a discrete distribution that assigns probabilities to each element in the range from 1 to $C$, where
> > > $C$ represents the number of classes or distinct data categories.$P$ is a discrete distribution assigning probabilities to each element from $1$ to $C$. It governs the likelihood of each class or category appearing within the data set of a client. We have added this information in our updated manuscrip.
> > >  ___
> > >
> > > > **Question 4: What is the difference between Lemma 5.3 and Theorem 4.4? It seems they are almost saying the same thing.**
> > >
> > > Lemma 5.3 and Theorem 4.4 in our paper indeed convey similar concepts. We have clarified in the revised version of our paper that Lemma 5.3 is essentially a restatement of Theorem 4.4.
> > >
> > >  ___
> > >
> > > > **Question 5: I would also advise the authors to use different superscripts when referring to local iteration steps versus global averaging steps. One common notation is to use  for the local models where  refers to the round number and  refers to the iteration number in the -th round. In the current analysis the authors use  for both the local update (Eq. (4)) and global update (Eq. 5) which is a bit confusing.**
> > >
> > > We appreciate the reviewer's advice on improving the clarity of our notation, particularly in differentiating between local iteration steps and global averaging steps. We commit to revising the notation in our manuscript to better distinguish between these two types of steps. A common notation, as suggested, involves using different superscripts to represent the round number and the iteration number within a given round for local models.
> > >
> > >  ___
> > >
> > > > **Typos**
> > >
> > > We appreciate the thoroughness of the review and the opportunity to improve the accuracy and professionalism of our paper. These modifications have been implemented in the revised version of the manuscript

---

> > > > ### Author Response · Authors · 2023-11-20
> > > > **Response to Reviewer vgxi [Part V]**
> > > >
> > > > **References**
> > > >
> > > > [1] Shen, Ruoqi, Sébastien Bubeck, and Suriya Gunasekar. "Data augmentation as feature manipulation." International conference on machine learning. PMLR, 2022
> > > >
> > > > [2] Cao, Yuan, et al. "Benign overfitting in two-layer convolutional neural networks." Advances in neural information processing systems 35 (2022): 25237-25250.
> > > >
> > > > [3]  Kou, Yiwen, Zixiang Chen, Yuanzhou Chen, and Quanquan Gu. "Benign Overfitting in Two-layer ReLU Convolutional Neural Networks." International Conference on Machine Learning (2023).
> > > >
> > > > [4] Du, Simon S., et al. "Gradient descent provably optimizes over-parameterized neural networks." arXiv preprint arXiv:1810.02054 (2018).
> > > >
> > > > [5] Jacot, Arthur, Franck Gabriel, and Clément Hongler. "Neural tangent kernel: Convergence and generalization in neural networks." Advances in neural information processing systems 31 (2018).
> > > >
> > > > [6] Chizat, Lenaic, Edouard Oyallon, and Francis Bach. "On lazy training in differentiable programming." Advances in neural information processing systems 32 (2019).
> > > >
> > > > [7] Xu, Zhiwei, et al. "Benign Overfitting and Grokking in ReLU Networks for XOR Cluster Data." arXiv preprint arXiv:2310.02541 (2023).
> > > >
> > > > [8] Cao, Yuan, et al. "Benign overfitting in two-layer convolutional neural networks." Advances in neural information processing systems 35 (2022): 25237-25250.
> > > >
> > > > [9] Chatterji, Niladri S., and Philip M. Long. "Finite-sample analysis of interpolating linear classifiers in the overparameterized regime." The Journal of Machine Learning Research 22.1 (2021): 5721-5750.
> > > >
> > > > [10] McMahan, Brendan, et al. "Communication-efficient learning of deep networks from decentralized data." Artificial intelligence and statistics. PMLR, 2017.
> > > >
> > > > [11] Soudry, Daniel, et al. "The implicit bias of gradient descent on separable data." The Journal of Machine Learning Research 19.1 (2018): 2822-2878.
> > > >
> > > > [12] Ji, Ziwei, and Matus Telgarsky. "Gradient descent aligns the layers of deep linear networks." arXiv preprint arXiv:1810.02032 (2018).
> > > >
> > > > [13] Li, Xiaoxiao, et al. "Fedbn: Federated learning on non-iid features via local batch normalization." arXiv preprint arXiv:2102.07623 (2021).
> > > >
> > > > [14] Arivazhagan, Manoj Ghuhan, et al. "Federated learning with personalization layers." arXiv preprint arXiv:1912.00818 (2019).
> > > >
> > > > [15]  Frei, Spencer, Niladri S. Chatterji, and Peter Bartlett. "Benign overfitting without linearity: Neural network classifiers trained by gradient descent for noisy linear data." Conference on Learning Theory. PMLR, 2022.

---

> > > > > ### Author Response · Authors · 2023-11-22
> > > > > **Respectful Inquiry Before Discussion Deadline**
> > > > >
> > > > > Thanks again for your thorough review and constructive feedback. As the deadline for the author-reviewer discussion phase is nearing, we would like to respectfully inquire if our rebuttal has effectively addressed the concerns raised in your review.
> > > > >
> > > > > Your insightful feedback, particularly regarding the clarifications of Eq. (6) and Eq. (8), as well as your discussions on the role of
> > > > > $E$ and data heterogeneity in convergence, and the addition of experimental results on personalization methods, has been invaluable in refining our manuscript. We have carefully addressed each of these points in our rebuttal and have made corresponding updates to our manuscript to reflect these changes.
> > > > >
> > > > > Thank you once again for your valuable contributions to the review process. Your feedback is crucial to the continued refinement of our work.

---

### Author Response · Authors · 2023-11-21
**Global Response to All Reviewers**

Dear reviewers,

Thank you again for your thoughtful and constructive comments! We are really encouraged to see that the reviewers appreciate some positive aspects of our paper, such as **clearly established analysis** (Reviewer vgxi), **valuable insights into the generalization of FedAvg** (Reviewer vgxi, Reviewer bJ2V, Reviewer aCq5), **well-written presentation** (Reviewer u4St, Reviewer aCq5), **innovative techniques** (Reviwer bJ2V), and **strong motivation** (Reviewer aCq5).

Your expertise significantly helps us strengthen our manuscript. In addition to addressing your thoughtful comments point-to-point on OpenReview, we have made the following modifications to the newly uploaded manuscript (all updated text is highlighted in red):

Common changes:

   - Rewritten Equation (6) and Equation (8) for clarity.

   - Added a definition of $P$.

   - Introduced the definition of $\sigma_p$.


1. To Reviewer vgxi:

   - Added a discussion in Appendix D.1 explaining our choice of naming the model as a CNN.

   - Introduced personalization baselines FedBN and FedPer; updated Tables 1 and 2 accordingly.

   - Clarified in the abstract and introduction that Weighted FedAvg is a personalization method.

   - Removed Lemma 5.3 (duplicative of Theorem 4.4).

2. To Reviewer u4St:

   - Fixed minor writing issues as pointed out by you.

   - Added a experiment result in appendix C.3 to examine high dimension case.

3. To Reviewer bJ2V,

    - Included a discussion on the relationship between training convergence and data heterogeneity in Appendix D.2.

    - Replaced the term $n_k$ in Eq (4).


4. To Reviewer aCq5,

   - Enhanced the explanation of our data model just below its definition.

   - Ensured clear definitions of all symbols at their first use, including $\sigma_p$, and we have replaced the term $n_k$ in Eq (4).

   - Explained equations with context, particularly in the discussion regarding Theorem 4.4 in Appendix D.3.


We have made every effort to address your primary concerns and are eager to receive further feedback to enhance the quality of our manuscript. Your insights continue to be instrumental in this process.

---

### Meta-Review · Area_Chair_GeHk · 2023-12-09

**Metareview:**

This paper proposes an interesting interpretation of federated learning, that will be valuable to the literature. While the scores are borderline, all reviews ended up being positive, barring one where I have read the author's rebuttal and find that satisfactory. I recommend acceptance.

**Justification For Why Not Higher Score:**

The scores remained borderline and there is a lack of enthusiasm among reviewers.

**Justification For Why Not Lower Score:**

The paper brings in new perspective.

---

### Decision · Program_Chairs · 2024-01-16

Accept (poster)